# Learned Reference-based Diffusion Sampling for Multi-modal Distributions

**Maxence Noble**[*], **Louis Grenioux**[*], **Marylou Gabrié & Alain Oliviero Durmus**
CMAP, CNRS
École polytechnique, Institut Polytechnique de Paris
91120 Palaiseau, France

## Abstract

Over the past few years, several approaches utilizing score-based diffusion have been proposed to sample from probability distributions, that is without having access to exact samples and relying solely on evaluations of unnormalized densities. The resulting samplers approximate the time-reversal of a noising diffusion process, bridging the target distribution to an easy-to-sample base distribution. In practice, the performance of these methods heavily depends on key hyperparameters that require ground truth samples to be accurately tuned. Our work aims to highlight and address this fundamental issue, focusing in particular on multi-modal distributions, which pose significant challenges for existing sampling methods. Building on existing approaches, we introduce *Learned Reference-based Diffusion Sampler* (LRDS), a methodology specifically designed to leverage prior knowledge on the location of the target modes in order to bypass the obstacle of hyperparameter tuning. LRDS proceeds in two steps by (i) learning a *reference* diffusion model on samples located in high-density space regions and tailored for multimodality, and (ii) using this reference model to foster the training of a diffusion-based sampler. We experimentally demonstrate that LRDS best exploits prior knowledge on the target distribution compared to competing algorithms on a variety of challenging distributions.

## 1 Introduction

We consider the problem of sampling from a probability density known up to a normalizing constant. More precisely, we consider a target distribution $\pi \in \mathscr{P}(\mathbb{R}^d)$ with probability density given by $x \mapsto \gamma(x)/\mathcal{Z}$, where $\gamma : \mathbb{R}^d \to \mathbb{R}_+$ can be pointwise evaluated and the normalizing constant $\mathcal{Z} = \int_{\mathbb{R}^d} \gamma(x)\mathrm{d}x$ is intractable. This problem appears in a wide variety of applications such as Bayesian statistics (Liu & Liu, 2001; Kroese et al., 2011), statistical mechanics (Krauth, 2006) or molecular dynamics (Stoltz et al., 2010). In particular, we are interested in sampling from *multimodal* distributions, *i.e.*, distributions whose density admits multiple local maxima, called *modes*. Finding the modes of such distributions is a notoriously hard problem, yet, maybe surprisingly, even if the location of the modes is known, sampling $\pi$ remains a very challenging problem (Noé et al., 2019; Pompe et al., 2020; Grenioux et al., 2023). In this work, we aim to address this specific issue and will assume that we have access to the location of the modes as prior information on $\pi$. However, we do not assume to have access *a priori* to ground truth samples from $\pi$.

**Annealed MCMC.** Markov Chain Monte Carlo (MCMC) samplers are among the most popular approaches for sampling. In particular, gradient-based methods based on discretizations of Langevin or Hamiltonian dynamics (Roberts & Tweedie, 1996; Neal, 2012; Hoffman & Gelman, 2014) are guaranteed to be efficient for high-dimensional target distributions that are log-concave or satisfy functional inequalities (Dalalyan, 2017; Durmus & Moulines, 2017). However, when applied to multi-modal distributions, these MCMC methods suffer from high mixing time, that dramatically increases with the magnitude of the energy barriers between the modes, see Appendix I.3 for an illustration. To alleviate this issue, *annealing* strategies have been proposed. They consist in bridging an easy-to-sample *base* distribution $\pi^{\text{base}}$ to the *target* distribution $\pi$ via intermediate distributions

---

[*]Both authors contributed equally.
Corresponding authors: {`maxence.noble-bourillot,louis.grenioux`}@polytechnique.edu

$\{\pi_k\}_{k=0}^K$ with $K \geq 1$, $\pi_0 = \pi^{\text{base}}$ and $\pi_K = \pi$. This sequence should form a smooth path in the distribution space, so that sampling from $\pi_{k+1}$ starting from samples from $\pi_k$ is relatively simple. Typically, up to normalizing constants again, the densities of these annealed distributions are chosen as geometric interpolation between $\pi^{\text{base}}$ and $\gamma$, and are sampled either sequentially (Neal, 2001; Del Moral et al., 2006) or in parallel (Swendsen & Wang, 1986). However, these approaches may be prone to mode switching, *i.e.*, the relative weight of the modes of the intermediate distributions may vary dramatically along the prescribed path (Woodard et al., 2009; Tawn et al., 2020; Syed et al., 2022). Hence, recovering the proportions of the different modes of $\pi$ is hard in practice.

Alternatively, the bridging distributions $\{\pi_k\}_{k=0}^K$ can be implicitly defined as the marginals of a diffusion process. A particular case of interest is the annealing path given by the time-reversal of a noising process, which is at the foundation of diffusion-based generative models (Sohl-Dickstein et al., 2015; Song et al., 2021; Ho et al., 2020), and has been proved to be robust to mode switching (Phillips et al., 2024). Following this approach, several diffusion-based samplers relying exclusively on Monte Carlo methods have recently shown promises for sampling from multi-modal distributions (Huang et al., 2024a;b; Grenioux et al., 2024). However, these non-parametric approaches require the repeated estimation of an intractable drift at each draw and at each annealing step.

**Annealed VI.** Due to their ability to amortize inference, Variational Inference (VI) techniques are a popular alternative to MCMC methods. Given a class of parameterized (variational) distributions that are easy to sample from, they aim to find the optimal parameters that minimize a fixed discrepancy metric with respect to $\pi$. Standard VI settings involve mean-field approximations (Wainwright & Jordan, 2008), mixture models (Arenz et al., 2023), and more recently normalizing flows (Rezende et al., 2014; Papamakarios et al., 2021), that typically optimize a reverse Kullback-Leibler (KL) objective. However, these methods struggle with multi-modal target distributions as they suffer from mode collapse (Jerfel et al., 2021; Blessing et al., 2024).

To tackle this issue, it has been proposed to follow the annealing paradigm by considering an extended target distribution with marginals corresponding to the $\{\pi_k\}_{k=0}^K$. This approach has been explored following explicit bridging paths, *i.e.*, $\{\pi_k\}_{k=0}^K$ admit tractable unnormalized densities (Wu et al., 2020; Arbel et al., 2021; Geffner & Domke, 2021; Matthews et al., 2022; Doucet et al., 2022; Geffner & Domke, 2023; Vargas et al., 2024), but the resulting samplers may suffer from mode switching, as explained above. On the other hand, a recent class of annealed VI samplers using implicit diffusion-based paths has emerged (Tzen & Raginsky, 2019; Holdijk et al., 2023; Pavon, 2022; Zhang & Chen, 2022; Berner et al., 2023; Vargas et al., 2023a;b; Zhang & Chen, 2022; Berner et al., 2023; Vargas et al., 2023b; Akhound-Sadegh et al., 2024; Phillips et al., 2024) and seems well-suited for multi-modal target distributions. As we will further evidence in the present work, the promising results obtained with the latter methods are nevertheless mitigated by the need for careful tuning of their hyperparameters, which requires access to ground truth samples.

**Contributions** Within this context, we propose the *Learned Reference-based Diffusion Sampler* (LRDS), a variational diffusion-based sampler specifically designed for multi-modal distributions, which extends works from Zhang & Chen (2022); Vargas et al. (2023a); Richter et al. (2023):
• In the multi-modal setting, we highlight the sensitivity of previous variational diffusion-based methods with respect to their hyperparameters, which restrains their use in practice.
• To address this limitation, LRDS leverages the knowledge of the mode locations following two approaches: (a) GMM-LRDS relies on Gaussian Mixture Models and is well suited for a large variety of target distributions while being relatively lightweight, and (b) EBM-LRDS, a more computationally intensive scheme, takes advantage of Energy-Based Models for harder sampling problems.
• We show that GMM-LRDS and EBM-LRDS outperform previous diffusion-based samplers on challenging multi-modal settings.

**Notation.** For any measurable space $(\mathsf{X}, \mathcal{X})$, we denote by $\mathscr{P}(\mathsf{X})$ the space of probability measures defined on $(\mathsf{X}, \mathcal{X})$. Unless specified, if $\mathsf{X}$ is a topological space, $\mathcal{X}$ is defined as the Borel $\sigma$-field of $\mathsf{X}$. For any $T > 0$, we denote by $\mathbf{C}_T = \mathrm{C}([0, T], \mathbb{R}^d)$ the space of continuous functions from $[0, T]$ to $\mathbb{R}^d$ endowed with the uniform topology; hence $\mathscr{P}(\mathbf{C}_T)$ is the set of continuous-time stochastic processes (or path measures) defined on $[0, T]$. For any $\mathbb{P} \in \mathscr{P}(\mathbf{C}_T)$, we denote by $\mathbb{P}^R$ its *time-reversal*, defined such that if $(X_t)_{t \in [0,T]} \sim \mathbb{P}$, then $(X_{T-t})_{t \in [0,T]} \sim \mathbb{P}^R$. For any $t \in [0, T]$, we denote by $\mathbb{P}_t$ the marginal distribution of $\mathbb{P}$ at time $t$. The density of the Gaussian distribution with mean $\mathbf{m} \in \mathbb{R}^d$ and covariance $\Sigma \in \mathbb{R}^{d \times d}$ is denoted by $x \mapsto \mathrm{N}(x; \mathbf{m}, \Sigma)$ and $\Delta_J$ refers to the $J$-dimensional simplex, *i.e.*, $\Delta_J = \{(w_j)_{j=1}^J \in [0,1]^J : \sum_{j=1}^J w_j = 1\}$ where $J \geq 1$.

Finally, for ease of understanding, we will denote *noising* diffusion processes by $(X_t)_{t\in[0,T]}$, driven by Brownian motion $(W_t)_{t\in[0,T]}$, while $(Y_t)_{t\in[0,T]}$ will refer to *denoising* diffusion processes, driven by Brownian motion $(B_t)_{t\in[0,T]}$. In this paper, $\Theta$ will refer to as the variational parameter space.

## 2 REFERENCE-BASED DIFFUSION SAMPLING

### 2.1 THEORETICAL FRAMEWORK

We first recall the variational diffusion-based framework formulated by Richter et al. (2023) in continuous time, that generalizes the approaches from Zhang & Chen (2022) and Vargas et al. (2023a).

**Time-reversed sampling process.** Consider a general *noising* diffusion process defined on $[0, T]$ by the linear SDE

$$\mathrm{d}X_t = f(t)X_t\mathrm{d}t + \sqrt{\beta(t)}\mathrm{d}W_t \ , \ X_0 \sim \pi \ , \tag{1}$$

where the horizon $T > 0$ is fixed, $(W_t)_{t\in[0,T]}$ is a standard $d$-dimensional Brownian motion, $\beta : [0,T] \to (0,\infty)$ and $f : [0,T] \to \mathbb{R}$. We denote by $\mathbb{P}^\star$ the path measure associated to the SDE (1). With appropriate choices of $f$ and $\beta$, this SDE admits explicit transition kernels and $X_T$ is approximately, or exactly, distributed according to an easy-to-sample *base* distribution $\pi^{\mathrm{base}}$.

In the case where $\mathbb{P}_T^\star = \pi^{\mathrm{base}}$, we refer to this setting as an 'exact' noising scheme. One particular instance of this setting the Pinned Brownian Motion (PBM), considered by Tzen & Raginsky (2019); Zhang & Chen (2022); Vargas et al. (2023b), where $\pi^{\mathrm{base}} = \delta_0$. Otherwise, when we only have $\mathbb{P}_T^\star \approx \pi^{\mathrm{base}}$, we define this setting as an 'ergodic' noising scheme. A well-known example is the Variance Preserving (VP) noising process for which $\pi^{\mathrm{base}} = \mathrm{N}(0, \sigma^2\, \mathrm{I}_d)$ with $\sigma > 0$, proposed by Song et al. (2021) for score-based generative models and Vargas et al. (2023a) for sampling. Numerical experiments presented in this paper focus on these two schemes to provide fair comparison with previous methods, but the presented approach is applicable for an arbitrary noising scheme. We refer to Appendix C for more details.

Denote by $p_t^\star$ the density of $\mathbb{P}_t^\star$ w.r.t. the Lebesgue measure. Under mild assumptions on $\pi$, $f$ and $\beta$, see e.g., Cattiaux et al. (2023), the time-reversal of the noising process $\mathbb{P}^\star$, *i.e.*, the distribution of $(X_{T-t})_{t\in[0,T]}$ and denoted by $(\mathbb{P}^\star)^R$, is associated to the SDE

$$\mathrm{d}Y_t = -f(T-t)Y_t\mathrm{d}t + \beta(T-t)\nabla \log p_{T-t}^\star(Y_t)\mathrm{d}t + \sqrt{\beta(T-t)}\mathrm{d}B_t \ , \ Y_0 \sim \mathbb{P}_T^\star \ , \tag{2}$$

where $(B_t)_{t\in[0,T]}$ is another standard $d$-dimensional Brownian motion. By definition of the time-reversal, it holds that $Y_T \sim \pi$. Therefore, if we were able to simulate this diffusion process, we would obtain approximate samples from $\pi$. We therefore refer to $(Y_t)_{t\in[0,T]}$ as the *target process*. However, the scores $(\nabla \log p_t^\star)_{t\in[0,T]}$ involved in the drift function of (2) are intractable in general. In addition, these scores cannot be estimated via usual score matching techniques used in generative modeling (Hyvärinen & Dayan, 2005; Vincent, 2011) since samples from $\pi$ are not available *a priori* in the setting at hand. This point has been addressed by Richter et al. (2023) who alternatively proposed a general variational approach on path measure space, which requires the definition of a reference process as described next.

**Variational reference-based approach.** This approach relies on a *reference process* that is solution of SDE (1) with initial condition $X_0 \sim \pi^{\mathrm{ref}}$, where $\pi^{\mathrm{ref}} \in \mathscr{P}(\mathbb{R}^d)$ is chosen such that the marginal scores of the associated path measure $\mathbb{P}^{\mathrm{ref}}$ are *tractable*. Similarly to $\pi$, we assume that the probability density of $\pi^{\mathrm{ref}}$ is known up to a multiplicative constant and is given by $x \mapsto \gamma^{\mathrm{ref}}(x)/\mathcal{Z}^{\mathrm{ref}}$, where $\gamma^{\mathrm{ref}} : \mathbb{R}^d \to \mathbb{R}_+$ and $\mathcal{Z}^{\mathrm{ref}} = \int_{\mathbb{R}^d} \gamma^{\mathrm{ref}}(x)\mathrm{d}x$ is *not necessarily tractable*. Finally, for any $t \in [0,T]$, we denote the marginal scores $s_t^{\mathrm{ref}} = \nabla \log p_t^{\mathrm{ref}}$, with $p_t^{\mathrm{ref}}$ being the density of $\mathbb{P}_t^{\mathrm{ref}}$ w.r.t. the Lebesgue measure. Based on the tractable reference scores, the SDE describing the target process (2) can be rewritten as

$$\mathrm{d}Y_t = -f(T-t)Y_t\mathrm{d}t + \beta(T-t)\{s_{T-t}^{\mathrm{ref}}(Y_t) + g_{T-t}(Y_t)\}\mathrm{d}t + \sqrt{\beta(T-t)}\mathrm{d}B_t \ , \ Y_0 \sim \mathbb{P}_T^\star \ , \tag{3}$$

where $g_t = \nabla \log p_t^\star/p_t^{\mathrm{ref}}$ is now the only intractable term. This formulation of the time-reversed SDE has already been largely used in the diffusion model literature, especially for conditional generative models; see e.g., Dhariwal & Nichol (2021). In this specific context, $g_t$ is called a guidance term. From a probabilistic perspective, $g_t$ can be interpreted as a Doob-h transform control, see Appendix I.1. Exploiting the reference-based formulation given by (3), Richter et al. (2023) propose to estimate the guidance terms $(g_t)_{t\in[0,T]}$ rather than the target scores $(\nabla \log p_t^\star)_{t\in[0,T]}$.

Table 1: Connection to prior works. Here, $E^\varphi : [0,T] \times \mathbb{R}^d \to \mathbb{R}$ refers to a neural network.

| Method | Noising | $\gamma^{\text{ref}}(x)$ | $\mathcal{Z}^{\text{ref}}$ | $s_t^{\text{ref}}(x)$ | Reference parameters |
|---|---|---|---|---|---|
| **PIS** (Zhang & Chen, 2022) | PBM($\sigma$) | $\mathrm{N}(x; 0, \sigma^2 T \mathrm{I}_d)$ | 1 | $-x/\sigma^2(T-t)$ | $\sigma$ *(tuned)* |
| **DDS** (Vargas et al., 2023a) | VP($\sigma$) | $\mathrm{N}(x; 0, \sigma^2 \mathrm{I}_d)$ | 1 | $-x/\sigma^2$ | $\sigma$ *(tuned)* |
| **GMM-LRDS** (Section 3.2) | arbitrary | $\sum_{j=1}^J w_j \mathrm{N}(x; \mathbf{m}_j, \Sigma_j)$ | 1 | analytical | $\{w_j, \mathbf{m}_j, \Sigma_j\}_{j=1}^J$ (learned) |
| **EBM-LRDS** (Section 3.3) | arbitrary | $\exp(-E^\varphi(t=0, x))$ | unknown | $-\nabla_x E^\varphi(t,x)$ | $\varphi$ (learned) |

To this end, the authors consider the class of *variational* path measures $(\mathbb{P}^\theta)_{\theta \in \Theta} \subset \mathscr{P}(\mathbf{C}_T)$ where for any $\theta \in \Theta$, $\mathbb{P}^\theta$ is associated to the diffusion process

$$\mathrm{d}Y_t^\theta = -f(T-t)Y_t^\theta \mathrm{d}t + \beta(T-t)\{s_{T-t}^{\text{ref}}(Y_t^\theta) + g_{T-t}^\theta(Y_t^\theta)\}\mathrm{d}t + \sqrt{\beta(T-t)}\mathrm{d}B_t \, , Y_0^\theta \sim \pi^{\text{base}} \, , \quad (4)$$

where $g^\theta : [0,T] \times \mathbb{R}^d \to \mathbb{R}^d$ is typically a neural network. In particular, choosing $\pi^{\text{ref}}$ as a Gaussian distribution recovers exactly the previous VI methods *Path Integral Sampler* (PIS) (Zhang & Chen, 2022) and *Denoising Diffusion Sampler* (DDS) (Vargas et al., 2023a), see Table 1. Furthermore, Richter et al. (2023) propose to minimize the following LV-based objective

$$\mathcal{L}_{\text{LV}}(\theta) = \mathrm{Var}\left[\log\{(\mathrm{d}\mathbb{P}^\theta/\mathrm{d}(\mathbb{P}^\star)^R)(Y_{[0,T]}^{\hat\theta})\}\right] \, , Y_{[0,T]}^{\hat\theta} \sim \mathbb{P}^{\hat\theta} \, , \quad (5)$$

where $\hat\theta$ is a detached version of $\theta$, meaning that gradients with respect to $\theta$ will not see $\hat\theta$. In contrast to the reverse KL divergence (previously used in PIS and DDS), which is notably known to suffer from mode collapse (Midgley et al., 2023), the LV divergence (Nüsken & Richter, 2021) has the benefits to improve mode exploration, to avoid costly computations of the loss gradients and to have zero variance at the optimal solution (Richter et al., 2023, Section 2.3). Finally, the LV loss (5) can be further explicited, as detailed in the following proposition.

**Proposition 1.** *Assume that $\mathbb{P}_T^\star = \mathbb{P}_T^{ref} = \pi^{base}$ and there exists $\theta^\star \in \Theta$ such that $g_t^{\theta^\star} = g_t$. Then, the loss defined in (5) achieves optimal solution at $\theta^\star$ and, setting $\varrho = \log(\gamma^{\text{ref}}/\gamma)$, it simplifies as*

$$\mathcal{L}_{\text{LV}}(\theta) = \mathrm{Var}\left[\int_0^T \frac{\beta(T-t)}{2}\left\|g_{T-t}^\theta(Y_t^{\hat\theta})\right\|^2 \mathrm{d}t + \int_0^T \sqrt{\beta(T-t)}g_{T-t}^\theta(Y_t^{\hat\theta})^\top \mathrm{d}B_t + \varrho(Y_T^{\hat\theta})\right] \, . \quad (6)$$

This result is an adaptation of (Richter et al., 2023, Lemma 3.1), which proof is restated in Appendix D.1 for completeness. We also emphasize that the assumption made in Proposition 1 is only needed for ergodic noising schemes. In practice, Richter et al. (2023) only consider two specific versions of this continuous-time variational loss, where $\pi^{\text{ref}}$ is determined by the PIS and DDS settings. In the following, we will refer to these extensions of PIS and DDS as LV-PIS and LV-DDS. In contrast, we keep a general perspective, by letting $\pi^{\text{ref}}$ completely arbitrary, and describe our proposition of practical implementation of a Reference-based Diffusion Sampler (RDS) in this context.

## 2.2 REFERENCE-BASED DIFFUSION SAMPLING IN PRACTICE

**Discrete-time setting.** In practice, the parameterized process $(Y_t^{\hat\theta})_{t \in [0,T]}$ cannot be simulated exactly and SDE (4) can only be numerically solved with a small step-size using, for example, the Euler-Maruyama (EM) or Exponential Integration (EI) (Durmus & Moulines, 2015) discretization schemes, see Appendix B for additional details. Consider a time discretization of the interval $[0,T]$ given by an increasing sequence of timesteps $\{t_k\}_{k=0}^K$ such that $t_0 = 0$, $t_K = T$ and $K \geq 1$. Given an *arbitrary* reference distribution $\pi^{\text{ref}}$, we propose to approximate the continuous-time objective defined in (6) by the following *discrete time* RDS objective

$$\mathcal{L}_{\text{RDS}}(\theta) = \mathrm{Var}\left[\sum_{k=0}^{K-1} w_k g_{T-t_k}^\theta(Y_k)^\top \left\{s_{T-t_k}^{\hat\theta}(Y_k) - \frac{1}{2}g_{T-t_k}^\theta(Y_k)\right\} + \sum_{k=0}^{K-1} \sqrt{w_k} g_{T-t_k}^\theta(Y_k)^\top Z_k + \varrho(Y_K)\right] \, , \tag{7}$$

where $\{Z_k\}_{k=0}^{K-1}$ are independently distributed according to $\mathrm{N}(0, \mathrm{I}_d)$, $\{Y_k\}_{k=0}^K$ is recursively defined by $Y_0 \sim \pi^{\text{base}}$ and for any $k \in \{0, \ldots, K-1\}$,

$$Y_{k+1} = a_k Y_k + b_k \{s_{T-t_k}^{\text{ref}}(Y_k) + g_{T-t_k}^{\hat\theta}(Y_k)\} + \sqrt{c_k} Z_k \, , \quad (8)$$

with $\hat{\theta}$ being a detached version of $\theta$ and $\{w_k, a_k, b_k, c_k\}_{k=0}^{K-1}$ being *tractable* coefficients that depend on the choice of discretization and noising schemes. In particular, if $K$ is sufficiently large, $\{Y_k\}_{k=0}^K$ has approximately the same distribution as $\{Y_{t_k}^{\hat{\theta}}\}_{k=0}^K$. We refer to Appendix D.2 for a detailed explanation on the computation of our loss and explicit values of the coefficients. We emphasize that this variational objective can be implemented with any reference distribution. In comparison, the practical loss functions[1] exhibited by Richter et al. (2023) are only designed for LV-PIS and LV-DDS, and may be seen as particular instances of (7).

**Training and sampling with RDS.**   The training stage of RDS consists in running $N$ iterations of Stochastic Gradient Descent (SGD) to minimize the discrete-time objective (7), for some $N \geq 1$. Like other variational diffusion-based objectives, the RDS loss is *simulation-based*, meaning that, the whole process $\{Y_k\}_{k=0}^K$ defined in (8) must be simulated at each SGD step. The complete training procedure of RDS is summarized in Algorithm 2, stated in Appendix A. After training, the final parameter $\theta^\star$ is fixed, and approximate samples from $\pi$ are obtained by simulating $\{Y_k\}_{k=0}^K$ from $g^{\theta^\star}$, using (8), and taking $Y_K$.

# 3   LEARNED RDS FOR MULTI-MODAL DISTRIBUTIONS

In this section, we highlight the crucial role played by the reference distribution $\pi^{\mathrm{ref}}$ in RDS when targeting multi-modal distributions. We first show in Section 3.1 the limitations of previous variational reference-based methods and give intuition on how $\pi^{\mathrm{ref}}$ should be chosen. Based on this observation, we introduce *Learned Reference-based Diffusion Sampler* (LRDS), a novel diffusion-based sampler defined as a practical instance of RDS, where $\pi^{\mathrm{ref}}$ is learned from prior knowledge on $\pi$, namely the location of their modes. We present two approaches based on Gaussian Mixture Models (GMM-LRDS), see Section 3.2, and Energy-Based Models (EBM-LRDS), see Section 3.3.

## 3.1   ON THE IMPORTANCE OF THE REFERENCE DISTRIBUTION

Choosing a Gaussian distribution for $\pi^{\mathrm{ref}}$ ensures that the entire reference process is Gaussian. Therefore, the scores $(s_t^{\mathrm{ref}})_{t \in [0,T]}$ can be analytically computed. We will refer to this case as *Gaussian RDS* (G-RDS). In this setting, previous variational reference-based methods (PIS, DDS and their log-variance version) focused on using isotropic centered Gaussian distributions as reference distributions (see Table 1), *i.e.*, $\pi^{\mathrm{ref}} = \mathrm{N}(0, \sigma^2 \mathrm{I}_d)$ where the hyperparameter is $\sigma \in (0, \infty)$. This parameter is fixed before running the optimization over $\theta$ and a priori disconnected from the target distribution. To assess the sensitivity of the algorithms with respect to the hyperparameter $\sigma$, we run LV-DDS and LV-PIS on a simple multi-modal distribution – a 16-dimensional bi-modal Gaussian mixture with two distinct weights – and examine the accuracy of estimation of these weights (see Figure 1, left). We observe that the performance is highly dependent on the value of $\sigma$, with an optimal accuracy achieved when the isotropic variance is directly related to statistics of the target distribution; see Appendix I.2 for details. However, it is hard to estimate this quantity without access to ground truth samples.

**Towards a multi-modal reference distribution.**   The previous experiment suggests that choosing a reference distribution that is close to the target distribution improves the performance of the VI sampler. To validate this intuition, we target the same Gaussian mixture as above using RDS with PBM and VP noising schemes, setting $\pi^{\mathrm{ref}}$ as a Gaussian mixture with the same components as the target with their *relative weights* as hyperparameter (see Figure 1, right). We find that this design of $\pi^{\mathrm{ref}}$, with a well adjusted Gaussian mixture reference, enables RDS to robustly recover the information of the relative weight of the target modes. These observations motivate us to adopt the following paradigm for RDS: given a multi-modal target distribution $\pi$, we aim to design $\pi^{\mathrm{ref}}$ close to $\pi$, sharing *the same multi-modality characteristics*. Below, we present *Learned Reference-based Diffusion Sampler* (LRDS), a complete sampling methodology relying on the RDS framework that leverages prior knowledge on the target modes to learn a well-suited reference distribution.

---

[1]Although the authors do not provide any expression of a discrete-time version of (6) in their paper, a discrete-time loss is implemented for LV-PIS and LV-DDS in their official codebase available at `https://github.com/juliusberner/sde_sampler`.

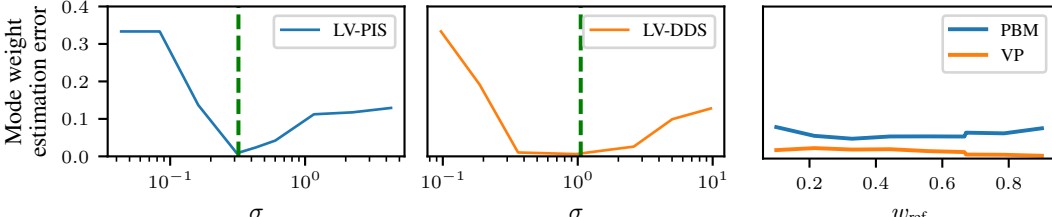

Figure 1: **Illustration of the decisive role of the reference distribution**. Here, we target a 16-dimensional Gaussian mixture with two modes, that have respective weights $w = 2/3$ and $1 - w = 1/3$, and display the estimation error of $w$ with different methods. **(Left)**: Results for LV-PIS and LV-DDS when varying the value of their hyperparameter $\sigma$ (which directly determines $\pi^{\text{ref}}$ as shown in Table 1). The green dotted line represents the optimal variance for Gaussian approximation of $\pi$, see Appendix I.2 for related computations. **(Right)** Results for RDS in PBM and VP settings when setting $\pi^{\text{ref}}$ as a Gaussian mixture with the same modes as $\pi$, but $w$ is replaced by $w_{\text{ref}}$. Details on the design of this experiment are given in Appendix H.1.

**LRDS pipeline.**   Our algorithm LRDS proceeds in three main steps:

(a) **Obtain *reference* samples**.   Given a standard MCMC sampler which targets $\pi$, such as the *Metropolis-Adjusted Langevin Algorithm* (MALA) (Roberts & Tweedie, 1996), we simulate multiple Markov chains that are initialized in the target mode locations. We refer to the obtained samples as the *reference samples* and denote by $\hat{\pi}^{\text{ref}}$ the corresponding empirical distribution. As recalled in Appendix I.3, in presence of high energy barriers between the modes, $\hat{\pi}^{\text{ref}}$ fails at estimating the global energy landscape. Moreover, $\hat{\pi}^{\text{ref}}$ does not admit a density w.r.t. the Lebesgue measure.

(b) **Define the reference process.**   In this stage, we aim at (i) learning $\pi^{\text{ref}}$ as a continuous approximation of $\hat{\pi}^{\text{ref}}$, and (ii) computing the marginal scores at times $\{T - t_k\}_{k=0}^{K-1}$ of the induced reference process $\mathbb{P}^{\text{ref}}$, *i.e.*, the noising process defined by SDE (1), where $X_0 \sim \pi^{\text{ref}}$. In practice, we set $\pi^{\text{ref}}$ either as a Gaussian Mixture Model, which relies on a light parameterization, or as an Energy-Based Model, which is more expressive at the cost of higher computational budget.

(c) **Run the RDS variational optimization.**   Once the reference distribution is learned and the reference scores are computed, we run the RDS procedure described in Section 2.2.

In the next sections, we describe two practical implementations of stage (b) in LRDS using different parameterizations of the reference distribution.

## 3.2   GAUSSIAN MIXTURE MODEL LRDS

We first propose to set $\pi^{\text{ref}}$ as a *Gaussian mixture* to integrate multi-modality in RDS. In particular, we have $\mathcal{Z}^{\text{ref}} = 1$ and any reference density is parameterized as $\gamma^{\text{ref}}(\cdot) = \sum_{j=1}^{J} w_j \mathrm{N}(\cdot ; \mathbf{m}_j, \Sigma_j)$ where $J \geq 1$ is the number of reference modes, $\{w_j\}_{j=1}^{J} \in \Delta_J$ are the reference weights, $\mathbf{m}_j \in \mathbb{R}^d$ and $\Sigma_j \in \mathbb{R}^{d \times d}$ are respectively the mean and the covariance of the $j$-th reference mode. In this setting, each marginal of $\mathbb{P}^{\text{ref}}$ is also a Gaussian mixture whose parameters can be simply deduced from $\gamma^{\text{ref}}$, making their score fully tractable, see Appendix B. Therefore, the only difficulty at stage (b) lies in the estimation of the reference parameters $\varphi = \{w_j, \mathbf{m}_j, \Sigma_j\}_{j=1}^{J}$ for some $J \geq 1$ fixed in advance. If $J > 1$, we refer to this version of the algorithm as *Gaussian Mixture Model LRDS* (GMM-LRDS). In this case, we learn $\varphi$ by running the Expectation Minimization (EM) algorithm (Dempster et al., 1977) with samples from $\hat{\pi}^{\text{ref}}$. If $J = 1$, LRDS reduces to *Gaussian LRDS* (G-LRDS), and we learn parameters $(\mathbf{m}_1, \Sigma_1)$ by Maximum Likelihood (ML) estimation. In practice, we observe that setting $J$ equal or larger to the number of target modes can lead to better performance. We summarize the whole GMM-LRDS methodology in Algorithm 3, provided in Appendix A, and illustrate in Figure 2 the behaviours of GMM-LRDS and G-LRDS for a bi-modal target distribution. Our results show that choosing a Gaussian reference distribution leads to mode collapse, while defining $\pi^{\text{ref}}$ as in GMM-LDRS enables RDS to recover the relative weights accurately.

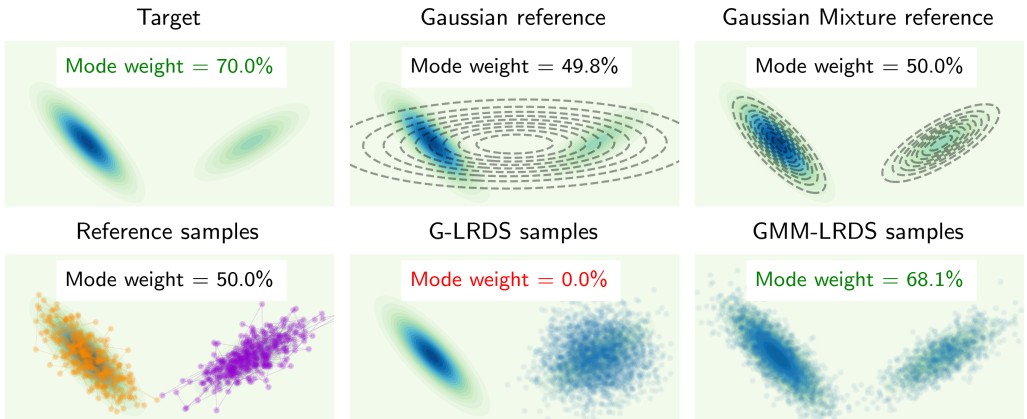

Figure 2: **Comparison between G-LRDS and GMM-LRDS.** Here, the target distribution is the same 16-dimensional Gaussian mixture as in Figure 1 (top left), see Appendix H.1 for more details. For illustration purpose, projections along the first two coordinates are used. In each cell, the value of 'Mode weight' refers to the effective weight of the left mode. Reference samples (bottom left) are obtained by running MALA sampler initialized in both target modes: each color (orange/purple) depicts one MALA Markov chain. In particular, none of them mixes between the modes. Running G-LRDS with an ML-estimated Gaussian reference (top middle) leads to mode collapse (bottom middle). Conversely, GMM-LRDS with an EM-estimated Gaussian mixture reference (top right) appropriately recovers the target distribution and the true mode proportions (bottom right).

### 3.3 ENERGY-BASED MODEL LRDS

Although GMM-LRDS efficiently introduces multi-modality in the reference distribution while conserving the tractability of the reference scores, we anticipate that in some cases, a Gaussian mixture may not provide a correct approximation of $\hat{\pi}^{\mathrm{ref}}$, regardless of the number of reference modes $J$, see Appendix I.4 for illustrations. To provide more flexibility, we propose a second version of LRDS where the reference distribution is parameterized by an *Energy-Based Model* (EBM), *i.e.*, $\gamma^{\mathrm{ref}}(x) = \exp(-E^{\varphi}(x))$ where $E^{\varphi} : \mathbb{R}^d \to \mathbb{R}$ is a neural network with parameters $\varphi$ and the normalizing constant $\mathcal{Z}^{\mathrm{ref}} = \int_{\mathbb{R}^d} \exp(-E^{\varphi}(x))\mathrm{d}x$ is *intractable*. In contrast to GMM-LRDS, the corresponding reference scores cannot be computed analytically anymore; hence, we suggest to *estimate* them with neural networks. In the following, we will denote $\pi^{\mathrm{ref}}$ by $p^{\varphi}$ to insist on its parametric nature.

At first sight, it seems natural to learn the density $\gamma^{\mathrm{ref}}$ and the scores $(s_t^{\mathrm{ref}})_{t \in [0,T]}$ independently from each other. Indeed, following the diffusion model literature (Song et al., 2021; Karras et al., 2022), the reference scores could be easily learned using a *Score Matching* (SM) objective with samples from $\hat{\pi}^{\mathrm{ref}}$ (Hyvärinen & Dayan, 2005; Vincent, 2011). On the other hand, one could learn $p^{\varphi}$ by maximizing the standard *Maximum Likelihood* (ML) objective $\mathcal{L}^{\mathrm{ML}} : \varphi \mapsto \mathbb{E}\left[\log p^{\varphi}(X)\right]$, where $X \sim \hat{\pi}^{\mathrm{ref}}$, with corresponding gradient given by $\varphi \mapsto \mathbb{E}[\nabla_{\varphi} E^{\varphi}(X^-)] - \mathbb{E}[\nabla_{\varphi} E^{\varphi}(X^+)]$, where $X^- \sim p^{\varphi}$ and $X^+ \sim \hat{\pi}^{\mathrm{ref}}$. Nevertheless, computing gradients of objective $\mathcal{L}^{\mathrm{ML}}$ requires to sample from $p^{\varphi}$ (*negative sampling*), which is a well known hurdle in EBM training since $p^{\varphi}$ is expected to be as multi-modal as $\hat{\pi}^{\mathrm{ref}}$, and thus $\pi$.

Therefore, we rather suggest to learn a path of parametric distributions $(p_t^{\varphi})_{t \in [0,T]}$ as the marginal distributions of $(\hat{X}_t^{\mathrm{ref}})_{t \in [0,T]}$, defined as the noising process induced by SDE (1), starting at $\hat{X}_0^{\mathrm{ref}} \sim \hat{\pi}^{\mathrm{ref}}$. To do so, we set $(p_t^{\varphi})_{t \in [0,T]}$ as a *multi-level* EBM, *i.e.*, $p_t^{\varphi}(x) = \exp(-E^{\varphi}(t,x))/\mathcal{Z}_t^{\varphi}$, where $E^{\varphi} : [0,T] \times \mathbb{R}^d \to \mathbb{R}$ is now a *time-dependent* neural network with parameters $\varphi$ and the normalizing constants $\mathcal{Z}_t^{\varphi} = \int_{\mathbb{R}^d} \exp(-E^{\varphi}(t,x))\mathrm{d}x$ are still intractable. In the RDS framework, this amounts to consider $\gamma^{\mathrm{ref}}(x) = \exp(-E^{\varphi}(0,x))$ and $s_t^{\mathrm{ref}}(x) = -\nabla_x E^{\varphi}(t,x)$. To learn this multi-level EBM, we propose to maximize the integrated ML objective $\varphi \mapsto \int_0^T \mathbb{E}[\log p_t^{\varphi}(\hat{X}_t^{\mathrm{ref}})]\mathrm{d}t$. In this case, we can leverage the correlations between the single-level EBMs to alleviate their individual issue of negative sampling, a strategy that has already been investigated in several works (Gao et al., 2021; Zhu et al., 2024; Zhang et al., 2023).

Here, since $(\hat{X}_t^{\mathrm{ref}})_{t\in[0,T]}$ defines a path of increasingly simpler distributions, *annealed MCMC samplers* can be conveniently used to sample from the multi-level EBM densities. This allows us to kill two birds in one stone: (i) we obtain negative samples for each single-level EBM, which is needed to compute the gradient of the ML objective, and (ii) we overcome the individual sampling issues at every level thanks to annealing. This method is completely detailed in Algorithm 11 of Appendix F together with previous literature on multi-level EBMs. Morever, Appendix E provides details on annealed MCMC samplers.

This version of LRDS called *Energy-Based Model LRDS* (EBM-LRDS) is summarized in Algorithm 4 of Appendix A. In Figure 3, we illustrate the superiority of EBM-LRDS over GMM-LRDS for a target distribution $\pi$ that exhibits complex geometry. While the GMM fails at capturing the local energy landscape of $\pi$, which results in a poor performance of GMM-LRDS, EBM-LRDS captures well the target distribution since the reference EBM recovers the complex geometry of the target.

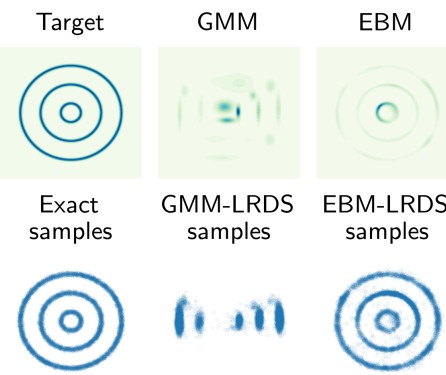

Figure 3: **Comparison between GMM-LRDS and EBM-LRDS in a multi-modal setting**. Here, we target the 2-dimensional Rings distribution, which has 3 *unbalanced* modes represented by the rings. (**Left**): Target density (top) and exact samples (bottom). (**Middle**): 16-component GMM reference distribution (top) and resulting GMM-LRDS samples (bottom). (**Right**): EBM reference distribution (top) and resulting EBM-LRDS samples (bottom).

## 4 RELATED WORKS

In the past years, numerous parametric methods have been proposed to approximate a diffusion process $(Y_t)_{t\in[0,T]} \sim \mathbb{P}$ induced by an SDE with intractable drift, such as (2), bridging an easy-to-sample distribution $\pi^{\mathrm{base}}$ to a target distribution $\pi$. Based on deep learning techniques, they suggest to learn a path measure $\mathbb{P}^\theta$, induced by a *neural* SDE that admits a neural network (parameterized by $\theta \in \Theta$ to be optimized) as drift function. Below, we review two main classes of such algorithms.

**Variational diffusion-based methods.** To optimize $\theta$, several approaches have adopted a variational formulation on path measure space, which consists in minimizing a divergence-based loss with samples from $\mathbb{P}^\theta$. We adopt the same perspective in the present work. For instance, a recent line of works has defined $\mathbb{P}$ as the time-reversal of a noising diffusion process, such as PIS (Zhang & Chen, 2022), DDS (Vargas et al., 2023a) and *Time-reversed Diffusion Sampler* (DIS) (Berner et al., 2023). In these methods, the neural network used in $\mathbb{P}^\theta$ is crucially required to be *target-informed*, *i.e.*, it is parameterized with the score of the target distribution. For a large number of variational training steps, this may result in a costly procedure, since each training step requires a full simulation of $\mathbb{P}^\theta$. While those algorithms were originally implemented with the largely used reverse KL divergence, Richter et al. (2023) demonstrate the benefits of using the Log-Variance (LV) divergence to avoid mode collapse in the variational optimization stage. However, even combined with a LV-based loss, these variational methods still require a target-informed parameterization. On the other hand, Vargas et al. (2024) propose a different perspective by defining $\mathbb{P}$ as a controlled version of an *annealed Langevin* diffusion, that is expected to follow a prescribed path of tractable marginal densities. Here, both $\mathbb{P}$ and $\mathbb{P}^\theta$ are induced by neural SDEs. The authors present two versions of their algorithm, *Controlled Monte Carlo Diffusion* (CMCD), using either reverse KL or LV divergence. For clear comparison with the RDS discrete-time setting presented in Section 2.2, we describe all of these approaches (PIS, DDS, DIS, CMCD) under the discrete time scope in Appendix D.2.

**Adaptive diffusion-based approaches.** To alleviate the computational difficulties of divergence-based losses, recent works have proposed to learn $\mathbb{P}^\theta$ with an adaptive procedure (Phillips et al., 2024; Akhound-Sadegh et al., 2024). These methods iterate two steps which consist of (a) obtaining approximate target samples by sampling from $\mathbb{P}^\theta$ and (b) optimizing $\theta$ using those samples via learning techniques usually restricted to generative modeling. In particular, Akhound-Sadegh et al. (2024) present *Iterated Denoising Energy Matching* (iDEM), where $\mathbb{P}$ corresponds to the time-reversal of a Variance-Exploding noising diffusion process (Song et al., 2021). In their setting, stage (a) is conducted by running the SDE induced by $\mathbb{P}^\theta$ and stage (b) relies on a novel energy-matching loss which directly depends on $\pi$. On the other hand, the *Particle Denoising Diffusion Sampler*

(PDDS) (Phillips et al., 2024) (a) introduces a SMC-based scheme when sampling from $\mathbb{P}^\theta$, see Appendix E for more details, and (b) implements an extension of the Target Score Matching loss (De Bortoli et al., 2024). When it comes to practice, we find that both of these algorithms face significant limitations: while PDDS performance is highly sensitive to the choice of the initial $\mathbb{P}^\theta$, the iDEM energy-matching loss suffers from very high variance.

## 5 NUMERICAL EVALUATION OF RDS

To validate our approach, we compare GMM-LRDS and EBM-LRDS on a variety of multi-modal distributions against the following annealed methods: (a) **annealed MCMC methods** – *Sequential Monte Carlo* (SMC) (Del Moral et al., 2006) and *Replica Exchange* (RE) (Swendsen & Wang, 1986); (b) **variational diffusion-based methods**, implemented with the LV loss – LV-PIS (Zhang & Chen, 2022), LV-DDS (Vargas et al., 2023a), LV-DIS (Berner et al., 2023) and LV-CMCD (Vargas et al., 2024); (c) **adaptive diffusion-based approaches** – iDEM (Akhound-Sadegh et al., 2024) and PDDS (Phillips et al., 2024). To assess the performance of each sampler in multi-modal settings, we will evaluate how well the obtained samples are able to recover the weights of the target modes[2]. The details of each target distribution can be found in Appendix H.1.

**General experimental setting.** To ensure fair comparison with previous approaches, we make sure that all competing methods are as informed as LRDS of prior knowledge on the target distribution. More specifically: we set $\pi^{\text{base}}$ as a Gaussian approximation of $\hat{\pi}^{\text{ref}}$ in SMC, RE or CMCD; we choose $\sigma$ based on a Gaussian isotropic approximation of $\hat{\pi}^{\text{ref}}$ for PIS, DDS or DIS (see Section 3.1); we standardize the target distributions using the empirical mean and variances of $\hat{\pi}^{\text{ref}}$ for iDEM and PDDS. Additionally, we pre-fill iDEM's training buffer with samples from $\hat{\pi}^{\text{ref}}$. Note that all competing methods use the score of the target distribution, either in their training procedure when computing the loss (iDEM, PDDS) or in the sampling procedure : through a target-informed neural network parameterization in PIS, DDS and DIS; through the target-informed base drift in CMCD; through the MCMC gradient steps in SMC or RE. In contrast, LRDS only requires evaluations of the target density, which makes it an interesting alternative in settings where the score of $\pi$ is expensive to compute. We refer to Appendix H.3 for complete details of the implementation of these methods.

**High-dimensional Gaussian mixtures.** We first consider a synthetic but challenging setting, where $\pi$ is a bi-modal Gaussian mixture whose modes are $\mathrm{N}(-\mathbf{1}_d, \Sigma_1)$, with weight $w_1 = 2/3$, and $\mathrm{N}(\mathbf{1}_d, \Sigma_2)$, with weight $w_2 = 1/3$, where $\mathbf{1}_d$ is the $d$-dimensional vector with all components equal to 1, and for $i \in \{1, 2\}$, $\Sigma_i \in \mathbb{R}^{d \times d}$ is a diagonal positive matrix with conditioning number equal to 100. For each method, the ground truth weight $w_1$ is estimated by a Monte Carlo estimator $\hat{w}_1$. We report the estimation error $|w_1 - \hat{w}_1|$ for increasing values of $d$ in Table 2. Note that mode collapse occurs when $\hat{w}_1 \in \{0, 1\}$. We observe that GMM-LRDS outperforms competing methods in all the considered dimensions. In Appendix I.5.1, we complete these mode weight results with probability metrics (see Appendix B), present further experiments with bi-modal Gaussian mixtures with lower condition numbers and dimension, which show that competing algorithms are able to perform on par with LRDS in these simpler settings, and conduct ablation studies on this two-mode type of target (by lightly perturbing the reference distribution or increasing the distance between the modes). To further assess the performance and robustness of LRDS, we also consider the extension to more than two modes in Appendix I.5.2, where LRDS still outperforms its competitors.

**Field system $\phi^4$ from statistical mechanics.** Next, we sample from the 1D $\phi^4$ model, previously studied (Gabrié et al., 2022; Grenioux et al., 2024). At the chosen temperature, the distribution has two well distinct modes with respective weights $w_-$ and $w_+$ such that the relative weight $w_-/w_+$ can be adjusted through a 'local-field' parameter $h$. We discretize this continuous model with a grid size of 32 (*i.e.*, $d = 32$). For each method, we compute a Monte Carlo estimation of $w_-/w_+$ and compare the results with a Laplace approximation (0-th and 2-nd orders), see Appendix H.1 for the computations. In this setting, all competing approaches suffer from mode collapse while GMM-LRDS is close to the ground truth as shown by Figure 4. Complete results showing the failure of the competing methods are provided in Appendix I.5.4. Given the satisfying results obtained with the lightweight GMM-LRDS sampler, we did not run EBM-LRDS for this target distribution.

---

[2]In Appendix G, we give expressions of additional variational metrics proposed by Blessing et al. (2024) to quantify mode collapse in the specific case of RDS.

Table 2: **Absolute mode weight estimation error** for a bimodal Gaussian mixture with growing $d$, averaged over 16 sampling runs. Bold font indicates best result, orange cells refer to settings with uninformative mode weight estimation (i.e., uniform mixture), red cells denote mode collapse. N/A denotes settings with numerical issues.

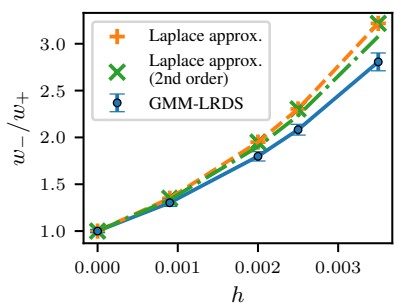

| Algorithm | $d = 16 \downarrow$ | $d = 32 \downarrow$ | $d = 64 \downarrow$ |
|---|---|---|---|
| SMC | $11.4\%_{\pm 9.1\%}$ | $15.8\%_{\pm 8.5\%}$ | $15.2\%_{\pm 7.5\%}$ |
| RE | $16.5\%_{\pm 1.3\%}$ | $15.9\%_{\pm 1.4\%}$ | $17.0\%_{\pm 1.4\%}$ |
| LV-PIS | $6.0\%_{\pm 3.4\%}$ | $33.2\%_{\pm 0.1\%}$ | $33.0\%_{\pm 0.1\%}$ |
| LV-DDS | $11.8\%_{\pm 9.3\%}$ | $31.5\%_{\pm 2.9\%}$ | $33.1\%_{\pm 0.1\%}$ |
| LV-DIS | $16.4\%_{\pm 0.5\%}$ | $16.5\%_{\pm 0.4\%}$ | $16.8\%_{\pm 0.6\%}$ |
| LV-CMCD | $32.2\%_{\pm 15.4\%}$ | $50.1\%_{\pm 8.8\%}$ | $16.3\%_{\pm 10.6\%}$ |
| iDEM | $33.3\%_{\pm 0.0\%}$ | $66.7\%_{\pm 0.0\%}$ | $11.7\%_{\pm 0.4\%}$ |
| PDDS | $\mathbf{0.8\%_{\pm 0.6\%}}$ | $66.7\%_{\pm 0.0\%}$ | N/A |
| GMM-LRDS | $1.7\%_{\pm 0.6\%}$ | $\mathbf{2.7\%_{\pm 0.8\%}}$ | $\mathbf{4.1\%_{\pm 0.6\%}}$ |

Figure 4: **Estimation of the relative weight** of $\phi^4$ modes with increasing $h$, averaged over 16 sampling runs.

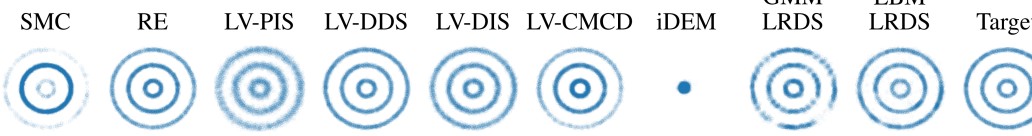

Figure 5: **Samples obtained for Rings distribution**. Reasonable results could not be obtained with PDDS due to numerical issues.

**Compactly supported multi-modal distributions.** Then, we aim to sample from 2-dimensional multi-modal distributions with complex geometries. We consider (a) *Rings distribution*, which has 3 ring-shaped modes, and (b) *Checkerboard distribution*, which has 8 square-shaped modes. In both cases, the modes are not evenly weighted. In this setting, we consider $J = 64$ components for GMM-LRDS and leverage the Replica Exchange algorithm as backbone annealed MCMC sampler in the EBM-LRDS training algorithm. Apart from adaptive methods, the structure of Rings modes are well recovered by all approaches, see Figure 5. However, we observe that non diffusion-based approaches fail to recover the ground truth mode weights. On the other hand, LRDS is the only sampling method to obtain samples that are close to exact for the Checkerboard distribution, see Figure 6, while being able to correctly estimate the ground truth mode weights.

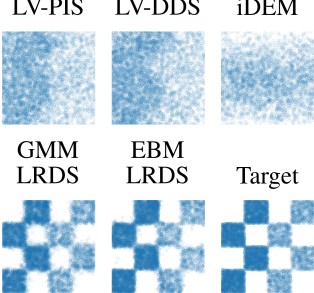

Figure 6: **Samples obtained for Checkerboard distribution.** All sampling methods except LRDS fail to provide accurate samples.

**Bayesian logistic regression models.** Lastly, we provide in Appendix I.5.5 sampling results on standard Bayesian posterior distributions, which are not however explicitly multi-modal.

## 6 DISCUSSION

Building on the class of recently developed annealed VI methods, this paper presents the *Learned Reference-based Diffusion Sampler* (LRDS), which specifically addresses the challenging case of multi-modal target distributions. In essence, LRDS aims at learning a reference process – based on GMMs or multi-level EBMs – adapted to the target distribution by using samples obtained via local MCMC samplers initialized in the modes. Our numerical experiments show that GMM-LRDS accurately recovers the global information of the relative weights of the modes in several tens of dimensions, unlike competing methods, and that EBM-LRDS can help tackling distributions with non-Gaussian properties such as sharp supports. However this advantage of LRDS comes at the computational cost of the necessary pre-training of the reference process model. Concerning future work, we note in particular that EBM-LRDS is a promising tool for real-world sampling tasks on non-euclidean spaces which may benefit from the flexible definition of the reference process as an EBM. An interesting test case of this kind that shall be considered is the sampling of the Boltzmann distribution of proteins in internal coordinates.

ACKNOWLEDGEMENTS

We would like to thank Lorenz Richter, Julius Berner, Denis Blessing, Junhua Chen, Francisco Vargas and Yazid Janati for many stimulating discussions. LG and MG acknowledge funding from Hi! Paris. AD and MN would like to thank the Isaac Newton Institute for Mathematical Sciences and the Alan Turing Institute for support and hospitality during the programme *Diffusions in machine learning: Foundations, generative models and non-convex optimisation* when work on this paper was undertaken. Part of the work of AD is funded by the European Union (ERC, Ocean, 101071601). Views and opinions expressed here are however those of the authors only and do not necessarily reflect those of the European Union or the European Research Council Executive Agency. Neither the European Union nor the granting authority can be held responsible for them. This work was partially performed using HPC resources from GENCI–IDRIS (AD011015234).

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

## Organization of the supplementary

The appendix is organized as follows. In Appendix A, we provide the pseudo-codes of every RDS instance presented in the main part of this paper (standard RDS, GMM-LRDS & EBM-LRDS). Appendix B summarizes general facts that will be useful for proofs and corresponding computations. In Appendix C, we describe the noising diffusion schemes considered in this work, namely the Pinned Brownian Motion and the Variance-Preserving setting, and propose novel schemes. For RDS and related variational diffusion-based methods, we dispense details on their continuous-time and corresponding discrete-time variational objectives in Appendix D. We respectively provide an overview of existing annealed MCMC methods and multi-level Energy-Based models in Appendix E and Appendix F. In Appendix G, we provide expressions of additional variational metrics proposed by Blessing et al. (2024) in the RDS setting. Details on our experimental settings, our implementation of RDS and other competing methods are given in Appendix H. Finally, we present additional theoretical and experimental results in Appendix I.

Our codebase is available at `https://github.com/h2o64/sde_sampler_lrds`.

**Notation.** Let $\mathbb{P} \in \mathscr{P}(\mathbf{C}_T)$ and $t \in [0, T]$. For any $x_t \in \mathbb{R}^d$, we denote by $\mathbb{P}^{x_t}_{|t} \in \mathscr{P}(\mathbf{C}_T)$ the path measure associated to the stochastic process $(X_t)_{t \in [0,T]} \sim \mathbb{P}$ conditioned on $X_t = x_t$. Furthermore, for any $\pi \in \mathscr{P}(\mathbb{R}^d)$, $\pi \otimes \mathbb{P}^{x_t}_{|t} \in \mathbf{C}_T$ stands for the path measure $\int_{\mathbb{R}^d} \mathbb{P}^{x_t}_{|t} \mathrm{d}\pi(x_t)$. For any $K \geq 1$, we denote the set of joint distributions $\mathscr{P}((\mathbb{R}^d)^{K+1})$ by $\mathscr{P}^{(K+1)}$. Finally, we adopt the following notation to design sample batches: for any $\bar{L} \geq \underline{L} \geq 0$ and $\bar{K} \geq \underline{K} \geq 0$, we denote by $X^{\underline{L}:\bar{L}}_{\underline{K}:\bar{K}}$ the sequence of samples $\{X^\ell_k\}^{\bar{K},\bar{L}}_{k=\underline{K},\ell=\underline{L}}$.

## A   Pseudo-codes of RDS-based algorithms

We respectively give sampling procedures and training procedures of a general version of RDS in Algorithm 1 and Algorithm 2. Relying on this, we derive the complete training schemes for GMM-LRDS (Algorithm 3) and EBM-LRDS (Algorithm 4).

---

**Algorithm 1:** Reference-based Diffusion Sampling (RDS) : sampling stage

**Input:** Time discretization $\{t_k\}^K_{k=0}$ of $[0, T]$, reference scores $\{s^{\mathrm{ref}}_{T-t_k}\}^{K-1}_{k=0}$, neural network
$\quad g^\theta : [0, T] \times \mathbb{R}^d \to \mathbb{R}^d$, variational coefficients $\{a_k, b_k, c_k\}^{K-1}_{k=0}$ defined in Appendix D.2
▷ Initialization
$Y_0 \sim \pi^{\mathrm{base}}$
$(Z_k)^{K-1}_{k=0} \overset{\mathrm{i.i.d.}}{\sim} \mathrm{N}(0, \mathrm{I}_d)$
**for** $k = 0, \ldots, K-1$ **do**
$\quad$ ▷ Compute the $k$-th diffusion step
$\quad Y_{k+1} = a_k Y_k + b_k \{s^{\mathrm{ref}}_{T-t_k}(Y_k) + g^{\hat\theta}_{T-t_k}(Y_k)\} + \sqrt{c_k} Z_k$
**Output:** Discrete time process $Y_{0:K}$ approximating $(Y^{\hat\theta}_{t_k})^K_{k=0}$

---

**Algorithm 2:** Reference-based Diffusion Sampling (RDS) : training stage

**Input:** Time discretization $\{t_k\}^K_{k=0}$ of $[0, T]$, target density $\gamma$, reference density $\gamma^{\mathrm{ref}}$ and scores
$\quad \{s^{\mathrm{ref}}_{T-t_k}\}^{K-1}_{k=0}$, neural network $g^\theta : [0, T] \times \mathbb{R}^d \to \mathbb{R}^d$ initialized such that $g^{\theta_0} = 0$, number of
$\quad$ training iterations $N$, batch size $B$, variational coefficients $\{w_k, a_k, b_k, c_k\}^{K-1}_{k=0}$ defined in
$\quad$ Appendix D.2
**for** $n = 0, \ldots, N-1$ **do**
$\quad$ ▷ Simulate $B$ trajectories of the process $(Y^{\hat\theta}_t)_{t \in [0,T]}$
$\quad Y^{1:B}_{0:K} \overset{\mathrm{i.i.d.}}{\sim} \mathrm{SamplingRDS}(g^{\theta_n})$, see Algorithm 1
$\quad$ ▷ Apply a stochastic gradient descent step on $\theta_n$
$\quad$ Compute a MC estimator $\hat{\mathcal{L}}_{\mathrm{RDS}}(\theta_n)$ of the loss $\mathcal{L}_{\mathrm{RDS}}(\theta_n)$ defined in (7) with samples $Y^{1:B}_{0:K}$
$\quad$ Compute the gradient $\nabla_\theta \hat{\mathcal{L}}_{\mathrm{RDS}}(\theta_n)$ and update $\theta_n$ to $\theta_{n+1}$ with Adam optimizer
**Output:** Learned guidance $g^{\theta_N}$

---

---

**Algorithm 3:** Gaussian Mixture Model LRDS (GMM-LRDS) : training stage

---

**Input:** Time discretization $\{t_k\}_{k=0}^K$ of $[0, T]$, target density $\gamma$, location of the target modes $\{\mathbf{x}_i\}_{i=1}^I$, number of Markov chains per target mode $M \geq 1$, size of Markov chains $N_{\text{tot}} \geq 1$, effective size of Markov chains $N_{\text{eff}} \in [\![1, N_{\text{tot}}]\!]$, number of reference modes $J \geq I$

▷ (a) Obtain *reference samples*

For each $i \in \{1, \ldots, I\}$, build via MALA $M$ Markov chains $\{X_{0:N_{\text{tot}}}^{i,m}\}_{m=1}^M$ of size $N_{\text{tot}}$ starting at $\mathbf{x}_i$

Keep the last $N_{\text{eff}}$ samples of each Markov chain to define $\hat{\pi}^{\text{ref}} \cong \{X_{N_{\text{tot}}-N_{\text{eff}}+1:N_{\text{tot}}}^{i,m}\}_{i=1,m=1}^{I,M}$

▷ (b) Define the reference process

**if** $J = 1$ **then**

$\quad$ Fit on $\hat{\pi}^{\text{ref}}$ a Gaussian model with parameterized density $\gamma^\varphi$ (Maximum Likelihood estimation)

**else**

$\quad$ Fit on $\hat{\pi}^{\text{ref}}$ a $J$-component Gaussian Mixture Model with parameterized density $\gamma^\varphi$ (EM algorithm)

Set $\gamma^{\text{ref}} = \gamma^\varphi$ and compute $\{s_{T-t_k}^{\text{ref}}\}_{k=0}^{K-1}$ analytically, see Appendix B

▷ (c) Run the RDS variational optimization

Run Algorithm 2

**Output:** Trained GMM-LRDS sampler

---

**Algorithm 4:** Energy-Based Model LRDS (EBM-LRDS) : training stage

---

**Input:** Time discretization $\{t_k\}_{k=0}^K$ of $[0, T]$, target density $\gamma$, location of the target modes $\{\mathbf{x}_i\}_{i=1}^I$, number of Markov chains per target mode $M \geq 1$, size of Markov chains $N_{\text{tot}} \geq 1$, effective size of Markov chains $N_{\text{eff}} \in [\![1, N_{\text{tot}}]\!]$

▷ (a) Obtain *reference samples*

For each $i \in \{1, \ldots, I\}$, build via MALA $M$ Markov chains $\{X_{0:N_{\text{tot}}}^{i,m}\}_{m=1}^M$ of size $N_{\text{tot}}$ starting at $\mathbf{x}_i$

Keep the last $N_{\text{eff}}$ samples of each Markov chain to define $\hat{\pi}^{\text{ref}} \cong \{X_{N_{\text{tot}}-N_{\text{eff}}+1:N_{\text{tot}}}^{i,m}\}_{i=1,m=1}^{I,M}$

▷ (b) Define the reference process

Based on $\hat{\pi}^{\text{ref}}$, fit a multi-level EBM $E^\varphi$ using Algorithm 11

Set $\gamma^{\text{ref}}(x) = \exp(-E^\varphi(0, x))$ and $s_{T-t_k}^{\text{ref}}(x) = -\nabla_x E^\varphi(T - t_k, x)$ for any $k \in \{0, \ldots, K-1\}$

▷ (c) Run the RDS variational optimization

Run Algorithm 2

**Output:** Trained EBM-RDS sampler

---

# B PRELIMINARIES

**Linear SDE integration.** We first dispense a useful lemma to compute exact integration in SDEs with linear drift.

**A0** (Integrability conditions on $f$ and $\beta$)**.** *Coefficients $f : [0, T] \to \mathbb{R}$ and $\beta : [0, T] \to (0, \infty)$ are such that (a) $f$ is integrable on $(0, T)$ and (b) $\beta$ is integrable on $(0, T)$.*

**Lemma 2.** *Let $T > 0$ and $b \in \mathbb{R}^d$. Consider the SDE defined on $[0, T]$ by $\mathrm{d}Y_t = f(t)(Y_t + b)\mathrm{d}t + \sqrt{\beta(t)}\mathrm{d}B_t$, where coefficients $f$ and $\beta$ verify A0. Then, for any pair of time-steps $(s, t)$ such that $T \geq t > s \geq 0$, the conditional distribution of $Y_t$ given $Y_s = y_s \in \mathbb{R}^d$, denoted by $p_{t|s}(\cdot|y_s)$, verifies*

$$p_{t|s}(\cdot|y_s) = \mathrm{N}\left(\exp(\textstyle\int_s^t f(u)\mathrm{d}u)y_s + \left(\exp(\textstyle\int_s^t f(u)\mathrm{d}u) - 1\right)b, \textstyle\int_s^t \beta(u)\exp(2\textstyle\int_u^t f(r)\mathrm{d}r)\mathrm{d}u\,\mathrm{I}_d\right).$$

*Proof.* Assume A0. Define the function $\zeta : t \in [0, T] \to \exp(-\int_0^t f(u)\mathrm{d}u)$ and consider the stochastic process $(Z_t)_{t\in[0,T]}$ defined by $Z_t = \zeta(t)Y_t$ for any $t \in [0, T]$. By Îto's formula, we have $\mathrm{d}Z_t = f(t)\zeta(t)b\mathrm{d}t + \zeta(t)\sqrt{\beta(t)}\mathrm{d}B_t = -\dot{\zeta}(t)b\mathrm{d}t + \zeta(t)\sqrt{\beta(t)}\mathrm{d}B_t$. Therefore, for any time-steps $(s, t)$ such that $T \geq t > s \geq 0$, we have

$$\zeta(t)Y_t - \zeta(s)Y_s = \{\zeta(s) - \zeta(t)\}b + \textstyle\int_s^t \zeta(u)\sqrt{\beta(u)}\mathrm{d}B_u \ ,$$

and then

$$Y_t = \exp(\textstyle\int_s^t f(u)\mathrm{d}u)Y_s + \left(\exp(\textstyle\int_s^t f(u)\mathrm{d}u) - 1\right)b + \textstyle\int_s^t \sqrt{\beta(u)}\exp(\textstyle\int_u^t f(r)\mathrm{d}r)\mathrm{d}B_u \ ,$$

which gives the result using Îto's isometry and the fact that $Y_s$ is independent from $(B_t - B_s)_{t\in[s,T]}$. $\quad\square$

**General SDE integration.** Consider a general SDE of the form $\mathrm{d}Y_t = f(t)(Y_t + g_t(Y_t))\mathrm{d}t + \sqrt{\beta(t)}\mathrm{d}B_t$ defined on $[0, T]$, where $f$, $\beta$ satisfy **A0** and $g : [0, T] \times \mathbb{R}^d \to \mathbb{R}^d$ is a black-box function. Let $(s, t)$ be a pair of time-steps such that $T \geq t > s \geq 0$. We aim to compute the conditional distribution of $Y_t$ given $Y_s = y_s \in \mathbb{R}^d$. While this distribution is intractable for general function $g$, it can be approximated by a tractable Gaussian kernel if $t$ is close enough to $s$. Below, we restate two common first-order methods that produce such approximation following the result of Lemma 2.

(a) *Euler-Maruyama (EM) scheme*, which consists of exactly integrating on time interval $[s, t]$ the SDE defined by $\mathrm{d}Y_u = f(s)(y_s + g_s(y_s))\mathrm{d}u + \sqrt{\beta(s)}\mathrm{d}B_u$ with constant drift and constant volatility.

(b) *Exponential Integration (EI) scheme* (Durmus & Moulines, 2015), more precise than the EM scheme, which consists of exactly integrating on time interval $[s, t]$ the SDE defined by $\mathrm{d}Y_u = f(u)(Y_u + g_s(y_s))\mathrm{d}u + \sqrt{\beta(u)}\mathrm{d}B_u$ with linear drift, where the intractable term is frozen at $g_s(y_s)$.

**Computation of scores for Gaussian mixtures.** Below, we provide a useful lemma to compute the marginal scores of a noising diffusion process applied to a general Gaussian mixture.

**Lemma 3.** *Let $\pi \in \mathscr{P}(\mathbb{R}^d)$. Consider the noising process $(X_t)_{t \in [0,T]}$ defined by SDE* (2) *where $X_0 \sim \pi$. Then, for any $t \in (0, T)$, the conditional distribution of $X_t$ given $X_0 = x_0 \in \mathbb{R}^d$ is defined by the Gaussian kernel*

$$q_{t|0}(\cdot|x_0) = \mathrm{N}(S(t)x_0, S(t)^2\sigma^2(t)\,\mathrm{I}_d)\,, \tag{9}$$

*where $S(t) = \exp(\int_0^t f(u)\mathrm{d}u)$ and $\sigma^2(t) = \int_0^t \beta(u)/S(u)^2\mathrm{d}u$.*

*In particular, if $\pi$ is a Gaussian distribution i.e., $\pi = \mathrm{N}(\mathbf{m}, \Sigma)$ with mean $\mathbf{m} \in \mathbb{R}^d$ and covariance $\Sigma \in \mathbb{R}^{d \times d}$, then for any $t \in [0, T]$, the marginal distribution of $X_t$ is Gaussian, defined by*

$$p_t = \mathrm{N}(S(t)\mathbf{m}, S(t)^2\Sigma + S(t)^2\sigma^2(t)\,\mathrm{I}_d)\,. \tag{10}$$

*On the other hand, if $\pi$ is a Gaussian mixture i.e., $\pi = \sum_{j=1}^{J} w_j\mathrm{N}(\mathbf{m}_j, \Sigma_j)$ where $J \geq 1$ is the number of components, $\{w_j\}_{j=1}^{J} \in \Delta_J$ are the component weights, $\mathbf{m}_j \in \mathbb{R}^d$ and $\Sigma_j \in \mathbb{R}^{d \times d}$ are respectively the mean and the covariance of the $j$-th component, for any $t \in [0, T]$, the marginal distribution of $X_t$ is a Gaussian mixture, defined by*

$$p_t = \sum_{j=1}^{J} w_j\mathrm{N}(S(t)\mathbf{m}_j, S(t)^2\Sigma_j + S(t)^2\sigma^2(t)\,\mathrm{I}_d)\,. \tag{11}$$

*Note that differentiating* (11) *provides closed-form solutions of the score of the stochastic process.*

*Proof.* The proof of (9) directly follows from Lemma 2. Then, (10) and (11) are obtained by applying the affine mapping defined by the transition kernel (9) to the target distribution $\pi$. □

In particular, (11) demonstrates that any noising diffusion scheme applied to a Gaussian mixture does not show any mode switching.

**Integral probability Metrics.** Finally, we dispense the definition of standard probability metrics, that will be used to assess the performance of the considered samplers, *when having access to ground truth samples*. Consider two distributions $\mu \in \mathscr{P}(\mathbb{R}^d)$ and $\nu \in \mathscr{P}(\mathbb{R}^d)$ that we aim to compare.

We recall that the 2-Wasserstein distance between $\mu$ and $\nu$ is given by

$$W_2(\mu, \nu) = \inf\{\int_{\mathbb{R}^d \times \mathbb{R}^d} \|x_1 - x_0\|^2\mathrm{d}\pi(x_0, x_1) : \pi \in \mathscr{P}(\mathbb{R}^d \times \mathbb{R}^d), \pi_0 = \mu, \pi_1 = \nu\}^{1/2}\,,$$

where $\pi_i$ denotes the $i$-th marginal of $\pi$ for $i \in \{0, 1\}$. In practice, we rather turn to its entropy regularized version (Peyré et al., 2019)

$$W_{2,\varepsilon}(\mu, \nu) = \inf\{\int_{\mathbb{R}^d \times \mathbb{R}^d} \|x_1 - x_0\|^2\mathrm{d}\pi(x_0, x_1) - \mathscr{H}(\pi) : \pi \in \mathscr{P}(\mathbb{R}^d \times \mathbb{R}^d), \pi_0 = \mu, \pi_1 = \nu\}^{1/2}\,,$$

where $\varepsilon > 0$ is a regularization hyper-parameter and $\mathscr{H}(\pi) = -\int_{\mathbb{R}^d \times \mathbb{R}^d} \log\pi(x_0, x_1)\mathrm{d}\pi(x_0, x_1)$ refers to as the entropy of $\pi$. When having access to samples from $\mu$ and $\nu$, we compute a statistical

estimation of $W_{2,\varepsilon}(\mu, \nu)$ via Sinkhorn algorithm (Cuturi, 2013) with $\varepsilon = 10^{-3}$, based on the GPU-oriented implementation from `https://github.com/fwilliams/scalable-pytorch-sinkhorn`.

We also consider the Maximum Mean Discrepancy (MMD) (Gretton et al., 2012), which quantifies the dissimilarity between $\mu$ and $\nu$ by comparing their mean embeddings in a reproducing kernel Hilbert space $\mathcal{H}_k$ with kernel $k$, as followed

$$\mathrm{MMD}(\mu, \nu) = \sup_{f \in \mathcal{H}_K : \|f\|_{\mathcal{H}_K} \leq 1} \mathbb{E}[f(X) - f(Y)], \ X \sim \mu, Y \sim \nu \ .$$

In particular, $\mathrm{MMD}(\mu, \nu) = 0$ if and only if $\mu = \nu$ almost surely. When having access to samples $\{x_i\}_{i=1}^n$ from $\mu$ and $\{y_i\}_{i=1}^m$ from $\nu$, we compute an unbiased statistical estimation of $\mathrm{MMD}(\mu, \nu)$ given by

$$\sqrt{\tfrac{2}{n(n-1)} \sum_{1 \leq i < j \leq n} k(x_i, x_j) + \tfrac{2}{m(m-1)} \sum_{1 \leq i < j \leq m} k(y_i, y_j) - \tfrac{2}{mn} \sum_{i=1}^n \sum_{j=1}^m k(x_i, y_j)} \ ,$$

where $k(x, y) = \exp\left(-\|x - y\|^2 / (2\alpha)\right)$ is the squared exponential kernel whose positive bandwidth is fixed as the median of the squared pairwise distances computed on the joint set $\{x_i\}_{i=1}^n \cup \{y_j\}_{j=1}^m$ (Gretton et al., 2012).

Finally, we recall that the Kolmogorov-Smirnov (KS) distance between $\mu$ and $\nu$ is defined by

$$\mathrm{KS}(\mu, \nu) = \sup_{x \in \mathbb{R}^d} |F_\mu(x) - F_\nu(x)| \ ,$$

where $F_\mu$ (respectively $F_\nu$) denotes the cumulative distribution function of $\mu$ (respectively $\nu$). When having access to samples from $\mu$ and $\nu$, we implement a statistical estimation of the sliced KS distance (Grenioux et al., 2023) using 128 random projections.

## C  DETAILS ON NOISING DIFFUSION PROCESSES

In this section, we consider a noising diffusion process $(X_t)_{t \in [0, T]}$ induced by SDE (1) for an arbitrary target distribution $\pi \in \mathscr{P}(\mathbb{R}^d)$ and $T > 0$, and denote by $\mathbb{P}^\star \in \mathscr{P}(\mathbf{C}_T)$ the corresponding path measure. Below, we provide detailed results on this diffusion process, depending on the choice of $f$ and $\beta$: we first provide details on the general setting in Appendix C.1; then, we fully describe the Pinned Brownian Motion and the Variance-Preserving scheme in Appendix C.2 and Appendix C.3, that are used in our experiments; finally, we present novel noising schemes in Appendix C.4.

### C.1  GENERAL NOISING SCHEME

Here, we consider the most general form of SDE (1).

**Lemma 4.** *Let $\pi \in \mathscr{P}(\mathbb{R}^d)$. Assume that $f$ and $\beta$ both verify A0. Then, for any pair of time-steps $(s, t)$ such that $T \geq t > s \geq 0$, the conditional distribution of $X_t$ given $X_s = x_s \in \mathbb{R}^d$ is defined by the Gaussian kernel*

$$q_{t|s}(\cdot|x_s) = \mathrm{N}\left(\exp(\log S(t) - \log S(s))x_s, S(t)^2\{\sigma^2(t) - \sigma^2(s)\} \mathrm{I}_d\right) \ , \tag{12}$$

*where $S(t) = \exp(\int_0^t f(u)\mathrm{d}u)$ and $\sigma^2(t) = \int_0^t \beta(u)/S(u)^2 \mathrm{d}u$.*

*Proof.* The proof of (12) directly follows from Lemma 2. $\qquad\square$

### C.2  PINNED BROWNIAN MOTION

For any $t \in [0, T]$, define $\alpha(t) = \int_0^t \beta(t)\mathrm{d}t$ and consider the case where $f(t) = -\frac{\beta(t)}{\alpha(T) - \alpha(t)}$. Then, SDE (1) can be rewritten as

$$\mathrm{d}X_t = -\frac{\beta(t)X_t}{\alpha(T) - \alpha(t)}\mathrm{d}t + \sqrt{\beta(t)}\mathrm{d}W_t \ , \ X_0 \sim \pi \ .$$

This noising scheme, known as the *Pinned Brownian Motion* (PBM) and previously considered by Tzen & Raginsky (2019); Zhang & Chen (2022); Vargas et al. (2023b), can be obtained by applying a Doob's h-transform on the uncontrolled SDE $dX_t = \sqrt{\beta(t)}dW_t$, associated to path measure $\mathbb{Q}$, to hit 0 at time $T$. Interestingly, it can be shown that the resulting path measure $(\mathbb{P}^\star)^R$ is solution to the following stochastic optimal control problem

$$\arg\min\{\mathrm{KL}(\mathbb{P} \mid \mathbb{Q}) \,:\, \mathbb{P} \in \mathscr{P}(\mathbf{C}_T), \mathbb{P}_T = \pi\} \,,$$

which is often referred to as a half Schrödinger Bridge problem.

**Lemma 5.** *Let $\pi \in \mathscr{P}(\mathbb{R}^d)$. Assume that $f(t) = -\frac{\beta(t)}{\alpha(T)-\alpha(t)}$ and $\beta$ verifies **A0**. Then, for any pair of time-steps $(s,t)$ such that $T \geq t > s \geq 0$, the conditional distribution of $X_t$ given $X_s = x_s \in \mathbb{R}^d$ is defined by the Gaussian kernel*

$$q_{t|s}(\cdot|x_s) = \mathrm{N}\left(\frac{\alpha(T)-\alpha(t)}{\alpha(T)-\alpha(s)}x_s, \frac{(\alpha(T)-\alpha(t))(\alpha(t)-\alpha(s))}{\alpha(T)-\alpha(s)}\mathrm{I}_d\right) \,. \tag{13}$$

*Since $p_T^\star(x) = \int_{\mathbb{R}^d} q_{T|0}(x|x_0)d\pi(x_0)$, it results that $\mathbb{P}_T^\star = \delta_0$.*

*Proof.* The proof of (13) directly follows from Lemma 2. $\qquad\square$

Following Lemma 5, the PBM is an 'exact' noising scheme and we have $\pi^{\mathrm{base}} = \delta_0$ in this setting. Moreover, under mild assumptions on $\pi$, the time-reversed SDE (2) writes as

$$dY_t = \beta(T-t)\left\{\frac{Y_t}{\alpha(T)-\alpha(T-t)} + \nabla\log p_{T-t}^\star(Y_t)\right\}dt + \sqrt{\beta(T-t)}dB_t, \; Y_0 = 0 \,.$$

**On the choice of the $\beta$-schedule.** Previous works have considered constant schedule $\beta(t) = \sigma^2$ (and therefore, $\alpha(t) = \sigma^2 t$), where $\sigma > 0$ can be arbitrarily chosen, see e.g., Zhang & Chen (2022); Richter et al. (2023). We also follow this setting in practice.

## C.3 Variance-Preserving diffusion

For any $t \in [0,T]$, denote $\alpha(t) = \int_0^t \beta(t)dt$ and consider the case where $f(t) = -\beta(t)/2\sigma^2$ with $\sigma > 0$ and $\beta$ being such that $\int_0^T \beta(s)ds \gg 1$. Then, SDE (1) can be rewritten as

$$dX_t = -\frac{\beta(t)X_t}{2\sigma^2}dt + \sqrt{\beta(t)}dW_t \,, \; X_0 \sim \pi \,.$$

This noising scheme, known as the *Variance-Preserving* (VP) scheme (Song et al., 2021) and previously considered by Vargas et al. (2023a), is an Ornstein-Uhlenbeck diffusion process and is largely used in score-based generative models.

**Lemma 6.** *Let $\pi \in \mathscr{P}(\mathbb{R}^d)$ and $\sigma > 0$. Assume that $f(t) = -\beta(t)/2\sigma^2$ and $\beta$ verifies **A0** such that $\int_0^T \beta(s)ds \gg 1$. Then, for any pair of time-steps $(s,t)$ such that $T \geq t > s \geq 0$, the conditional distribution of $X_t$ given $X_s = x_s \in \mathbb{R}^d$ is defined by the Gaussian kernel*

$$q_{t|s}(\cdot|x_s) = \mathrm{N}\left(\sqrt{1-\lambda_{s,t}}x_s, \sigma^2\lambda_{s,t}\mathrm{I}_d\right) \,, \tag{14}$$

*where $\lambda_{s,t} = 1 - \exp(\alpha(s)-\alpha(t))$. Since $p_T^\star(x) = \int_{\mathbb{R}^d} q_{T|0}(x|x_0)d\pi(x_0)$, it results that $\mathbb{P}_T^\star \approx \mathrm{N}(0, \sigma^2\mathrm{I}_d)$.*

*Proof.* The proof of (14) directly follows from Lemma 2, along with fact that $\lambda_{0,T} \approx 1$. $\qquad\square$

Following Lemma 6, the VP scheme is an 'ergodic' noising scheme, converging exponentially fast to the Gaussian distribution $\mathrm{N}(0, \sigma^2\mathrm{I}_d)$; therefore, we have $\pi^{\mathrm{base}} = \mathrm{N}(0, \sigma^2\mathrm{I}_d)$ in this setting. Moreover, under mild assumptions on $\pi$, the time-reversed SDE (2) writes as

$$dY_t = \frac{\beta(T-t)}{2}\{Y_t + 2\sigma^2\nabla\log p_{T-t}^\star(Y_t)\}dt + \sigma\sqrt{\beta(T-t)}dB_t, \; Y_0 \sim \mathbb{P}_T^\star \,,$$

after simple linear time reparameterization.

**On the choice of the $\beta$-schedule.** Previous works have considered a linear schedule $\beta(t) = \beta_{\min}(1 - t/T) + \beta_{\max}(t/T)$ where $\beta_{\min} = 0.1$, $\beta_{\max} \in \{10, 20\}$ and $T = 1$, see e.g., Song et al. (2021); Richter et al. (2023); Reu et al. (2024) or cosine parameterization (Nichol & Dhariwal, 2021; Vargas et al., 2023a), which has been proved to perform better in generative modeling. In our sampling experiments, we did not observe any significant difference between these two settings. Hence, we fix the linear $\beta$-schedule to be the default setting for our numerics (except DDS, originally implemented with the cosine schedule), and let $\sigma$ be arbitrarily chosen.

## C.4 ADDITIONAL EXAMPLES OF NOISING DIFFUSION PROCESSES

Below, we present original noising schemes, but did not consider them in our experiments.

**Pinned Ornstein-Uhlenbeck Motion.** We first propose to consider the case where $f(t) = -\frac{\beta(t)}{2} \coth(\int_t^T \frac{\beta(u)}{2} du)$. Then, SDE (1) can be rewritten as

$$\mathrm{d}X_t = -\frac{\beta(t)}{2} \coth(\int_t^T \tfrac{\beta(u)}{2} du) X_t \mathrm{d}t + \sqrt{\beta(t)} \mathrm{d}W_t, \ X_0 \sim \pi \ .$$

Analogously to the PBM, one can show that this noising scheme can be obtained by applying a Doob's h-transform on the Ornstein-Uhlenbeck process induced by the SDE $\mathrm{d}\tilde{X}_t = -\frac{\beta(t)}{2} \tilde{X}_t \mathrm{d}t + \sqrt{\beta(t)} \mathrm{d}W_t$ to hit 0 at time $T$. We refer to this noising diffusion scheme as the *Pinned Ornstein-Uhlenbeck Motion* (POUM).

**Lemma 7.** *Let $\pi \in \mathscr{P}(\mathbb{R}^d)$. Assume that $f(t) = -\frac{\beta(t)}{2} \coth(\int_t^T \frac{\beta(u)}{2} du)$ and $\beta$ verifies A0. Then, for any pair of time-steps $(s, t)$ such that $T \geq t > s \geq 0$, the conditional distribution of $X_t$ given $X_s = x_s \in \mathbb{R}^d$ is defined by the Gaussian kernel $q_{t|s}(\cdot|x_s)$ given by*

$$\mathrm{N}\left( \frac{|\sinh(\int_t^T \frac{\beta(u)}{2} du)|}{|\sinh(\int_s^T \frac{\beta(u)}{2} du)|} x_s, 2 \sinh(\int_t^T \tfrac{\beta(u)}{2} du)^2 \left\{ \coth(\int_s^T \tfrac{\beta(u)}{2} du) - \coth(\int_t^T \tfrac{\beta(u)}{2} du) \right\} \mathrm{I}_d \right) \ . \tag{15}$$

*Since $p_T^\star(x) = \int_{\mathbb{R}^d} q_{T|0}(x|x_0) \mathrm{d}\pi(x_0)$, it results that $\mathbb{P}_T^\star = \delta_0$.*

*Proof.* The proof of (15) directly follows from Lemma 2. □

Following Lemma 7, the POUM scheme is an 'exact' noising scheme and we have $\pi^{\mathrm{base}} = \delta_0$ in this setting. Moreover, under mild assumptions on $\pi$, the time-reversed SDE (2) writes as

$$\mathrm{d}Y_t = \frac{\beta(T-t)}{2} \{ \coth(\int_{T-t}^T \tfrac{\beta(u)}{2} du) Y_t + 2\nabla \log p_{T-t}^\star(Y_t) \} \mathrm{d}t + \sqrt{\beta(T-t)} \mathrm{d}B_t, \ Y_0 = 0 \ .$$

**Gaussianized Pinned Brownian Motion.** Let $\zeta : [0, T] \to (0, \infty)$ be a square integrable function on $(0, T)$. Denote $\alpha(t) = \int_0^t \zeta(u) \mathrm{d}u$. Inspired by Dai et al. (2023), we propose to consider $f(t) = -\frac{\zeta(t)}{\alpha(T) - \alpha(t)}$ and $\beta(t) = \frac{2\zeta(t)}{\alpha(T) - \alpha(t)}$. Then, SDE (1) can be rewritten as

$$\mathrm{d}X_t = -\frac{\zeta(t) X_t}{\alpha(T) - \alpha(t)} \mathrm{d}t + \sqrt{\frac{2\zeta(t)}{\alpha(T) - \alpha(t)}} \mathrm{d}W_t, \ X_0 \sim \pi \ .$$

**Lemma 8.** *Let $\pi \in \mathscr{P}(\mathbb{R}^d)$. Assume that $f(t) = -\frac{\zeta(t)}{\alpha(T) - \alpha(t)}$ and $\beta(t) = \frac{2\zeta(t)}{\alpha(T) - \alpha(t)}$, where $\zeta : [0, T] \to (0, \infty)$ is a square integrable function on $(0, T)$. Then, for any pair of time-steps $(s, t)$ such that $T \geq t > s \geq 0$, the conditional distribution of $X_t$ given $X_s = x_s \in \mathbb{R}^d$ is defined by the Gaussian kernel*

$$q_{t|s}(\cdot|x_s) = \mathrm{N}\left( \frac{\alpha(T) - \alpha(t)}{\alpha(T) - \alpha(s)} x_s, \left( 1 - \frac{(\alpha(T) - \alpha(t))^2}{(\alpha(T) - \alpha(s))^2} \right) \mathrm{I}_d \right) \ . \tag{16}$$

*Since $p_T^\star(x) = \int_{\mathbb{R}^d} q_{T|0}(x|x_0) \mathrm{d}\pi(x_0)$, it results that $\mathbb{P}_T^\star = \mathrm{N}(0, \mathrm{I}_d)$.*

*Proof.* The proof of (16) directly follows from Lemma 2. □

Following Lemma 7, the noising scheme given above is 'exact' with base distribution given by $\pi^{\text{base}} = \mathrm{N}(0, \mathrm{I}_d)$, this setting being quite unusual in diffusion model community. Based on this, we refer to this noising diffusion process as the *Gaussianized Pinned Brownian Motion* (GPBM). Moreover, under mild assumptions on $\pi$, the time-reverse SDE (2) writes as

$$\mathrm{d}Y_t = \frac{\zeta(T-t)}{\alpha(T) - \alpha(T-t)} \left\{ Y_t + 2\nabla \log p_{T-t}^{\star}(Y_t) \right\} \mathrm{d}t + \sqrt{\frac{2\zeta(T-t)}{\alpha(T) - \alpha(T-t)}} \mathrm{d}B_t, \ Y_0 \sim \mathrm{N}(0, \mathrm{I}_d) \ .$$

## D  VARIATIONAL OBJECTIVES FOR DIFFUSION-BASED METHODS

In this section, we first provide details on the continuous-time framework of variational diffusion-based methods (including RDS) in Appendix D.1. Then, we describe in Appendix D.2 the RDS discrete-time formulation, related to the variational loss $\mathcal{L}_{\text{RDS}}$ given in (7), and compare this framework to discrete-time formulations of previous variational methods.

In the rest of the section, we consider a target distribution $\pi \in \mathscr{P}(\mathbb{R}^d)$ and recall that $\mathbb{P}^\star$ denotes the path measure associated to the noising SDE (1) initialized at $X_0 \sim \pi$.

### D.1  CONTINUOUS TIME FORMULATION

#### D.1.1  RDS SETTING

Let $\pi^{\text{ref}} \in \mathscr{P}(\mathbb{R}^d)$. We recall that $\mathbb{P}^{\text{ref}}$ denotes the path measure associated to the noising SDE (1) starting at $X_0 \sim \pi^{\text{ref}}$. First note that $\mathbb{P}^\star = \pi \otimes \mathbb{P}^{\text{ref}}_{|0}$. Therefore we have the following relation $\mathbb{P}^\star = \frac{\mathrm{d}\pi}{\mathrm{d}\pi^{\text{ref}}} \cdot \mathbb{P}^{\text{ref}}$, *i.e.*, $\mathbb{P}^\star$ is absolutely continuous with respect to $\mathbb{P}^{\text{ref}}$, with Radon-Nikodym derivative given by $(\mathrm{d}\mathbb{P}^\star/\mathrm{d}\mathbb{P}^{\text{ref}})(X_{[0,T]}) = (\mathrm{d}\pi/\mathrm{d}\pi^{\text{ref}})(X_0)$, for any diffusion process $(X_t)_{t\in[0,T]}$ with *forward* time direction.

Below, we first provide the proof of Proposition 1.

*Proof of Proposition 1.* Assume that $\mathbb{P}^\star_T = \mathbb{P}^{\text{ref}}_T = \pi^{\text{base}}$ and there exists $\theta^\star \in \Theta$ such that $g_t^{\theta^\star} = g_t$. Under those two assumptions, it is clear that $(Y_t^{\theta^\star})_{t\in[0,T]} \sim (\mathbb{P}^\star)^R$; therefore, we have $\mathcal{L}_{\text{LV}}(\theta^\star) = 0$, *i.e.*, $\theta^\star$ achieves optimal solution. We now detail how to obtain (6).

Consider the detached diffusion process $(Y_t^{\hat{\theta}})_{t\in[0,T]} \sim \mathbb{P}^{\hat{\theta}}$. Using the fact that $\mathbb{P}^\star = \frac{\mathrm{d}\pi}{\mathrm{d}\pi^{\text{ref}}} \cdot \mathbb{P}^{\text{ref}}$, the log-ratio in the objective (5) can be computed as

$$\log \frac{\mathrm{d}\mathbb{P}^\theta}{\mathrm{d}(\mathbb{P}^\star)^R}(Y_{[0,T]}^{\hat{\theta}}) = \log \frac{\mathrm{d}\mathbb{P}^\theta}{\mathrm{d}(\mathbb{P}^{\text{ref}})^R}\left(Y_{[0,T]}^{\hat{\theta}}\right) + \log \frac{\gamma^{\text{ref}}}{\gamma}\left(Y_T^{\hat{\theta}}\right) - \log \mathcal{Z}^{\text{ref}} + \log \mathcal{Z} \ . \quad (17)$$

Moreover, by applying Girsanov's theorem to the path measures $(\mathbb{P}^{\text{ref}})^R$ and $\mathbb{P}^\theta$ with the assumption $\mathbb{P}^{\text{ref}}_T = \pi^{\text{base}}$, we have

$$\log \frac{\mathrm{d}\mathbb{P}^\theta}{\mathrm{d}(\mathbb{P}^{\text{ref}})^R}(Y_{[0,T]}^{\hat{\theta}}) = \int_0^T \frac{\beta(T-t)}{2} \left\| g_{T-t}^\theta(Y_t^{\hat{\theta}}) \right\|^2 \mathrm{d}t + \int_0^T \sqrt{\beta(T-t)} g_{T-t}^\theta(Y_t^{\hat{\theta}})^\top \mathrm{d}B_t \ . \quad (18)$$

Since the Radon-Nikodym derivative between path measures and their respective time-reversals is the same, we obtain the explicit LV objective combine the previous results to obtain the LV objective (6) by combining (17) and (18). $\qquad\square$

In contrast, previous reference-based works from Zhang & Chen (2022); Vargas et al. (2023a) relied on the reverse KL continuous-time objective

$$\mathcal{L}_{\text{KL}}^c(\theta) = \mathrm{KL}(\mathbb{P}^\theta \mid (\mathbb{P}^\star)^R) = \mathbb{E}\left[\log \frac{\mathrm{d}\mathbb{P}^\theta}{\mathrm{d}(\mathbb{P}^\star)^R}\left(Y_{[0,T]}^\theta\right)\right] \ , \ Y_{[0,T]}^\theta \sim \mathbb{P}^\theta \quad (19)$$

where the expectation is taken w.r.t. to the parameterized process, not the detached one. In this case too, the objective may be simplified as presented next.

**Proposition 9.** *Assume there exists $\theta^\star \in \Theta$ such that $g_t^{\theta^\star} = g_t$. Then, the loss defined in* (19) *achieves optimal solution at $\theta^\star$ and, setting $\varrho = \log(\gamma^{\mathrm{ref}}/\gamma)$, it simplifies as*

$$\mathcal{L}_{KL}^{\mathrm{c}}(\theta) = \mathbb{E}\left[ \int_0^T \frac{\beta(T-t)}{2} \left\| g_{T-t}^\theta(Y_t^\theta) \right\|^2 \mathrm{d}t + \varrho(Y_T^\theta) \right], \tag{20}$$

*where the equality holds, up to constants independent of $\theta$.*

*Proof.* For any $\theta \in \Theta$, we have by the KL chain rule

$$\mathcal{L}_{\mathrm{KL}}^{\mathrm{c}}(\theta) = \mathrm{KL}(\pi^{\mathrm{base}} \mid \mathbb{P}_T^\star) + \mathbb{E}_{\pi^{\mathrm{base}}}\left[ \mathrm{KL}(\mathbb{P}_{|0}^\theta \mid (\mathbb{P}^\star)_{|0}^R) \right],$$

where the first term does not depend on $\theta$. Then, it is clear that $\mathcal{L}_{\mathrm{KL}}^{\mathrm{c}}$ is minimized when $\theta = \theta^\star$. The result of (20) directly follows from (17) and (18); in particular, the additional constants are $-\log \mathcal{Z}^{\mathrm{ref}}$ and $\log \mathcal{Z}$. $\qquad\square$

Based on the simplification (20), one may recognize a standard formulation of stochastic control-affine problem with terminal cost $\varrho$, where the uncontrolled process is given by $\mathbb{P}^{\mathrm{pref}}$.

### D.1.2 DIS SETTING

In their paper, Berner et al. (2023) seek to approximate the time-reversal of $\mathbb{P}^\star$ for the VP noising scheme, see Appendix C.3. To do so, they consider the class of variational path measures $(\mathbb{P}^\theta)_{\theta \in \Theta} \subset \mathscr{P}(\mathbf{C}_T)$ where for any $\theta \in \Theta$, $\mathbb{P}^\theta$ is associated to the SDE

$$\mathrm{d}Y_t^\theta = \frac{\beta(T-t)}{2} Y_t^\theta \mathrm{d}t + \sigma^2 \beta(T-t) s_{T-t}^\theta(Y_t^\theta)\mathrm{d}t + \sigma\sqrt{\beta(T-t)}\mathrm{d}B_t \,, Y_0^\theta \sim \mathrm{N}(0, \sigma^2 \mathrm{I}_d), \quad (21)$$

where the $\beta$-schedule is detailed in Appendix C.3, and $s^\theta : [0, T] \times \mathbb{R}^d$ is a neural network that aims at approximating the marginal scores of $\mathbb{P}^\star$, i.e., $s^\theta(t, \cdot) \approx \nabla \log p_t^\star$. The resulting variational method, DIS, then consists in minimizing the KL-based objective $\theta \mapsto \mathrm{KL}(\mathbb{P}^\theta \mid (\mathbb{P}^\star)^R)$, later extended by Richter et al. (2023) to the LV-based setting. We refer to (Richter et al., 2023, Section 3.2.) for detailed expressions of these continuous-time losses.

### D.1.3 CMCD SETTING

Alternatively, Vargas et al. (2024) propose to approximate a diffusion process with marginals prescribed by a curve of distributions $(\pi_t)_{t \in [0,T]}$, such that $\pi_0 = \pi^{\mathrm{base}}$ and $\pi_T = \pi$, with unnnormalized densities that are assumed to be tractable. To do so, they consider the class of pairs of variational path measures $\left((\mathbb{P}^\theta, \mathbb{Q}^\theta)\right)_{\theta \in \Theta} \subset \mathscr{P}(\mathbf{C}_T)^2$ where, for any $\theta \in \Theta$, $\mathbb{Q}^\theta$ is associated to the SDE

$$\mathrm{d}X_t^\theta = \frac{\sigma^2}{2}(\nabla \log \pi_{T-t}(X_t^\theta) - h_{T-t}^\theta(X_t^\theta))\mathrm{d}t + \sigma\mathrm{d}W_t, \quad X_0^\theta \sim \pi \,. \tag{22}$$

and $\mathbb{P}^\theta$ is associated to the SDE

$$\mathrm{d}Y_t^\theta = \frac{\sigma^2}{2}(\nabla \log \pi_t(Y_t^\theta) + h_t^\theta(Y_t^\theta))\mathrm{d}t + \sigma\mathrm{d}B_t, \quad Y_0^\theta \sim \pi^{\mathrm{base}} \,, \tag{23}$$

with $h^\theta : [0, T] \times \mathbb{R}^d \to \mathbb{R}^d$ being a neural network. The resulting variational method, CMCD, consists in minimizing the KL-based objective $\theta \mapsto \mathrm{KL}(\mathbb{P}^\theta \mid (\mathbb{Q}^\theta)^R)$ or the LV-based extension. We refer to the original paper for detailed expressions of these continuous-time losses.

### D.2 DISCRETE TIME FORMULATION

Following the setting of Section 2.2, we consider a time discretization of the interval $[0, T]$ given by an increasing sequence of timesteps $\{t_k\}_{k=0}^K$ such that $t_0 = 0^3$, $t_K = T$ and $K \geq 1$. In the rest of the section, we denote $\delta_k = t_{k+1} - t_k$.

---

[3]In the only case of PBM, we consider $t_0 > 0$ to avoid numerical integration error (typically $t_0 = 10^{-4}$).

This section is organized as follows. We first explain in Appendix D.3 the discretization setting that we adopt for RDS, before deriving the RDS variational objective obtained by EM integration (Appendix D.3.1), EI integration in the PBM case (Appendix D.3.2) and EI integration in the VP case (Appendix D.3.3). Finally, for comparison purpose, we provide a discrete time approach to DIS in Appendix D.4 and CMCD in Appendix D.5. All of the presented settings are implemented in our code.

### D.3 GENERAL RDS SETTING

Let $\theta \in \Theta$ and let $\pi^{\mathrm{ref}} \in \mathscr{P}(\mathbb{R}^d)$ be an arbitrary reference distribution. Consider the variational diffusion process $(Y_t^\theta)_{t \in [0,T]}$ defined by SDE (4). We recall that under mild assumptions on $\pi^{\mathrm{ref}}$, see e.g., Cattiaux et al. (2023), the time-reversal of $\mathbb{P}^{\mathrm{ref}}$ is associated to the SDE

$$\mathrm{d}Y_t^{\mathrm{ref}} = -f(T-t)Y_t^{\mathrm{ref}}\mathrm{d}t + \beta(T-t)s_{T-t}^{\mathrm{ref}}(Y_t^{\mathrm{ref}})\mathrm{d}t + \sqrt{\beta(T-t)}\mathrm{d}B_t , \quad Y_0^{\mathrm{ref}} \sim \mathbb{P}_T^{\mathrm{ref}} . \tag{24}$$

Assume that we are given transition kernels $(\mathcal{A}, y_k) \mapsto p_{k+1|k}^\theta(\mathcal{A}|y_k)$ and $(\mathcal{A}, y_k) \mapsto p_{k+1|k}^{\mathrm{ref}}(\mathcal{A}|y_k)$ respectively approximating $\mathbb{P}(Y_{t_{k+1}}^\theta \in \mathcal{A} \mid Y_{t_k}^\theta = y_k)$ and $\mathbb{P}(Y_{t_{k+1}}^{\mathrm{ref}} \in \mathcal{A} \mid Y_{t_k}^{\mathrm{ref}} = y_k)$. Then, for $K$ sufficiently large, the path measures $\mathbb{P}^\theta$ and $(\mathbb{P}^{\mathrm{ref}})^R$ may respectively be approximated by the joint distributions $p_{0:K}^\theta \in \mathscr{P}^{(K+1)}$ and $p_{0:K}^{\mathrm{ref}} \in \mathscr{P}^{(K+1)}$, defined by $p_{0:K}^\theta = \pi^{\mathrm{base}} \prod_{k=0}^{K-1} p_{k+1|k}^\theta$ and $p_{0:K}^{\mathrm{ref}} = \pi^{\mathrm{base}} \prod_{k=0}^{K-1} p_{k+1|k}^{\mathrm{ref}}$. Using the relation $\mathbb{P}^\star = \frac{\mathrm{d}\pi}{\mathrm{d}\pi^{\mathrm{ref}}} \cdot \mathbb{P}^{\mathrm{ref}}$, $(\mathbb{P}^\star)^R$ may also be approximated by the joint distribution $p_{0:K}^\star = \frac{\mathrm{d}\pi}{\mathrm{d}\pi^{\mathrm{ref}}} \cdot p_{0:K}^{\mathrm{ref}} \in \mathscr{P}^{(K+1)}$.

Using this correspondence between path measures and joint distributions, we propose to approximate $\mathcal{L}_{\mathrm{LV}}$ defined in (5) by the discrete time LV-based objective

$$\mathcal{L}_{\mathrm{RDS}}(\theta) = \mathrm{Var}\left[\log \frac{\mathrm{d}p_{0:K}^\theta}{\mathrm{d}p_{0:K}^\star}(Y_{0:K}^{\hat\theta})\right] , \quad Y_{0:K}^{\hat\theta} \sim p_{0:K}^{\hat\theta} . \tag{25}$$

In a similar manner, $\mathcal{L}_{\mathrm{KL}}^c$ defined in (19) may be approximated by the discrete time KL-based objective

$$\mathcal{L}_{\mathrm{KL}}(\theta) = \mathbb{E}\left[\log \frac{\mathrm{d}p_{0:K}^\theta}{\mathrm{d}p_{0:K}^\star}(Y_{0:K}^\theta)\right] , \quad Y_{0:K}^\theta \sim p_{0:K}^\theta . \tag{26}$$

By definition of the joint distributions, and similarly to the continuous-time relation , the log-ratio in (25) and (26) can be simplified as

$$\log \frac{\mathrm{d}p_{0:K}^\theta}{\mathrm{d}p_{0:K}^\star}(Y_{0:K}^{\bar\theta}) = \log \frac{\mathrm{d}p_{0:K}^\theta}{\mathrm{d}p_{0:K}^{\mathrm{ref}}}(Y_{0:K}^{\bar\theta}) + \log \frac{\gamma^{\mathrm{ref}}}{\gamma}(Y_K^{\bar\theta}) + \log \mathcal{Z} - \log \mathcal{Z}^{\mathrm{ref}} . \tag{27}$$

where $\bar\theta \in \{\theta, \hat\theta\}$. In the following sections, building upon (27), we derive the exact expression of the variational loss (25) for each combination of noising and discretization schemes. We emphasize that these derivations are in particular applicable to PIS and DDS settings by taking $\pi^{\mathrm{ref}}$ as given in Table 1.

### D.3.1 EM-BASED RDS

In this section, we consider an arbitrary noising scheme that fits within the framework detailed in Appendix C.1. For any $k \in \{0, \ldots, K-1\}$, define the coefficients

$$w_k^{\mathrm{EM}} = \beta(T-t_k)\delta_k ,$$
$$a_k^{\mathrm{EM}} = 1 - f(T-t_k)\delta_k ,$$
$$b_k^{\mathrm{EM}} = \beta(T-t_k)\delta_k ,$$
$$c_k^{\mathrm{EM}} = \beta(T-t_k)\delta_k .$$

To approximate SDEs (24) and (4) on time interval $[t_k, t_{k+1}]$, we may define the following transition kernels

$$p_{k+1|k}^{\mathrm{ref}}(\cdot|y_k) = \mathrm{N}(a_k^{\mathrm{EM}}y_k + b_k^{\mathrm{EM}}s_{T-t_k}^{\mathrm{ref}}(y_k), c_k^{\mathrm{EM}}\mathrm{I}_d) , \tag{28}$$

$$p_{k+1|k}^\theta(\cdot|y_k) = \mathrm{N}(a_k^{\mathrm{EM}}y_k + b_k^{\mathrm{EM}}\{s_{T-t_k}^{\mathrm{ref}}(y_k) + g_{T-t_k}^\theta(y_k)\}, c_k^{\mathrm{EM}}\mathrm{I}_d) , \tag{29}$$

that can be obtained by applying the EM integration scheme, see Appendix B. In this case, the log-ratio $\log \mathrm{d}p^\theta / \mathrm{d}p^{\mathrm{ref}}$ can be computed as follows.

**Lemma 10.** *Assume that the joint distributions $p^\theta$ and $p^{ref}$ are respectively induced by the transition kernels defined in* (28) *and* (29). *Then, for any $y_{0:K} \in (\mathbb{R}^d)^{K+1}$, we have*

$$\log \frac{\mathrm{d}p^\theta}{\mathrm{d}p^{ref}}(y_{0:K}) = \sum_{k=0}^{K-1} \frac{w_k^{EM}}{2} \|g_{T-t_k}^\theta(y_k)\|^2 + \sum_{k=0}^{K-1} \sqrt{w_k^{EM}} g_{T-t_k}^\theta(y_k)^\top \epsilon_k \ ,$$

$$\text{where } \epsilon_k = \frac{1}{\sqrt{c_k^{EM}}} \left( y_{k+1} - a_k^{EM} y_k - b_k^{EM} \{s_{T-t_k}^{ref}(y_k) + g_{T-t_k}^\theta(y_k)\} \right) \ . \tag{30}$$

*Proof.* In this proof, for ease of reading, we will respectively denote $a_k^{\mathrm{EM}}$, $b_k^{\mathrm{EM}}$, $c_k^{\mathrm{EM}}$ and $w_k^{\mathrm{EM}}$ by $a_k$, $b_k$, $c_k$ and $w_k$. Let $y_{0:K} \in (\mathbb{R}^d)^{K+1}$. We have

$$\log \frac{\mathrm{d}p^\theta}{\mathrm{d}p^{\mathrm{ref}}}(y_{0:K}) = \sum_{k=0}^{K-1} \log \frac{\mathrm{d}p_{k+1|k}^\theta(y_{k+1}|y_k)}{\mathrm{d}p_{k+1|k}^{\mathrm{ref}}(y_{k+1}|y_k)}$$

$$= \sum_{k=0}^{K-1} \left\{ -\frac{1}{2} \|\epsilon_k\|^2 + \frac{1}{2} \|\epsilon_k + (b_k/\sqrt{c_k}) g_{T-t_k}^\theta(y_k)\|^2 \right\}$$

$$= \sum_{k=0}^{K-1} \frac{b_k^2}{2c_k} \|g_{T-t_k}^\theta(y_k)\|^2 + \sum_{k=0}^{K-1} \frac{b_k}{\sqrt{c_k}} g_{T-t_k}^\theta(y_k)^\top \epsilon_k \ ,$$

where $\epsilon_k$ is defined by (30). The result of Lemma 10 directly follows by noting that $w_k = b_k^2/c_k$. $\square$

By combining Lemma 10 with (27), we deduce the expression of the RDS loss in this setting.

**Corollary 11** (Expression of RDS loss – EM setting). *When using the EM scheme to integrate the SDE* (4), *the RDS loss simplifies as*

$$\mathcal{L}_{RDS}(\theta) = \mathrm{Var}\left[ \sum_{k=0}^{K-1} w_k^{EM} g_{T-t_k}^\theta(Y_k)^\top \{g_{T-t_k}^{\hat\theta}(Y_k) - \frac{1}{2} g_{T-t_k}^\theta(Y_k)\} \right.$$

$$\left. + \sum_{k=0}^{K-1} \sqrt{w_k^{EM}} g_{T-t_k}^\theta(Y_k)^\top Z_k + \log \frac{\gamma^{ref}}{\gamma}(Y_K) \right] \ ,$$

*where $\{Z_k\}_{k=0}^{K-1}$ are independently distributed according to $\mathrm{N}(0, \mathrm{I}_d)$, and $\{Y_k\}_{k=0}^K$ is recursively defined by $Y_0 \sim \pi^{base}$, and for any $k \in \{0, \dots, K-1\}$,*

$$Y_{k+1} = a_k^{EM} Y_k + b_k^{EM} \{s_{T-t_k}^{ref}(Y_k) + g_{T-t_k}^{\hat\theta}(Y_k)\} + \sqrt{c_k^{EM}} Z_k \ , \tag{31}$$

*with $\hat\theta$ being a detached version of $\theta$.*

*Proof.* Consider the discrete time process $\{Y_k\}_{k=0}^K$ defined by recursion (31). With the same notation as in Lemma 10, we have for any $k \in \{0, \dots, K-1\}$

$$\epsilon_k = Z_k + \sqrt{w_k^{\mathrm{EM}}} \{g_{T-t_k}^{\hat\theta}(Y_k) - g_{T-t_k}^\theta(Y_k)\} \ ,$$

recalling that $w_k^{\mathrm{EM}} = (b_k^{\mathrm{EM}})^2 / c_k^{\mathrm{EM}}$. Using (27), we finally obtain the expression of the RDS loss. $\square$

For completeness, we also provide a similar result for the KL-based objective given in (26).

**Corollary 12** (Expression of reverse KL loss – EM setting). *When using the EM scheme to integrate the SDE* (4), *up to additional constants independent of $\theta$, the discrete time KL loss simplifies as*

$$\mathcal{L}_{KL}(\theta) = \mathbb{E}\left[ \sum_{k=0}^{K-1} \frac{w_k^{EM}}{2} \|g_{T-t_k}^\theta(Y_k^\theta)\|^2 + \log \frac{\gamma^{ref}}{\gamma}(Y_K^\theta) \right] \ ,$$

where $\{Z_k\}_{k=0}^{K-1}$ are independently distributed according to $\mathrm{N}(0, \mathrm{I}_d)$, and $\{Y_k^\theta\}_{k=0}^K$ is recursively defined by $Y_0^\theta \sim \pi^{base}$, and for any $k \in \{0, \ldots, K-1\}$

$$Y_{k+1}^\theta = a_k^{EM} Y_k^\theta + b_k^{EM}\{s_{T-t_k}^{ref}(Y_k^\theta) + g_{T-t_k}^\theta(Y_k^\theta)\} + \sqrt{c_k^{EM}} Z_k \ .$$

As observed by Vargas et al. (2023a), there may be a significant bias raising from the use of the Euler-Maruyama discretization when using diffusion-based sampling methods. This motivates us to consider EI discretization in the RDS setting.

### D.3.2 EI-BASED RDS (PBM)

We first provide the expression of the RDS loss in the case of the PBM noising scheme, which is detailed in Appendix C.2. We recall the notation $\alpha(t) = \int_0^t \beta(u)\mathrm{d}u$. For any $k \in \{0, \ldots, K-1\}$, define the coefficients

$$w_k^{\mathrm{PBM}} = \frac{(\alpha(T) - \alpha(T - t_k))(\alpha(T - t_k) - \alpha(T - t_{k+1}))}{\alpha(T) - \alpha(T - t_{k+1})} \ ,$$

$$a_k^{\mathrm{PBM}} = \frac{\alpha(T) - \alpha(T - t_{k+1})}{\alpha(T) - \alpha(T - t_k)} \ ,$$

$$b_k^{\mathrm{PBM}} = \alpha(T - t_k) - \alpha(T - t_{k+1}) \ ,$$

$$c_k^{\mathrm{PBM}} = \frac{(\alpha(T) - \alpha(T - t_{k+1}))(\alpha(T - t_k) - \alpha(T - t_{k+1}))}{\alpha(T) - \alpha(T - t_k)} \ .$$

To approximate SDEs (24) and (4) on time interval $[t_k, t_{k+1}]$, we may define the following transition kernels

$$p_{k+1|k}^{\mathrm{ref}}(\cdot|y_k) = \mathrm{N}(a_k^{\mathrm{PBM}} y_k + b_k^{\mathrm{PBM}} s_{T-t_k}^{\mathrm{ref}}(y_k), c_k^{\mathrm{PBM}} \mathrm{I}_d) \ , \tag{32}$$

$$p_{k+1|k}^{\theta}(\cdot|y_k) = \mathrm{N}(a_k^{\mathrm{PBM}} y_k + b_k^{\mathrm{PBM}}\{s_{T-t_k}^{\mathrm{ref}}(y_k) + g_{T-t_k}^\theta(y_k)\}, c_k^{\mathrm{PBM}} \mathrm{I}_d) \ , \tag{33}$$

that can be obtained by applying the EI scheme, see Appendix B. In this case, the log ratio $\log \mathrm{d}p^\theta/\mathrm{d}p^{\mathrm{ref}}$ can be computed as follows.

**Lemma 13.** *Assume that the joint distributions $p^\theta$ and $p^{ref}$ are respectively induced by the transition kernels defined in (32) and (33). Then for any $y_{0:K} \in (\mathbb{R}^d)^{K+1}$, we have*

$$\log \frac{\mathrm{d}p^\theta}{\mathrm{d}p^{ref}}(y_{0:K}) = \sum_{k=0}^{K-1} \frac{w_k^{PBM}}{2}\|g_{T-t_k}^\theta(y_k)\|^2 + \sum_{k=0}^{K-1} \sqrt{w_k^{PBM}} g_{T-t_k}^\theta(y_k)^\top \epsilon_k \ ,$$

*where* $\epsilon_k = \frac{1}{\sqrt{c_k^{PBM}}}\left(y_{k+1} - a_k^{PBM} y_k - b_k^{PBM}\{s_{T-t_k}^{ref}(y_k) + g_{T-t_k}^\theta(y_k)\}\right) \ .$

*Proof.* The proof is exactly the same as Lemma 10, noting that we still have $w_k^{\mathrm{PBM}} = (b_k^{\mathrm{PBM}})^2/c_k^{\mathrm{PBM}}$. □

By combining Lemma 13 with (27), we deduce the expression of the RDS loss in this setting. The proof of the following is the same as in Corollary 11.

**Corollary 14** (Expression of RDS loss – EI setting – PBM scheme)**.** *When using the EI scheme to integrate SDE* (4) *implemented with PBM, the RDS loss simplifies as*

$$\mathcal{L}_{RDS}(\theta) = \mathrm{Var}\left[\sum_{k=0}^{K-1} w_k^{PBM} g_{T-t_k}^\theta(Y_k)^\top\{g_{T-t_k}^{\hat{\theta}}(Y_k) - \frac{1}{2}g_{T-t_k}^\theta(Y_k)\}\right.$$

$$\left. + \sum_{k=0}^{K-1} \sqrt{w_k^{PBM}} g_{T-t_k}^\theta(Y_k)^\top Z_k + \log \frac{\gamma^{ref}}{\gamma}(Y_K)\right] \ ,$$

where $\{Z_k\}_{k=0}^{K-1}$ are independently distributed according to $\mathrm{N}(0, \mathrm{I}_d)$, and $\{Y_k\}_{k=0}^K$ is recursively defined by $Y_0 \sim \pi^{base}$, and for any $k \in \{0, \dots, K-1\}$

$$Y_{k+1} = a_k^{PBM} Y_k + b_k^{PBM} \{s_{T-t_k}^{ref}(Y_k) + g_{T-t_k}^{\hat{\theta}}(Y_k)\} + \sqrt{c_k^{PBM}} Z_k \ ,$$

with $\hat{\theta}$ being a detached version of $\theta$.

For completeness, we also provide a similar result for the KL-based objective given in (26).

**Corollary 15** (Expression of reverse KL loss – EI setting – PBM noising)**.** *When using the EI scheme to integrate SDE (4) implemented with PBM, up to additional constants independent of $\theta$, the discrete time KL loss simplifies as*

$$\mathcal{L}_{KL}(\theta) = \mathbb{E}\left[\sum_{k=0}^{K-1} \frac{w_k^{PBM}}{2} \left\|g_{T-t_k}^{\theta}(Y_k^{\theta})\right\|^2 + \log\frac{\gamma^{ref}}{\gamma}(Y_K^{\theta})\right] \ ,$$

*where $\{Z_k\}_{k=0}^{K-1}$ are independently distributed according to $\mathrm{N}(0, \mathrm{I}_d)$, and $\{Y_k^{\theta}\}_{k=0}^K$ is recursively defined by $Y_0^{\theta} \sim \pi^{base}$, and for any $k \in \{0, \dots, K-1\}$*

$$Y_{k+1}^{\theta} = a_k^{PBM} Y_k^{\theta} + b_k^{PBM} \{s_{T-t_k}^{ref}(Y_k^{\theta}) + g_{T-t_k}^{\theta}(Y_k^{\theta})\} + \sqrt{c_k^{PBM}} Z_k \ .$$

### D.3.3   EI-BASED RDS (VP)

Here, we provide the expression of the RDS loss in the case of the VP noising scheme, which is detailed in Appendix C.3. We recall the notation $\alpha(t) = \int_0^t \beta(u)\mathrm{d}u$. For any $k \in \{0, \dots, K-1\}$, define $\lambda_k = \exp\left((\alpha(T-t_k) - \alpha(T-t_{k+1})) - 1\right.$ and the coefficients

$$
\begin{aligned}
w_k^{\mathrm{VP}} &= \frac{4\sigma^2(\sqrt{1+\lambda_k}-1)^2}{\lambda_k} = 4\sigma^2 \tanh\left(\frac{\alpha(T-t_k) - \alpha(T-t_{k+1})}{4}\right) \\
a_k^{\mathrm{VP}} &= \sqrt{1+\lambda_k} \\
b_k^{\mathrm{VP}} &= 2\sigma^2\{\sqrt{1+\lambda_k}-1\} \\
c_k^{\mathrm{VP}} &= \sigma^2\lambda_k
\end{aligned}
$$

To approximate SDEs (24) and (4) on time interval $[t_k, t_{k+1}]$, we may define the following transition kernels

$$
\begin{aligned}
p_{k+1|k}^{ref}(\cdot|y_k) &= \mathrm{N}(a_k^{\mathrm{VP}} y_k + b_k^{\mathrm{VP}} s_{T-t_k}^{ref}(y_k), c_k^{\mathrm{VP}} \mathrm{I}_d) \ , \\
p_{k+1|k}^{\theta}(\cdot|y_k) &= \mathrm{N}(a_k^{\mathrm{VP}} y_k + b_k^{\mathrm{VP}}\{s_{T-t_k}^{ref}(y_k) + g_{T-t_k}^{\theta}(y_k)\}, c_k^{\mathrm{VP}} \mathrm{I}_d) \ ,
\end{aligned}
\tag{34}
$$

that can be obtained by applying the EI scheme, see Appendix B. In this case, the log ratio $\log \mathrm{d}p^{\theta}/\mathrm{d}p^{ref}$ can be computed as follows.

**Lemma 16.** *Assume that the joint distributions $p^{\theta}$ and $p^{ref}$ are respectively induced by the transition kernels defined in (34) and (34). Then, for any $y_{0:K} \in (\mathbb{R}^d)^{K+1}$, we have*

$$\log\frac{\mathrm{d}p^{\theta}}{\mathrm{d}p^{ref}}(y_{0:K}) = \sum_{k=0}^{K-1} \frac{w_k^{VP}}{2}\|g_{T-t_k}^{\theta}(y_k)\|^2 + \sum_{k=0}^{K-1}\sqrt{w_k^{VP}} g_{T-t_k}^{\theta}(y_k)^{\top}\epsilon_k$$

$$\text{where } \epsilon_k = \frac{1}{\sqrt{c_k^{VP}}}\left(y_{k+1} - a_k^{VP} y_k - b_k^{VP}\{s_{T-t_k}^{ref}(y_k) + g_{T-t_k}^{\theta}(y_k)\}\right)$$

*Proof.* The proof is exactly the same as Lemma 10, noting that we still have $w_k^{\mathrm{VP}} = (b_k^{\mathrm{VP}})^2/c_k^{\mathrm{VP}}$. $\square$

By combining Lemma 16 with (27), we deduce the expression of the RDS loss in this setting. The proof of the following is the same as in Corollary 11.

**Corollary 17** (Expression of RDS loss – EI setting - VP scheme). *When using the EI scheme to integrate SDE* (4) *implemented with VP , the RDS loss simplifies as*

$$\mathcal{L}_{RDS}(\theta) = \mathrm{Var}\left[\sum_{k=0}^{K-1} w_k^{VP} g_{T-t_k}^{\theta}(Y_k)^{\top}\{g_{T-t_k}^{\hat{\theta}}(Y_k) - \frac{1}{2}g_{T-t_k}^{\theta}(Y_k)\}\right.$$
$$\left. + \sum_{k=0}^{K-1} \sqrt{w_k^{VP}} g_{T-t_k}^{\theta}(Y_k)^{\top} Z_k + \log\frac{\gamma^{ref}}{\gamma}(Y_K)\right] ,$$

*where* $\{Z_k\}_{k=0}^{K-1}$ *are independently distributed according to* $\mathrm{N}(0,\mathrm{I}_d)$, *and* $\{Y_k\}_{k=0}^{K}$ *is recursively defined by* $Y_0 \sim \pi^{base}$, *and for any* $k \in \{0,\dots,K-1\}$

$$Y_{k+1} = a_k^{VP} Y_k + b_k^{VP}\{s_{T-t_k}^{ref}(Y_k) + g_{T-t_k}^{\hat{\theta}}(Y_k)\} + \sqrt{c_k^{VP}} Z_k ,$$

*with* $\hat{\theta}$ *being a detached version of* $\theta$.

For completeness, we also provide a similar result for the KL-based objective given in (26).

**Corollary 18** (Expression of reverse KL loss – EI setting - VP noising). *When using the EI scheme to integrate SDE* (4) *implemented with VP, up to additional constants independent of* $\theta$, *the discrete time KL loss simplifies as*

$$\mathcal{L}_{KL}(\theta) = \mathbb{E}\left[\sum_{k=0}^{K-1} \frac{w_k^{VP}}{2} \left\| g_{T-t_k}^{\theta}(Y_k^{\theta})\right\|^2 + \log\frac{\gamma^{ref}}{\gamma}(Y_K^{\theta})\right] ,$$

*where* $\{Z_k\}_{k=0}^{K-1}$ *are independently distributed according to* $\mathrm{N}(0,\mathrm{I}_d)$, *and* $\{Y_k^{\theta}\}_{k=0}^{K}$ *is recursively defined by* $Y_0^{\theta} \sim \pi^{base}$, *and for any* $k \in \{0,\dots,K-1\}$,

$$Y_{k+1}^{\theta} = a_k^{VP} Y_k^{\theta} + b_k^{VP}\{s_{T-t_k}^{ref}(Y_k^{\theta}) + g_{T-t_k}^{\theta}(Y_k^{\theta})\} + \sqrt{c_k^{VP}} Z_k .$$

### D.4 DIS SETTING

Here, we propose a discrete time alternative to the continuous time objective proposed by Berner et al. (2023) and Richter et al. (2023). Let $\theta \in \Theta$. Consider the variational diffusion process $(Y_t^{\theta})_{t\in[0,T]}$ defined by SDE (21).

Assume that we are given transition kernels $(\mathcal{A}, y_k) \mapsto p_{k+1|k}^{\theta}(\mathcal{A}|y_k)$ and $(\mathcal{A}, y_{k+1}) \mapsto p_{k|k+1}^{\star}(\mathcal{A}|y_{k+1})$ respectively approximating $\mathbb{P}(Y_{t_{k+1}}^{\theta} \in \mathcal{A} \mid Y_{t_k}^{\theta} = y_k)$ and $\mathbb{P}(X_{t_k} \in \mathcal{A} \mid X_{t_{k+1}} = y_{k+1})$. Then, for $K$ sufficiently large, the path measures $\mathbb{P}^{\theta}$ and $(\mathbb{P}^{\star})^R$ may be approximated by the joint distributions $p_{0:K}^{\theta} \in \mathscr{P}^{(K+1)}$ and $p_{0:K}^{\star} \in \mathscr{P}^{(K+1)}$ defined by $p_{0:K}^{\theta} = \pi^{base}\prod_{k=0}^{K-1} p_{k+1|k}^{\theta}$ and $p_{0:K}^{\star} = \pi \prod_{k=0}^{K-1} p_{k|k+1}^{\star}$.

Following the RDS methodology, we propose to approximate the KL-based continuous-time DIS objective by

$$\mathcal{L}_{KL}^{DIS}(\theta) = \mathbb{E}\left[\log\frac{\mathrm{d}p_{0:K}^{\theta}}{\mathrm{d}p_{0:K}^{\star}}(Y_{0:K}^{\theta})\right] , \quad Y_{0:K}^{\theta} \sim p_{0:K}^{\theta} , \tag{35}$$

and the LV-based continuous-time DIS objective by

$$\mathcal{L}_{LV}^{DIS}(\theta) = \mathrm{Var}\left[\log\frac{\mathrm{d}p_{0:K}^{\theta}}{\mathrm{d}p_{0:K}^{\star}}(Y_{0:K}^{\hat{\theta}})\right] , \quad Y_{0:K}^{\hat{\theta}} \sim p_{0:K}^{\hat{\theta}}. \tag{36}$$

To approximate SDEs (21) and (1) on time interval $[t_k, t_{k+1}]$ in the case of the VP noising scheme, we may define the following transition kernels

$$p_{k|k+1}^{\star}(\cdot|y_{k+1}) = \mathrm{N}(\sqrt{1-\lambda_{T-t_{k+1},T-t_k}}\,y_{k+1}, \sigma^2\lambda_{T-t_{k+1},T-t_k}\,\mathrm{I}_d) , \tag{37}$$

$$p_{k+1|k}^{\theta}(\cdot|y_k) = \mathrm{N}(a_k^{VP}y_k + b_k^{VP} + s_{T-t_k}^{\theta}(y_k), c_k^{VP}\mathrm{I}_d) , \tag{38}$$

where $\lambda_{s,t}$ is defined in Lemma 6 with $s = T - t_{k+1}$ and $t = T - t_k$, and $\{a_k^{\mathrm{VP}}, b_k^{\mathrm{VP}}, c_k^{\mathrm{VP}}\}$ are defined via the VP-EI integration scheme detailed in Appendix D.3.3. In particular, the following equalities hold:

$$\sqrt{1 - \lambda_{T-t_{k+1}, T-t_k}} = \exp\left(\frac{1}{2}[\alpha(T - t_{k+1}) - \alpha(T - t_k)]\right) = \frac{1}{a_k^{\mathrm{VP}}} \ ,$$

$$\sigma^2 \lambda_{T-t_{k+1}, T-t_k} = \sigma^2[1 - \exp\left(\alpha(T - t_{k+1}) - \alpha(T - t_k)\right)] = \frac{c_k^{\mathrm{VP}}}{(a_k^{\mathrm{VP}})^2} \ .$$

Therefore, we have $p_{k|k+1}^\star(y_k|y_{k+1}) = \mathrm{N}(y_{k+1}; a_k^{\mathrm{VP}} y_k, c_k^{\mathrm{VP}} \mathrm{I}_d)$. In this case, the log-ratio $\log \mathrm{d}p^\theta / \mathrm{d}p^\star$ can be computed as follows.

**Lemma 19.** *Assume that the joint distributions $p^\star$ and $p^\theta$ are respectively induced by the transition kernels defined in (37) and (38). Then, for any $y_{0:K} \in (\mathbb{R}^d)^{K+1}$, we have*

$$\log \frac{\mathrm{d}p^\theta}{\mathrm{d}p^\star}(y_{0:K}) = \log Z + \log \frac{\pi^{base}(y_0)}{\gamma(y_K)} + \sum_{k=0}^{K-1} \log \frac{\mathrm{d}p_{k+1|k}^\theta(y_{k+1}|y_k)}{\mathrm{d}p_{k|k+1}^\star(y_k|y_{k+1})}$$

$$= \log Z + \log \frac{\pi^{base}(y_0)}{\gamma(y_K)} + \sum_{k=0}^{K-1} \frac{w_k^{VP}}{2} \|s_{T-t_k}^\theta(y_k)\|^2 + \sum_{k=0}^{K-1} \sqrt{w_k^{VP}} s_{T-t_k}^\theta(y_k)^\top \epsilon_k$$

*where* $\epsilon_k = \dfrac{1}{\sqrt{c_k^{VP}}}\left(y_{k+1} - a_k^{VP} y_k - b_k^{VP} s_{T-t_k}^\theta(y_k)\right)$

*Proof.* This result lies on the following identity

$$\frac{\mathrm{d}p_{k+1|k}^\theta(y_{k+1}|y_k)}{\mathrm{d}p_{k|k+1}^\star(y_k|y_{k+1})} = -\frac{1}{2}\|\epsilon_k\|^2 + \frac{1}{2}\left\|\epsilon_k + \sqrt{w_k^{\mathrm{VP}}} s_{T-t_k}^\theta(y_k)\right\|^2$$

$\square$

Based on Lemma 19, we simply deduce the expression of the discrete time versions of LV-based and KL-based DIS losses.

**Corollary 20** (Expression of LV-DIS loss). *When using the EI scheme to integrate SDE (21), the LV-based discrete time DIS loss (36) simplifies as*

$$\mathcal{L}_{LV}^{DIS}(\theta) = \mathrm{Var}\left[\sum_{k=0}^{K-1} w_k^{VP} s_{T-t_k}^\theta(Y_k)^\top \{s_{T-t_k}^{\hat{\theta}}(Y_k) - \frac{1}{2} s_{T-t_k}^\theta(Y_k)\}\right.$$

$$\left. + \sum_{k=0}^{K-1} \sqrt{w_k^{VP}} s_{T-t_k}^\theta(Y_k)^\top Z_k + \log \frac{\pi^{base}(Y_0)}{\gamma(Y_K)}\right] \ ,$$

*where $\{Z_k\}_{k=0}^{K-1}$ are independently distributed according to $\mathrm{N}(0, \mathrm{I}_d)$, and $\{Y_k\}_{k=0}^K$ is recursively defined by $Y_0 \sim \pi^{base}$, and for any $k \in \{0, \ldots, K-1\}$,*

$$Y_{k+1} = a_k^{VP} Y_k + b_k^{VP} s_{T-t_k}^{\hat{\theta}}(Y_k) + \sqrt{c_k^{VP}} Z_k \ ,$$

*with $\hat{\theta}$ being a detached version of $\theta$.*

*Proof.* This result is an application from Lemma 19, where we have $\epsilon_k = Z_k + \sqrt{w_k^{\mathrm{VP}}}\{s_{T-t_k}^{\hat{\theta}}(Y_k) - s_{T-t_k}^\theta(Y_k)\}$ for any $k \in \{0, \ldots, K-1\}$. $\square$

**Corollary 21** (Expression of KL-DIS loss). *When using the EM scheme to integrate SDE (21), up to additional constants independent of $\theta$, the KL-based discrete time DIS loss (35) simplifies as*

$$\mathcal{L}_{KL}^{DIS}(\theta) = \mathbb{E}\left[\sum_{k=0}^{K-1} \frac{w_k^{VP}}{2}\left\|s_{T-t_k}^\theta(Y_k^\theta)\right\|^2 + \log \frac{\pi^{base}(Y_0^\theta)}{\gamma(Y_K^\theta)}\right] \ ,$$

where $\{Z_k\}_{k=0}^{K-1}$ are independently distributed according to $\mathrm{N}(0, \mathrm{I}_d)$, and $\{Y_k^\theta\}_{k=0}^K$ is recursively defined by $Y_0^\theta \sim \pi^{base}$, and for any $k \in \{0, \dots, K-1\}$,

$$Y_{k+1}^\theta = a_k^{VP} Y_k^\theta + b_k^{VP} s_{T-t_k}^\theta(Y_k^\theta) + \sqrt{c_k^{VP}} Z_k \ .$$

## D.5 CMCD SETTING

Here, we propose a discrete time alternative to the continuous time objective proposed by Vargas et al. (2024). Let $\theta \in \Theta$. Consider the variational diffusion processes $(Y_t^\theta)_{t \in [0,T]}$ defined by SDE (23), and $(X_t^\theta)_{t \in [0,T]}$, defined by (22).

Assume that we are given transition kernels $(\mathcal{A}, y_k) \mapsto p_{k+1|k}^\theta(\mathcal{A}|y_k)$ and $(\mathcal{A}, y_{k+1}) \mapsto q_{k|k+1}^\theta(\mathcal{A}|y_{k+1})$ respectively approximating $\mathbb{P}(Y_{t_{k+1}}^\theta \in \mathcal{A} \mid Y_{t_k}^\theta = y_k)$ and $\mathbb{Q}^\theta(X_{t_k}^\theta \in \mathcal{A} \mid X_{t_{k+1}}^\theta = y_{k+1})$. Then, for $K$ sufficiently large, the path measures $\mathbb{P}^\theta$ and $(\mathbb{Q}^\theta)^R$ may be approximated by the joint distributions $p_{0:K}^\theta \in \mathscr{P}^{(K+1)}$ and $q_{0:K}^\theta \in \mathscr{P}^{(K+1)}$ defined by $p_{0:K}^\theta = \pi^{base} \prod_{k=0}^{K-1} p_{k+1|k}^\theta$ and $q_{0:K}^\theta = \pi \prod_{k=0}^{K-1} q_{k|k+1}^\theta$.

Following the RDS methodology, we propose to approximate the KL-based continuous-time CMCD objective by

$$\mathcal{L}_{\mathrm{KL}}^{\mathrm{CMCD}}(\theta) = \mathbb{E}\left[\log \frac{\mathrm{d}p_{0:K}^\theta}{\mathrm{d}q_{0:K}^\theta}(Y_{0:K}^\theta)\right] \ , \ Y_{0:K}^\theta \sim p_{0:K}^\theta \ , \tag{39}$$

and the LV-based continuous-time CMCD objective by

$$\mathcal{L}_{\mathrm{LV}}^{\mathrm{CMCD}}(\theta) = \mathrm{Var}\left[\log \frac{\mathrm{d}p_{0:K}^\theta}{\mathrm{d}q_{0:K}^\theta}(Y_{0:K}^{\hat{\theta}})\right] \ , \ Y_{0:K}^{\hat{\theta}} \sim p_{0:K}^{\hat{\theta}} \ . \tag{40}$$

For any $k \in \{0, \dots, K-1\}$, let us define $w_k^{\mathrm{CMCD}} = \sigma^2 \delta_k / 4$.

To approximate SDEs (22) and (23) on time interval $[t_k, t_{k+1}]$, we may define the following transition kernels

$$q_{k|k+1}^\theta(\cdot|y_{k+1}) = \mathrm{N}(y_{k+1} + \frac{\sigma^2 \delta_k}{2}(\nabla \log \pi_{t_{k+1}}(y_{k+1}) - h_{t_{k+1}}^\theta(y_{k+1})), \sigma^2 \delta_k \, \mathrm{I}_d) \ , \tag{41}$$

$$p_{k+1|k}^\theta(\cdot|y_k) = \mathrm{N}(y_k + \frac{\sigma^2 \delta_k}{2}(\nabla \log \pi_{t_k}(y_k) + h_{t_k}^\theta(y_k)), \sigma^2 \delta_k \, \mathrm{I}_d) \ , \tag{42}$$

that can be obtained by applying the EM integration scheme, see Appendix B. In this case, the log-ratio $\log \mathrm{d}p^\theta / \mathrm{d}q^\theta$ can be computed as follows.

**Lemma 22.** *Assume that the joint distributions $q^\theta$ and $p^\theta$ are respectively induced by the transition kernels defined in* (41) *and* (42). *Then, for any $y_{0:K} \in (\mathbb{R}^d)^{K+1}$, we have*

$$\log \frac{\mathrm{d}p^\theta}{\mathrm{d}q^\theta}(y_{0:K}) = \log Z + \log \frac{\pi^{base}(y_0)}{\gamma(y_K)} + \sum_{k=0}^{K-1} \log \frac{\mathrm{d}p_{k+1|k}^\theta(y_{k+1}|y_k)}{\mathrm{d}q_{k|k+1}^\theta(y_k|y_{k+1})}$$

$$= \log Z + \log \frac{\pi^{base}(y_0)}{\gamma(y_K)} + \frac{1}{2}\sum_{k=0}^{K-1} w_k^{CMCD} \|u_k\|^2 + \sum_{k=0}^{K-1} \sqrt{w_k^{CMCD}} u_k^\top \epsilon_k \ ,$$

*where*

$$u_k = \nabla \log \pi_{t_k}(y_k) + \nabla \log \pi_{t_{k+1}}(y_{k+1}) + h_{t_k}^\theta(y_k) - h_{t_{k+1}}^\theta(y_{k+1}) \ ,$$

$$\epsilon_k = \frac{1}{\sigma\sqrt{\delta_k}}\left(y_{k+1} - y_k - \frac{\sigma^2 \delta_k}{2}\{\nabla \log \pi_{t_k}(y_k) + h_{t_k}^\theta(y_k)\}\right) \ .$$

*Proof.* Based on the notation of $u_k$ and $\epsilon_k$, we obtain the result by noting that

$$\log \frac{\mathrm{d}p_{k+1|k}^\theta(y_{k+1}|y_k)}{\mathrm{d}q_{k|k+1}^\theta(y_k|y_{k+1})} = -\frac{1}{2}\|\epsilon_k\|^2 + \frac{1}{2}\left\|-\epsilon_k - \sqrt{w_k^{CMCD}} u_k\right\|^2$$

$\square$

Based on Lemma 22, we simply deduce the expression of the discrete time versions of LV-based and KL-based CMCD losses.

**Corollary 23** (Expression of LV-CMCD loss). *When using the EM scheme to integrate SDE (23), the LV-based discrete time CMCD loss (40) simplifies as*

$$\mathcal{L}_{LV}^{CMCD}(\theta) = \mathrm{Var}\left[\log\frac{\pi^{base}(Y_0)}{\gamma(Y_K)} + \sum_{k=0}^{K-1} w_k^{CMCD}\left\{\frac{1}{2}\left\|u_k^\theta\right\|^2 + (h_{t_k}^{\hat\theta}(Y_k) - h_{t_k}^\theta(Y_k))^\top u_k^\theta\right\}\right.$$
$$\left. + \sum_{k=0}^{K-1}\sqrt{w_k^{CMCD}}(u_k^\theta)^\top Z_k\right],$$
$$with\ u_k^\theta = \nabla\log\pi_{t_k}(Y_k) + \nabla\log\pi_{t_{k+1}}(Y_{k+1}) + h_{t_k}^\theta(Y_k) - h_{t_{k+1}}^\theta(Y_{k+1}),$$

*where $\{Z_k\}_{k=0}^{K-1}$ are independently distributed according to $\mathrm{N}(0, \mathrm{I}_d)$, and $\{Y_k\}_{k=0}^K$ is recursively defined by $Y_0 \sim \pi^{base}$, and for any $k \in \{0, \ldots, K-1\}$,*

$$Y_{k+1} = Y_k + \frac{\sigma^2\delta_k}{2}\{\nabla\log\pi_{t_k}(Y_k) + h_{t_k}^{\hat\theta}(Y_k)\} + \sigma\sqrt{\delta_k}Z_k.$$

*Proof.* This result is an application from Lemma 22, where we have $\epsilon_k = Z_k + \sqrt{w_k^{\mathrm{CMCD}}}\{h_{t_k}^{\hat\theta}(Y_k) - h_{t_k}^\theta(Y_k)\}$ for any $k \in \{0, \ldots, K-1\}$. ∎

**Corollary 24** (Expression of KL-CMCD loss). *When using the EM scheme to integrate SDE (23), up to additional constants independent of $\theta$, the KL-based discrete time CMCD loss (39) simplifies as*

$$\mathcal{L}_{KL}^{CMCD}(\theta) = \mathbb{E}\left[\log\frac{\pi^{base}(Y_0^\theta)}{\gamma(Y_K^\theta)} + \frac{1}{2}\sum_{k=0}^{K-1} w_k^{CMCD}\left\|u_k^\theta\right\|^2\right],$$
$$with\ u_k^\theta = \nabla\log\pi_{t_k}(Y_k^\theta) + \nabla\log\pi_{t_{k+1}}(Y_{k+1}^\theta) + h_{t_k}^\theta(Y_k^\theta) - h_{t_{k+1}}^\theta(Y_{k+1}^\theta),$$

*where $\{Z_k\}_{k=0}^{K-1}$ are independently distributed according to $\mathrm{N}(0, \mathrm{I}_d)$, and $\{Y_k^\theta\}_{k=0}^K$ is recursively defined by $Y_0^\theta \sim \pi^{base}$, and for any $k \in \{0, \ldots, K-1\}$,*

$$Y_{k+1}^\theta = Y_k^\theta + \frac{\sigma^2\delta_k}{2}\{\nabla\log\pi_{t_k}(Y_k^\theta) + h_{t_k}^\theta(Y_k^\theta)\} + \sigma\sqrt{\delta_k}Z_k.$$

# E ANNEALED MCMC SAMPLERS

In this section, we present a line of Monte Carlo algorithms that aim at sampling from a sequence of distributions $\{p_k\}_{k=0}^K \subset \mathscr{P}(\mathbb{R}^d)$, with $K \geq 1$, defined such that $p_0$ is an easy-to sample distribution and $p_k$ is harder and harder to sample as $k$ increases. In this section, we assume that each distribution $p_k$ is absolutely continuous with respect to the Lebesgue measure, with tractable *unnormalized* density and tractable score, where both can be evaluated pointwise.

**Reminders on the Metropolis-Adjusted Langevin Algorithm (MALA).** This sampling algorithm is the only one in this section to be designed so as to sample from a single distribution $\pi \in \mathscr{P}(\mathbb{R}^d)$ with density $\gamma : \mathbb{R}^d \to \mathbb{R}_+$ known up to a normalizing constant. It is an extension of the largely used Unadjusted Langevin Algorithm (ULA) (Roberts & Tweedie, 1996).

Given a number of steps $L \geq 1$, a step-size $\lambda > 0$ and an easy-to-sample distribution $p^{\mathrm{init}} \in \mathscr{P}(\mathbb{R}^d)$, ULA builds a Markov chain $(X^\ell)_{\ell=0}^L$ defined by $X^0 \sim p^{\mathrm{init}}$ and for any $\ell \in \{0, \ldots, L-1\}$,

$$X^{\ell+1} = X^\ell + \lambda\nabla\log\gamma(X^\ell) + \sqrt{2\lambda}Z^\ell,\ Z^\ell \sim \mathrm{N}(0, \mathrm{I}_d),$$

which reduces to a time discretization of the Langevin diffusion that admits $\pi$ as invariant distribution. Building upon this recursion, MALA (Roberts & Tweedie, 1996) addresses the discretization bias by adding a Metropolis-Hastings acceptance/rejection step at each iteration, see Algorithm 5. In practice, one can easily adapt the step-size $\lambda$ by targeting a specific acceptance ratio. We follow this paradigm in our numerical experiments, see Appendix H for more details.

---

**Algorithm 5:** Metropolis-Adjusted Langevin Algorithm (MALA)

---

**Input:** Target distribution $\pi$ with unnormalized density $\gamma$, step size $\lambda > 0$, number of steps $L \geq 1$, initial distribution $p^{\text{init}}$

▷ Initialization

$X^0 \sim p^{\text{init}}$

**for** $\ell \in \{0, \ldots, L-1\}$ **do**

    ▷ Propose a new sample

    $\tilde{X}^{\ell+1} = X^\ell + \lambda \nabla \log \gamma(X^\ell) + \sqrt{2\lambda} Z^\ell, \quad Z^\ell \sim N(0, I_d)$

    ▷ Compute the acceptance ratio

    $\alpha = \min\left(1, \frac{\gamma(\tilde{X}^{\ell+1})q(X^\ell | \tilde{X}^{\ell+1})}{\gamma(X^\ell)q(\tilde{X}^{\ell+1}|X^\ell)}\right)$ where $q(y \mid x) = N(y; x + \lambda \nabla \log \gamma(x), 2\lambda I_d)$

    ▷ Accept or reject depending on $\alpha$

    $U \sim U([0,1])$

    **if** $U \leq \alpha$ **then**

        $X^{\ell+1} = \tilde{X}^{\ell+1}$

    **else**

        $X^{\ell+1} = X^\ell$

**Output:** Sequence of samples $X^{0:L}$ such that $X^{0:L} \overset{\text{iid}}{\sim} \pi$ (approximately)

---

**Annealed Langevin MCMC (AL-MCMC).** Based on exact samples from $p_0$, the *Annealed Langevin MCMC* algorithm (Song & Ermon, 2019; 2020) consists in sequentially sampling from $p_{k+1}$ with MALA (Algorithm 5) initialized in the last samples obtained at step $k$. The core idea behind this procedure is that the warm initialization obtained by the easy-to-sample distributions (*i.e.*, when $k$ is low) can help to overcome sampling issues, such as high energy barriers, in the more complex distributions (*i.e.*, when $k$ is large). We refer to Algorithm 6 for the pseudo-code of this approach.

---

**Algorithm 6:** Annealed Langevin MCMC (AL-MCMC) algorithm

---

**Input:** Sequence of target distributions $\{p_k\}_{k=0}^K$, step-sizes $\{\lambda_k\}_{k=1}^K \in (0, \infty)^K$, number of MALA steps per level $L \geq 1$

▷ Initialization

$X_0^{0:L} \overset{\text{iid}}{\sim} p_0$

**for** $k = 0, \ldots, K-1$ **do**

    ▷ Sample from $p_{k+1}$ by starting at samples obtained from $p_k$

    $X_{k+1}^{0:L} = \text{MALA}(p_{k+1}, \lambda_{k+1}, L, \delta_{X_k^L})$, see Algorithm 5

**Output:** Sequence of samples $X_{0:K}^{0:L}$ such that $X_k^{0:L} \overset{\text{iid}}{\sim} p_k$ (approximately)

---

**Sequential Monte Carlo (SMC) and extension.** The *Sequential Monte Carlo* (Neal, 2001; Del Moral et al., 2006) algorithm lies on the same paradigm as Algorithm 6 but proceeds in *parallel* over $N$ chains (which are referred as *particles*) while ensuring that the warm initialization is correct. To do so, SMC relies on an additional resampling step that may occur before certain MALA run, based on accumulated importance weights that we next define.

Let $N \geq 1$, $k \in \{0, \ldots, K-1\}$, $(\tilde{w}_k^n)_{n=1}^N$ be unnormalized importance weights accumulated on each chain up to step $k$ ($\tilde{w}_k^n = 1/N$ if $k = 0$), and $(x_k^n)_{n=1}^N$ be samples from $p_k$ obtained independently as the last samples of $N$ parallel MALA runs (or exactly obtained if $k = 0$). Then, the SMC importance weights designed to sample from $p_{k+1}$, denoted by $(\tilde{w}_{k+1}^n)_{n=1}^N$, and their renormalized versions, denoted by $(w_{k+1}^n)_{n=1}^N$, are defined as

$$\tilde{w}_{k+1}^n = \frac{p_{k+1}(x_k^n)}{p_k(x_k^n)} \tilde{w}_k^n, \quad w_{k+1}^n = \frac{\tilde{w}_{k+1}^n}{\sum_{j=1}^N \tilde{w}_{k+1}^j}.$$

The SMC resampling step then consists in obtaining $N$ new samples $(y_{k+1}^n)_{n=1}^N$ by random selection among the original samples $(x_k^n)_{n=1}^N$ via a multinomial distribution (with replacement) defined by the normalized weights $(w_{k+1}^n)_{n=1}^N$. These novel samples will then be used as starting points for

running MALA on the target $p_{k+1}$. In practice, this resampling step only occurs adaptively based on the value of the accumulated Effective Sample Size (ESS) defined at step $k$ by

$$\text{ESS}_k = \frac{\left(\sum_{n=1}^N \tilde{w}_k^n\right)^2}{\sum_{n=1}^N (\tilde{w}_k^n)^2} \ .$$

We detail the whole SMC procedure in Algorithm 7.

---

**Algorithm 7:** Sequential Monte Carlo (SMC) algorithm with adaptive resampling

---

**Input:** Sequence of target distributions $\{p_k\}_{k=0}^K$, step-sizes $\{\lambda_k\}_{k=1}^K \in (0, \infty)^K$, number of MCMC steps per level $L \geq 1$, number of particles $N \geq 1$, ESS resampling threshold $\alpha \in (0, 1]$

▷ Initialization
$X_0^{1:N} \overset{\text{iid}}{\sim} p_0$ , $\tilde{W}_0^{1:N} = 1/N$
**for** $k = 0, \ldots, K-1$ **do**
    ▷ 1. Compute the SMC importance weights (in parallel, for $n = 1, \ldots, N$)
    $\tilde{W}_{k+1}^n = (p_{k+1}(X_k^n)/p_k(X_k^n)) \tilde{W}_{k+1}^n$
    ▷ 2. Compute the accumulated ESS (in parallel, for $n = 1, \ldots, N$)
    $\text{ESS}_{k+1} = \left(\sum_{n=1}^N \tilde{W}_k^n\right)^2 / \left(\sum_{n=1}^N (\tilde{W}_k^n)^2\right)$
    ▷ 3. Resample $X_k^{1:N}$ if the ESS too low
    **if** $\text{ESS}_{k+1} < \alpha N$ **then**
        ▷ Normalize the importance weights (in parallel, for $n = 1, \ldots, N$)
        $W_{k+1}^n = \tilde{W}_{k+1}^n / \sum_{j=1}^N \tilde{W}_{k+1}^j$
        ▷ Resample the particles
        $I^{1:N} \sim \mathcal{M}(W_{k+1}^1, \ldots, W_{k+1}^N)$
        $Y_{k+1}^{1:N} = X_k^{I^{1:N}}$ , $\tilde{W}_k^{1:N} = 1/N$
    **else**
        $Y_{k+1}^n = X_k^n$
    ▷ 4. Sample from $p_{k+1}$ by starting from sample $Y_{k+1}^n$ (in parallel, for $n = 1, \ldots, N$)
    $\tilde{X}_{k+1}^{1:L} = \text{MALA}(p_{k+1}, \lambda_{k+1}, L, \delta_{Y_{k+1}^n})$, see Algorithm 5
    $X_{k+1}^n = \tilde{X}_{k+1}^L$
**Output:** Sequence of samples $X_{0:K}^{1:N}$ such that $X_k^{1:N} \overset{\text{iid}}{\sim} p_k$ (approximately)

---

Recently, Phillips et al. (2024) proposed an extension of the SMC algorithm, *Particle Denoising Diffusion Sampler* (PDDS), where they consider the specific case where the intermediate distributions $\{p_k\}_{k=0}^K$ are defined as the approximate marginals of a denoising diffusion process. Below, we explain the fundamentals of PDDS.

Let $\pi \in \mathscr{P}(\mathbb{R}^d)$ be a hard-to-sample distribution. With the same notations as in Section 2.1, consider the noising process $(X^\star)_{t \in [0,T]}$ induced by the following SDE

$$\mathrm{d}X_t^\star = f(t)X_t^\star \mathrm{d}t + \sqrt{\beta(t)}\mathrm{d}W_t \ , \ X_0^\star \sim \pi \ . \tag{43}$$

Then, assuming that $X_T^\star \sim \pi^{\text{base}}$, the time-reversal of SDE (43) is given by

$$\mathrm{d}Y_t^\star = -f(T-t)Y_t^\star \mathrm{d}t + \beta(T-t)\nabla \log p_{T-t}^\star(Y_t^*)\mathrm{d}t + \sqrt{\beta(t)}\mathrm{d}B_t, \quad Y_0^\star \sim \pi^{\text{base}} \ , \tag{44}$$

where $p_t^\star$ is the density of the marginal distribution associated to $X_t^\star$, which is intractable in general. Assume that we are given a sequence of intermediate unnormalized densities $(p_t)_{t \in [0,T]}$ that approximate the target densities $(p_t^\star)_{t \in [0,T]}$.

In particular, the time-reversed SDE (44) can be approximated by

$$\mathrm{d}Y_t = -f(T-t)Y_t\mathrm{d}t + \beta(T-t)\nabla \log p_{T-t}(Y_t)\mathrm{d}t + \sqrt{\beta(T-t)}\mathrm{d}B_t \ , Y_0 \sim \pi^{\text{base}} \ . \tag{45}$$

Consider a time discretization $\{t_k\}_{k=0}^K$ such that $t_0 = 0$ and $t_K = T$, and the sequence of target distributions $\{p_k\}_{k=0}^K$ defined by $p_k = p_{T-t_k}$, such that $p_k$ is expected to be harder and harder to sample from as $k$ increases.

Let $k \in \{0, \ldots, K-1\}$, define $\delta_k = t_{k+1} - t_k$. Since SDE (43) is linear, it can be exactly integrated on time interval $[T - t_{k+1}, T - t_k]$, see Lemma 2. We denote by $p^\star_{k|k+1}$ the corresponding transition density. On the other hand, SDE (45) can only be *approximately* integrated on time interval $[t_k, t_{k+1}]$ (due to the score term in the drift). For instance, this can be done using the Euler-Maruyama transition kernel[4]

$$p_{k+1|k}(\cdot | y_k) = \mathrm{N}(y_k - f(T - t_k)\delta_k y_k + \beta(T - t_k)\delta_k \nabla \log p_{T-t_k}(y_k), \beta(T - t_k)\delta_k \, \mathrm{I}_d) \ .$$

To sample sequentially from $\{p_k\}_{k=0}^K$, Phillips et al. (2024) suggest to apply the SMC procedure detailed in Algorithm 7, where the $k$-th resampling step (1-3) is now defined as follows.

0. Sample $\hat{X}_{k+1}^n \sim p_{k+1|k}(\cdot \mid X_k^n)$ (in parallel, for $n = 1, \ldots, N$);

1. Compute the importance weights (in parallel, for $n = 1, \ldots, N$)

$$\tilde{W}_{k+1}^n = \frac{p_{k+1}(\hat{X}_{k+1}^n) p^\star_{k|k+1}(X_k^n \mid \hat{X}_{k+1}^n)}{p_k(X_k^n) p_{k+1|k}(\hat{X}_{k+1}^n \mid X_k^n)} \tilde{W}_k^n;$$

2. Apply the resampling procedure (based on the ESS value) to the particles $\hat{X}_{k+1}^{1:N}$ with importance weights given by $\{W_{k+1}^n\}_{n=1}^N$.

Hence, this novel resampling step amounts to reweight the samples obtained from the *approximate* time-reversed SDE (45) by using the *exact* noising scheme as proposal.

**Replica Exchange (RE).** The *Replica Exchange* algorithm (Swendsen & Wang, 1986) aims at sampling from the annealed distributions $\{p_k\}_{k=0}^K$ with $K+1$ parallel Markov chains, *i.e.*, the $k$-th chain targets $p_k$. For the sake of pedagogy, we assume here that $K$ is even.

At each level, RE sampling is done via a local MCMC sampler, such as MALA, which is expected to have poor performance for large $k$. To alleviate this issue, every $L$ MALA steps, RE randomly performs a *swapping* between Markov chains. More specifically, each consecutive pair $(k, k+1)$ of Markov chains (*i.e.*, either $(k, k+1) \in \{(0, 1), \ldots, (K-2, K-1)\}$ or $(k, k+1) \in \{(1, 2), \ldots, (K-1, K)\}$), with respective current states $X_k$ and $X_{k+1}$, is swapped with probability

$$p_{k,k+1} = \min \left( 1, \frac{p_k(X_{k+1}) p_{k+1}(X_k)}{p_k(X_k) p_{k+1}(X_{k+1})} \right) \ .$$

We provide the pseudo-code of this algorithm in Algorithm 9.

---

**Algorithm 8:** Swapping step of Replica Exchange

---

**Input:** Sequence of target distributions $\{p_k\}_{k=0}^K$ with even $K$, current samples $X_{1:K}$, swapping state $s \in \{0, 1\}$

**if** $s = 0$ **then**
   | Indexes $= \{0, 2, \ldots, K-2\}$
**else**
   | Indexes $= \{1, 3, \ldots, K-1\}$

▷ Compute the swapping probability between $k$ and $k+1$ (in parallel, for $k \in$ Indexes)
$p_{k,k+1} = \min \left( 1, \frac{p_k(X_{k+1}) p_{k+1}(X_k)}{p_k(X_k) p_{k+1}(X_{k+1})} \right)$
▷ Swap depending on $p_{k,k+1}$ (in parallel, for $k \in$ Indexes)
$U \sim \mathrm{U}([0, 1])$
**if** $U < p_{k,k+1}$ **then**
   | $(X_k, X_{k+1}) = (X_{k+1}, X_k)$

**Output:** Randomly swapped sequence $X_{0:K}$

---

[4]In the original paper, the authors consider the VP noising scheme and compute $p_{k+1|k}$ as the transition kernel obtained by Exponential Integration of SDE (45) on $[t_k, t_{k+1}]$.

---

**Algorithm 9:** Replica Exchange (RE) algorithm

---

**Input:** Sequence of target distributions $\{p_k\}_{k=0}^K$ with even $K$, step-sizes $\{\lambda_k\}_{k=1}^K \in (0, \infty)^K$, total
   number of MCMC steps $M \geq 1$, swap frequency $S \geq 1$

▷ Initialization
$X_{0:K}^0 \overset{\text{iid}}{\sim} p_0$
$s = 0$
**for** $m = 0, \ldots, M - 1$ **do**
  **if** $(m + 1) \bmod S == 0$ **then**
     ▷ Swap Markov chains for indexes determined by $s$
     $X_{0:K}^{m+1} = \text{swapRE}(X_{0:K}^m, s)$, see Algorithm 8
     $s = (s + 1) \bmod 2$
  **else**
     ▷ Sample locally (in parallel, for $k = 0, \ldots, K$)
     $X_k^{m+1} = \text{MALA}(p_k, \lambda_k, 1, \delta_{X_k^m})$, see Algorithm 5

**Output:** Sequence of samples $X_{0:K}^{0:M}$ such that $X_k^{0:M} \overset{\text{iid}}{\sim} p_k$ (approximately)

---

**Known limitations of annealed MCMC samplers.** In general, annealed MCMC samplers usually target a sequence of distributions $\{p_k\}_{k=0}^K$, which forms a geometric bridge between an easy-to-sample distribution $\pi^{\text{base}}$ and a hard-to-sample distribution $\pi$, *i.e.*, $p_k \propto (\pi^{\text{base}})^{(K-k)/K} \pi^{k/K}$. However, this scheme suffer from mode-switching for multi-modal distribution $\pi$ (Phillips et al., 2024), *i.e.*, the mode weights of the intermediate distributions may vary a lot as $k$ increases. Therefore, transitions between annealing levels is hard in practice, as samples can easily get stuck in a high-probability modes. Moreover, due to the use of reweighting schemes, SMC and RE are highly sensitive to the overlap between consecutive distributions, which requires to take $K$ large in complex multi-modal settings, hence increasing the complexity of the methods.

## F ENERGY-BASED MODELS

This section is dedicated to describing Energy-Based Models.

### F.1 SINGLE-LEVEL EBMS

Let $p \in \mathscr{P}(\mathbb{R}^d)$. We assume having access to samples from $p$ but not to its density. Energy-based Models (EBM) perform a density estimation task by modeling the density of $p$ with $p^\varphi(x) = \exp(-E^\varphi(x))/\mathcal{Z}^\varphi$, where $E^\varphi : \mathbb{R}^d \to \mathbb{R}$ is a neural network and $\mathcal{Z}^\varphi = \int_{\mathbb{R}^d} \exp(-E^\varphi(x))\mathrm{d}x$ is an unknown normalizing constant. This model is trained by maximizing the log-likelihood of the model $\mathcal{L}(\varphi) = \mathbb{E}[\log p^\varphi(X)]$ where $X \sim p$, whose gradients can be computed as

$$\nabla_\varphi \mathcal{L}(\varphi) = \mathbb{E}[\nabla_\varphi E^\varphi(X^-)] - \mathbb{E}[\nabla_\varphi E^\varphi(X^+)], \quad X^- \sim p^\varphi, \ X^+ \sim p \,. \tag{46}$$

Given samples from $p$ (referred to as positive samples) and samples from $p^\varphi$ (referred to as negative samples), one can compute a Monte-Carlo estimator of $\nabla_\varphi \mathcal{L}(\varphi)$ using (46). In practice, the negative samples are usually obtained by running a MCMC sampler on $p^\varphi$ (which is possible because it is known up to a normalizing constant). However, obtaining truthful negative samples is the main challenge of the EBM training procedure as $p^\varphi$ is expected to be as multi-modal as $p$.

### F.2 MULTI-LEVEL EBMS

Let $p \in \mathscr{P}^{(K+1)}$, with $K \geq 1$. For $k = 1, \ldots, K$, we assume having access to samples from $p_k$ but not to its density. Regarding $p_0$, we assume it to be known entirely. Our goal is to approximate the density of $p_k$ at every level $k \in \{1, \ldots, K\}$. A naive approach would consist in defining single-level EBMs for each $k$. Assuming that the distributions $p_k$ are not completely unrelated, doing this would be however very inefficient. This section presents alternatives to this approach, which are all implemented in our code.

### F.2.1 PRIOR WORKS

**Diffusion Recovery Likelihood (DRL).** Recently, Gao et al. (2021); Zhu et al. (2024) proposed the DRL framework, based on the extra assumption that for any $k \in \{0, \ldots, K-1\}$, the conditional distribution $p_{k|k+1}$ is known and given by $p_{k|k+1}(\cdot \mid x_{k+1}) = \mathrm{N}(\alpha_{k+1}x_{k+1}, \sigma_{k+1}^2 \mathrm{I}_d)$ for some $\alpha_{k+1} \in \mathbb{R}$ and $\sigma_{k+1} > 0$. This is typically the case when the joint distribution $p$ is defined via a discrete time approximation of a linear SDE, see Appendix C. In the following, we denote $Y_k = \alpha_k X_k$ where $X_k \sim p_k$. In DRL, the authors suggest to use a multi-level EBM $p_k^\varphi(\cdot) = \exp(-E^\varphi(k, \cdot))/\mathcal{Z}_k^\varphi$, where $E^\varphi : \{1, \ldots, K\} \times \mathbb{R}^d \to \mathbb{R}$ is a neural network and the normalizing constant $\mathcal{Z}_k^\varphi = \int_{\mathbb{R}^d} \exp(-E^\varphi(k, x))\mathrm{d}x$ is intractable. In this case, the density of $p_k$ is approximated, up to a multiplicative constant, by $\exp(-E^\varphi(k, \cdot))$. Using Bayes rule, they define for any $k \in \{0, \ldots, K-1\}$ the following conditional EBM

$$p_{k+1|k}^\varphi(y_{k+1} \mid x_k) \propto p_{k|k+1}(x_k \mid y_{k+1}/\alpha_{k+1})p_{k+1}^\varphi(y_{k+1})$$

$$= (\mathcal{Z}^\varphi(k+1, x_k))^{-1} \exp\left(-\frac{1}{2\sigma_{k+1}^2} \|x_k - y_{k+1}\|^2 - E^\varphi(k+1, y_{k+1})\right),$$

where $\mathcal{Z}^\varphi(k+1, x_k) = \int_{\mathbb{R}^d} \exp\left(-\left(2\sigma_{k+1}^2\right)^{-1} \|x_k - y\|^2 - E^\varphi(k+1, y)\right) \mathrm{d}y$. Note that if $\sigma_{k+1}^2$ is small enough, for instance when $K$ is large, then the quadratic term dominates in the expression of $\log p_{k+1|k}^\varphi(\cdot \mid x_k)$, which means that this conditional distribution localizes on $x_k$. Therefore, by applying a first order Taylor expansion of the energy term at $x_k$, the following Gaussian approximation holds

$$p_{k+1|k}^\varphi(y_{k+1} \mid x_k) \approx \mathrm{N}\left(x_k - \sigma_{k+1}^2 \nabla E^\varphi(k+1, x_k), \sigma_{k+1}^2 \mathrm{I}_d\right).$$

Hence, if $\sigma_{k+1}^2$ is chosen small enough, this conditional distribution will be very easy to sample from. Based on this observation, the authors suggest to use the following maximum likelihood objective

$$\mathcal{L}^{\mathrm{DRL}}(\varphi) = \sum_{k=0}^{K-1} \mathcal{L}_k^{\mathrm{DRL}}(\varphi), \text{ with } \mathcal{L}_k^{\mathrm{DRL}}(\varphi) = \mathbb{E}\left[\log p_{k+1|k}^\varphi(\alpha_{k+1} X_{k+1} \mid X_k)\right], \tag{47}$$

where $X_{k+1} \sim p_{k+1}$ and $X_k \sim p_{k|k+1}(\cdot|X_{k+1})$. In particular, the gradient of the single-level DRL loss defined in (47) can be expressed as

$$\nabla_\varphi \mathcal{L}_k^{\mathrm{DRL}}(\varphi) = \mathbb{E}[\nabla_\varphi E^\varphi(k, Y_k^-)] - \mathbb{E}[\nabla_\varphi E^\varphi(k, \alpha_k X_k^+)],$$

where $X_k \sim p_k$, $Y_{k+1}^- \sim p_{k+1|k}^\varphi(\cdot \mid X_k)$ and $X_{k+1}^+ \sim p_{k+1}$. Up to a multiplicative constant, the density of $p_k$ can then be approximated by $x_k \mapsto p_k^\varphi(\alpha_k x_k)$.

---

**Algorithm 10:** Gibbs-within-Langevin MCMC transition kernel

---

**Input:** Previous state $(k, x) \in \{0, \ldots, K\} \times \mathbb{R}^d$, step-size $\lambda > 0$
▷ Sample $\tilde{x}$ given $k$ with standard MCMC
$\tilde{x} = \mathrm{MALA}(p^\varphi(\cdot \mid k), \lambda)$ where $p^\varphi(\cdot \mid k) \propto p_k^\varphi(\cdot)$
▷ Sample $\tilde{k}$ given $\tilde{x}$ with a multinomial distribution
$\tilde{k} \sim \mathcal{M}\left(\exp(-E^\varphi(0, \tilde{x})), \ldots, \exp(-E^\varphi(K, \tilde{x}))\right)$
**Output:** Next state $(\tilde{k}, \tilde{x})$

---

**Diffusion Assisted EBM (DA-EBM).** On the other hand, (Zhang et al., 2023) suggest to solve the problem by jointly modeling indexes and states. Here, the multi-level EBM is expressed as $p_k^\varphi(x) = \exp(-E^\varphi(k, x))/\mathcal{Z}^\varphi$, where $E^\varphi : \{0, \ldots, K\} \times \mathbb{R}^d \to \mathbb{R}$ is a neural network and the normalizing constant $\mathcal{Z}^\varphi = \sum_{k=0}^K \int_{\mathbb{R}^d} \exp(-E^\varphi(k, x))\mathrm{d}x$, which can be learn by maximizing the following maximum likelihood objective

$$\mathcal{L}^{\mathrm{DAEBM}}(\varphi) = \mathbb{E}\left[\log p_k^\varphi(X_k)\right], \quad k \sim \mathrm{U}(\{0, \ldots, K\}), \quad X_k \sim p_k. \tag{48}$$

The gradient of objective (48) can be written as

$$\nabla_\varphi \mathcal{L}^{\mathrm{DAEBM}}(\varphi) = \mathbb{E}[\nabla_\varphi E^\varphi(k^-, X^-)] - \mathbb{E}[\nabla_\varphi E^\varphi(k^+, X_k^+)],$$

where $(k^-, X^-) \sim p^\varphi$, $k^+ \sim \mathrm{U}(\{0, \ldots, K\})$ and $X_k^+ \sim p_{k^+}$. In the same fashion as Kim & Ye (2023), the authors suggest to perform the negative sampling from $p^\varphi$ by doing a Gibbs-within-Langevin procedure, whose transition kernel is summarized in Algorithm 10.

### F.2.2 OUR APPROACH: USING AN ANNEALED MCMC SAMPLER AS BACKBONE

In this paper, we also suggest to learn a multi-level EBM approximating $p$, defined by $p_k^\varphi(x) = \exp\left(-E^\varphi(k,x)\right)/\mathcal{Z}_k^\varphi$, where $E^\varphi : \{0,\ldots,K\} \times \mathbb{R}^d \to \mathbb{R}$ is a neural network and the normalizing constant $\mathcal{Z}_k^\varphi = \int_{\mathbb{R}^d} \exp\left(-E^\varphi(k,x)\right) \mathrm{d}x$. We propose to learn $\varphi$ by maximizing a simple yet novel maximum likelihood joint objective

$$\mathcal{L}^{\mathrm{ours}}(\varphi) = \sum_{k=0}^{K} \mathcal{L}_k^{\mathrm{ours}}(\varphi) \ , \ \mathcal{L}_k^{\mathrm{ours}}(\varphi) = \mathbb{E}\left[\log p_k^\varphi(X_k)\right], \ \ X_k \sim p_k \ , \tag{49}$$

with single-level gradient given by

$$\nabla_\varphi \mathcal{L}_k^{\mathrm{ours}}(\varphi) = \mathbb{E}[\nabla_\varphi E^\varphi(k, X_k^-)] - \mathbb{E}[\nabla_\varphi E^\varphi(k, X_k^+)] \ , \ X_k^- \sim p_k^\varphi \ , \ X_k^+ \sim p_k \ .$$

Unlike previous multi-level EBM algorithms, the negative sampling phase can be simply done in our framework by leveraging annealed MCMC algorithms presented in Appendix E on the sequence of EBM densities $\{p^\varphi(k,\cdot)\}_{k=0}^K$. This allows us to kill two birds in one stone as (i) we get negative samples for each single-level EBM and (ii) we overcome the negative sampling limitations of each EBM by leveraging the correlation between the consecutive levels. In practice, we use Replica Exchange as default annealed MCMC sampler because of its massively parallel capabilities. We summarize our training procedure in Algorithm 11. For ease of reading, we presented the continuous time analog of this approach in the main part of the paper, see Section 3.3.

---

**Algorithm 11:** Multi-level EBM training using annealed MCMC samplers

---

**Input:** Number of traing iterations $N \geq 1$, initial parameter $\varphi_0$, batch size $B \geq 1$, number of MCMC steps $L \geq 1$, predefined annealed MCMC sampler

**for** $n = 0, \ldots, N-1$ **do**

    ▷ Get $B$ samples from $p_k$ (in parallel, for $k = 0, \ldots, K$)

    $\{X_{k,b}^+\}_{b=1}^B \overset{\mathrm{iid}}{\sim} p_k$

    ▷ Get $B$ samples from each $p_k^{\varphi_n}$ using the annealed MCMC sampler

    $\{X_{k,b}^-\}_{b=1,k=0}^{B,K} = \mathrm{annealedMCMC}\left(\{p_k^{\varphi_n}\}_{k=0}^K, B\right)$

    ▷ Apply a stochastic gradient descent on $\varphi_n$

    Compute the MC estimator $\hat{g}_n$ of the gradient of the loss $\mathcal{L}^{\mathrm{ours}}(\varphi_n)$ defined in (49)

    $\hat{g}_n = B^{-1} \sum_{k=0}^K \left(\sum_{b=1}^B \nabla_\varphi E^{\varphi_n}\left(k, X_{k,b}^-\right) - \sum_{b=1}^B \nabla_\varphi E^{\varphi_n}\left(k, X_{k,b}^+\right)\right)$

    Update $\varphi_n$ to $\varphi_{n+1}$ with Adam optimizer based on $\hat{g}_n$

**Output:** Optimized parameter $\varphi_N$

---

**Multi-level EBM parameterization as GMM tilting.** To reduce the computational footprint of multi-level EBMs in the context of LRDS, we suggest to define our EBM model as a tilting of a learned GMM. Let $\bar{\varphi} = \{w_j, \mathbf{m}_j, \Sigma_j\}_{j=1}^J$ be the parameters learned by GMM-LRDS (Algorithm 3) to approximate $\hat{\pi}^{\mathrm{ref}}$ and denote by $p_t^{\bar{\varphi}}$ the $t$-th marginal density of the resulting reference process. Building on this, we define $p_t^\varphi$ as a tilting of $p_t^{\bar{\varphi}}$ by

$$\log p_t^\varphi(x) = \log p_t^{\bar{\varphi}}(x) - \mathrm{NN}^\varphi(t,x) \ ,$$

where $\mathrm{NN}^\varphi : [0,T] \times \mathbb{R}^d \to \mathbb{R}$ is a neural network with parameter $\varphi$. In this scenario, we design the initial $\varphi_0$ to ensure that $\mathrm{NN}^{\varphi_0} = 0$. In practice, in the case where the GMM shares very few similarities with $\hat{\pi}^{\mathrm{ref}}$ close to $t = 0$, we rather consider

$$\log p_t^\varphi(x) = \mathbf{1}_{t > t_{\mathrm{lim}}}(t) \times \log p_t^{\bar{\varphi}}(x) - \mathrm{NN}^\varphi(t,x) \ ,$$

where $t_{\mathrm{lim}} \in [0,T]$. For most of our numerical experiments, we found that $t^{\mathrm{lim}} = 0.2$ brought satisfying results.

## G  ADDITIONAL METRICS FOR VARIATIONAL DIFFUSION-BASED METHODS

When specifically using annealed VI methods, one may consider additional metrics to evaluate the performance of the corresponding samplers. For completeness purpose, we provide in this section the expressions of the sampling metrics presented by Blessing et al. (2024), for all discrete time variational diffusion-based methods presented in this paper: RDS (including PIS and DDS), DIS and CMCD. All of these metrics are implemented in our code.

Given a fixed level of time discretization $K \geq 1$, these performance criteria require the evaluation of a variational *importance weight function* $w : (\mathbb{R}^d)^{K+1} \to \mathbb{R}_+$, assessing the quality of the variational approximation, that is specific to each variational setting.

- For RDS (including PIS and DDS), we define

$$w(y_{0:K}) = \frac{\mathcal{Z}^{\mathrm{ref}}}{\mathcal{Z}} \frac{\mathrm{d}p_{0:K}^{\theta}(y_{0:K})}{\mathrm{d}p_{0:K}^{\star}(y_{0:K})} = \frac{\mathrm{d}p_{0:K}^{\theta}(y_{0:K})\gamma^{\mathrm{ref}}(y_K)}{\mathrm{d}p_{0:K}^{\mathrm{ref}}(y_{0:K})\gamma(y_K)} \,,$$

where $p^{\theta} \in \mathscr{P}^{(K+1)}$ and $p^{\mathrm{ref}} \in \mathscr{P}^{(K+1)}$ are respectively defined as discrete time approximations of the path measures $\mathbb{P}^{\theta}$ and $(\mathbb{P}^{\mathrm{ref}})^R$; see Appendix D.3 for more details.

- For DIS, we define

$$w(y_{0:K}) = \frac{1}{\mathcal{Z}} \frac{\mathrm{d}p_{0:K}^{\theta}(y_{0:K})}{\mathrm{d}p_{0:K}^{\star}(y_{0:K})} \,,$$

where $p^{\theta} \in \mathscr{P}^{(K+1)}$ and $p^{\star} \in \mathscr{P}^{(K+1)}$ are respectively defined as discrete time approximations of the path measures $\mathbb{P}^{\theta}$, induced by SDE (21), and $(\mathbb{P}^{\star})^R$; see Appendix D.4 for more details.

- For CMCD, we define

$$w(y_{0:K}) = \frac{1}{\mathcal{Z}} \frac{\mathrm{d}p_{0:K}^{\theta}(y_{0:K})}{\mathrm{d}q_{0:K}^{\theta}(y_{0:K})}$$

where $p^{\theta} \in \mathscr{P}^{(K+1)}$ and $q^{\theta} \in \mathscr{P}^{(K+1)}$ are respectively defined as discrete time approximations of the path measures $\mathbb{P}^{\theta}$, induced by SDE (23), and $(\mathbb{Q}^{\theta})^R$, where $\mathbb{Q}^{\theta}$ is induced by SDE (22); see Appendix D.5 for more details.

Note that, for each variational setting, $w$ does not depend on the normalization constant of the target distribution : in particular, $\mathcal{Z}$ simplifies in the DIS and CMCD expressions. This also stands for $\mathcal{Z}^{\mathrm{ref}}$ in the RDS setting.

### G.1  'REVERSE' PERFORMANCE CRITERIA

**Definition.**  Standard variational metrics, referred to as 'reverse' performance criteria by Blessing et al. (2024), are defined as expectations *with respect to the variational distribution*.

(a)  The **Evidence Lower Bound** (ELBO), expected to be maximized, defined by

$$\mathrm{ELBO} = -\mathbb{E}[\log w(Y_{0:K}^{\theta})] \,, \ Y_{0:K}^{\theta} \sim p_{0:K}^{\theta} \,.$$

For any RDS setting with intractable $\mathcal{Z}^{\mathrm{ref}}$, this criterion may assess the performance of the RDS sampler itself, but cannot be used for numerical comparison with different variational settings.

(b)  The **MC estimation of the normalizing constant** $\hat{\mathcal{Z}}_r$, expected to be close to $\mathcal{Z}$, defined by

$$\hat{\mathcal{Z}}_r = \mathbb{E}[w(Y_{0:K}^{\theta})] \,, \ Y_{0:K}^{\theta} \sim p_{0:K}^{\theta} \,.$$

For any RDS setting with intractable $\mathcal{Z}^{\mathrm{ref}}$, this criterion cannot be used.

(c)  The **normalized Explained Sum of Squares** (nESS$_r$), expected to be close to 1, defined by

$$\mathrm{nESS}_r = \frac{\mathbb{E}[w(Y_{0:K}^{\theta})]^2}{\mathbb{E}[w(Y_{0:K}^{\theta})^2]} \,, \ Y_{0:K}^{\theta} \sim p_{0:K}^{\theta} \,.$$

**Computation.**   Based on the results from Appendix D.2, we detail the simulation of $Y_{0:K}^\theta$ and the computation of $\log w(Y_{0:K}^\theta)$ for each setting.

- **RDS setting**: $Y_0^\theta \sim \pi^{\mathrm{base}}$, and for any $k \in \{0, \dots, K-1\}$,

$$Y_{k+1}^\theta = a_k Y_k^\theta + b_k \{s_{T-t_k}^{\mathrm{ref}}(Y_k^\theta) + g_{T-t_k}^\theta(Y_k^\theta)\} + \sqrt{c_k} Z_k \ , \ Z_k \sim \mathrm{N}(0, \mathrm{I}_d) \ ,$$

and the importance weight function is given by

$$\log w(Y_{0:K}^\theta) = \sum_{k=0}^{K-1} \frac{w_k}{2} \left\| g_{T-t_k}^\theta(Y_k^\theta) \right\|^2 + \sum_{k=0}^{K-1} \sqrt{w_k} g_{T-t_k}^\theta(Y_k^\theta)^\top Z_k + \log \frac{\gamma^{\mathrm{ref}}}{\gamma}(Y_K^\theta) \ ,$$

where $\{w_k, a_k, b_k, c_k\}_{k=0}^{K-1}$ depend on the noising scheme and the discretization setting.

- **DIS setting**: $Y_0^\theta \sim \pi^{\mathrm{base}}$, and for any $k \in \{0, \dots, K-1\}$,

$$Y_{k+1}^\theta = a_k Y_k^\theta + b_k s_{T-t_k}^\theta(Y_k^\theta) + \sqrt{c_k} Z_k \ , \ Z_k \sim \mathrm{N}(0, \mathrm{I}_d)$$

and the importance weight function is given by

$$\log w(Y_{0:K}^\theta) = \sum_{k=0}^{K-1} \frac{w_k}{2} \| s_{T-t_k}^\theta(Y_k^\theta) \|^2 + \sum_{k=0}^{K-1} \sqrt{w_k} s_{T-t_k}^\theta(Y_k^\theta)^\top Z_k + \log \frac{\pi^{\mathrm{base}}(Y_0^\theta)}{\gamma(Y_K^\theta)} \ ,$$

where $\{w_k, a_k, b_k, c_k\}_{k=0}^{K-1}$ come from the EI discretization of the VP noising scheme.

- **CMCD setting**: $Y_0^\theta \sim \pi^{\mathrm{base}}$, and for any $k \in \{0, \dots, K-1\}$,

$$Y_{k+1}^\theta = Y_k^\theta + \frac{\sigma^2 \delta_k}{2} \{\nabla \log \pi_{t_k}(Y_k^\theta) + h_{t_k}^\theta(Y_k^\theta)\} + \sigma \sqrt{\delta_k} Z_k \ , \ Z_k \sim \mathrm{N}(0, \mathrm{I}_d) \ ,$$

and the importance weight function is given by

$$\log w(Y_{0:K}^\theta) = \frac{1}{2} \sum_{k=0}^{K-1} w_k^{\mathrm{CMCD}} \left\| u_k^\theta \right\|^2 + \sum_{k=0}^{K-1} \sqrt{w_k^{\mathrm{CMCD}}} (u_k^\theta)^\top Z_k + \log \frac{\pi^{\mathrm{base}}(Y_0^\theta)}{\gamma(Y_K^\theta)} \ ,$$
$$\text{with } u_k^\theta = \nabla \log \pi_{t_k}(Y_k^\theta) + \nabla \log \pi_{t_{k+1}}(Y_{k+1}^\theta) + h_{t_k}^\theta(Y_k^\theta) - h_{t_{k+1}}^\theta(Y_{k+1}^\theta) \ .$$

### G.2   'Forward' performance criteria

**Definition.**   As observed by Blessing et al. (2024), the 'reverse' variational metrics presented above may not be able to quantify mode collapse. Hence, the authors propose novel variational metrics, referred to as 'forward' performance criteria, that are defined as expectations *with respect to the ground truth distribution*.

(a)  The **Evidence Upper Bound** (EUBO), expected to be minimized, defined by

$$\mathrm{EUBO} = \mathbb{E}[\log w(Y_{0:K})] \ , \ Y_{0:K} \sim p_{0:K}^\star \ .$$

For any RDS setting with intractable $\mathcal{Z}^{\mathrm{ref}}$, this criterion may assess the performance of the RDS sampler itself, but cannot be used for numerical comparison with different variational settings.

(b)  The **MC estimation of the normalizing constant** $\hat{\mathcal{Z}}_f$, expected to be close to $\mathcal{Z}$, defined by

$$\hat{\mathcal{Z}}_f = \left( \mathbb{E}[w(Y_{0:K})^{-1}] \right)^{-1} \ , \ Y_{0:K} \sim p_{0:K}^\star \ .$$

For any RDS setting with intractable $\mathcal{Z}^{\mathrm{ref}}$, this criterion cannot be used.

(c)  The **normalized Explained Sum of Squares** ($\mathrm{nESS}_f$), expected to be close to 1, defined by

$$\mathrm{nESS}_f = \left( \mathbb{E}[w(Y_{0:K})^{-1}] \mathbb{E}[w(Y_{0:K})] \right)^{-1} \ , \ Y_{0:K} \sim p_{0:K}^\star \ .$$

**Computation.** Based on the results from Appendix D.2, we detail the simulation of $Y_{0:K}$ and the computation of $\log w(Y_{0:K})$ for each setting.

- **RDS setting with EM scheme**: $Y_K \sim \pi$, and for any $k \in \{0, \ldots, K-1\}$,

$$Y_k = S_{k+1}(t_k)Y_{k+1} + S_{k+1}(t_k)\sigma_{k+1}(t_k)Z_k \ , \ Z_k \sim \mathrm{N}(0, \mathrm{I}_d) \ ,$$

where $S_k(t) = \exp\left(\int_{T-t_k}^{T-t} f(u)\mathrm{d}u\right)$, $\sigma_k^2(t) = \int_{T-t_k}^{T-t} \frac{\beta(u)}{S_k(u)^2}\mathrm{d}u$, and the importance weight function is given by

$$\log w(Y_{0:K}) = -\sum_{k=0}^{K-1} w_k^{\mathrm{EM}} g_{T-t_k}^\theta(Y_k)^\top \left\{ s_{T-t_k}^{\mathrm{ref}}(Y_k) + \frac{1}{2}g_{T-t_k}^\theta(Y_k) \right\}$$

$$-\sum_{k=0}^{K-1} \sigma_{k+1}(t_k)g_{T-t_k}^\theta(Y_k)^\top Z_k + \sum_{k=0}^{K-1} \left( \frac{1}{S_{k+1}(t_k)} - 1 + f(T-t_k)\delta_k \right) g_{T-t_k}^\theta(Y_k)^\top Y_k$$

$$+ \log \frac{\gamma^{\mathrm{ref}}}{\gamma}(Y_K) \ ,$$

as a consequence of Lemma 10.

- **RDS setting with EI scheme (PBM)**: $Y_K \sim \pi$, and for any $k \in \{0, \ldots, K-1\}$,

$$Y_k = \frac{Y_{k+1}}{a_k^{\mathrm{PBM}}} + \frac{\sqrt{c_k^{\mathrm{PBM}}}}{a_k^{\mathrm{PBM}}}Z_k \ , \ Z_k \sim \mathrm{N}(0, \mathrm{I}_d) \ ,$$

and the importance weight function is given by

$$\log w(Y_{0:K}) = -\sum_{k=0}^{K-1} w_k^{\mathrm{PBM}} g_{T-t_k}^\theta(Y_k)^\top \{ s_{T-t_k}^{\mathrm{ref}}(Y_k) + \frac{1}{2}g_{T-t_k}^\theta(Y_k) \}$$

$$-\sum_{k=0}^{K-1} \sqrt{w_k^{\mathrm{PBM}}} g_{T-t_k}^\theta(Y_k)^\top Z_k + \log \frac{\gamma^{\mathrm{ref}}}{\gamma}(Y_K) \ ,$$

as a consequence of Lemma 13.

- **RDS setting with EI scheme (VP)**: $Y_K \sim \pi$, and for any $k \in \{0, \ldots, K-1\}$,

$$Y_k = \frac{Y_{k+1}}{a_k^{\mathrm{VP}}} + \frac{\sqrt{c_k^{\mathrm{VP}}}}{a_k^{\mathrm{VP}}}Z_k \ , \ Z_k \sim \mathrm{N}(0, \mathrm{I}_d) \ ,$$

and the importance weight function is given by

$$\log w(Y_{0:K}) = -\sum_{k=0}^{K-1} w_k^{\mathrm{VP}} g_{T-t_k}^\theta(Y_k)^\top \{ s_{T-t_k}^{\mathrm{ref}}(Y_k) + \frac{1}{2}g_{T-t_k}^\theta(Y_k) \}$$

$$-\sum_{k=0}^{K-1} \sqrt{w_k^{\mathrm{VP}}} g_{T-t_k}^\theta(Y_k)^\top Z_k + \log \frac{\gamma^{\mathrm{ref}}}{\gamma}(Y_K) \ ,$$

as a consequence of Lemma 16.

- **DIS setting**: $Y_K \sim \pi$, and for any $k \in \{0, \ldots, K-1\}$,

$$Y_k = \frac{Y_{k+1}}{a_k^{\mathrm{VP}}} + \frac{\sqrt{c_k^{\mathrm{VP}}}}{a_k^{\mathrm{VP}}}Z_k \ , \ Z_k \sim \mathrm{N}(0, \mathrm{I}_d) \ ,$$

and the importance weight function is given by

$$\log w(Y_{0:K}) = -\frac{1}{2}\sum_{k=0}^{K-1} w_k^{\mathrm{VP}} \left\| s_{T-t_k}^\theta(Y_k) \right\|^2 - \sum_{k=0}^{K-1} \sqrt{w_k^{\mathrm{VP}}} s_{T-t_k}^\theta(Y_k)^\top Z_k + \log \frac{\pi^{\mathrm{base}}(Y_0)}{\gamma(Y_K)} \ ,$$

as a consequence of Lemma 19.

- **CMCD setting**: $Y_K \sim \pi$, and for any $k \in \{0, \dots, K-1\}$,

$$Y_k = Y_{k+1} + \frac{\sigma^2 \delta_k}{2}\{\nabla \log \pi_{t_{k+1}}(Y_{k+1}) - h^\theta_{t_{k+1}}(Y_{k+1})\} + \sigma\sqrt{\delta_k}Z_k \ , \ Z_k \sim \mathrm{N}(0, \mathrm{I}_d) \ ,$$

and the importance weight function is given by

$$\log w(Y_{0:K}) = -\frac{1}{2}\sum_{k=0}^{K-1} w_k^{\mathrm{CMCD}} \left\|u_k^\theta\right\|^2 - \sum_{k=0}^{K-1}\sqrt{w_k^{\mathrm{CMCD}}}(u_k^\theta)^\top Z_k + \log\frac{\pi^{\mathrm{base}}(Y_0)}{\gamma(Y_K)}$$

$$\text{with } u_k^\theta = \nabla\log\pi_{t_k}(Y_k) + \nabla\log\pi_{t_{k+1}}(Y_{k+1}) + h_{t_k}^\theta(Y_k) - h_{t_{k+1}}^\theta(Y_{k+1}) \ ,$$

as a consequence of Lemma 22.

# H IMPLEMENTATION DETAILS

## H.1 DETAILS ON TARGET DISTRIBUTIONS

**Gaussian mixture from Figure 2.** The distribution from Figure 2 is a 16-dimensional mixture between $\mathrm{N}(\mathbf{m}_1, \Sigma_1)$ and $\mathrm{N}(\mathbf{m}_2, \Sigma_2)$ with weights 0.7 and 0.3 respectively. Let $\mathbf{1}_d$ be the $d$-dimensional vector with all components equal to 1, then we have

$$\mathbf{m}_1 = -0.6 \times \mathbf{1}_{16}, \ \ \mathbf{m}_2 = +0.6 \times \mathbf{1}_{16} \ .$$

Moreover, denote $R_\theta$ a rotation matrix with angle $\theta$ between the first and last axes, then

$$\Sigma_1 = R_{\pi/4}\operatorname{diag}(10^{-2}, \dots, 10^{-2}, 10^{-1})R_{\pi/4}^T, \ \ \Sigma_2 = R_{\pi/6}\operatorname{diag}(10^{-1}, 10^{-2}, \dots, 10^{-2})R_{\pi/6}^T \ .$$

In practice, we estimate the weight of the first mode $w_1$ by computing a Monte Carlo estimator of

$$\int \mathbf{1}_{\mathrm{N}(x;\mathbf{m}_1, \Sigma_1) > \mathrm{N}(x;\mathbf{m}_2, \Sigma_2)}(x)\pi(x)\mathrm{d}x \ . \tag{50}$$

**Bi-modal Gaussian mixture.** This Gaussian mixture is used in Figure 1 and also in Section 5. It is a $d$-dimensional mixture between $\mathrm{N}(-\mathbf{1}_d, \Sigma)$ and $\mathrm{N}(\mathbf{1}_d, \Sigma)$ with weights $2/3$ and $1/3$ respectively. The different values of $\Sigma$ are recapped in Table 3. Just like the previous experiment, we also use the formula (50) to estimate the strongest mode weight with Monte Carlo approach.

**Multi-modal Gaussian mixtures.** Given a number of mixture components $L > 2$, we define a $d$-dimensional Gaussian mixture $x \in \mathbb{R}^d \mapsto \sum_{\ell=1}^{L} w_\ell \mathrm{N}(x; \mathbf{m}_\ell, 0.5\,\mathrm{I}_d)$, where the mean locations $\{\mathbf{m}_\ell\}_{\ell=1}^L$ are independently distributed according to $\mathrm{U}\left([-L, L]^d\right)$ and the mixture weights $\{w_\ell\}_{\ell=1}^L$ are strictly increasing, defined with a constant geometrical increment such that $w_L/w_1 = 3$. To assess the mode weight recovery in this multi-modal setting, we compute the total variation distance between the exact mode weight histogram and the mode weight histogram computed from Monte Carlo approximation.

**Rings distribution.** Let $T : \mathbb{R}^+ \times [0, 2\pi] \to \mathbb{R}^2$ be the transformation from polar to cartesian coordinates. Let $\theta \sim \mathrm{U}([0, 2\pi])$ and $R \sim \mathrm{N}(r, \sigma^2)$ with $r, \sigma > 0$, we say that $T(R, \theta)$ is distributed as a ring of radius $r$ and width $\sigma$. The Rings distribution is a mixture of rings with radiuses $r = 1, 3, 5$ and width $\sigma = 0.1$. The weights of each ring within the mixture are $1/9, 3/9$ and $5/9$. In this specific case, we use more MCMC chains and GMM components than the number of modes. The initial points of the MCMC samplers are fetched by drawing the same amount of uniformly distributed points on each ring.

**Checkerboard distribution.** We divide the square $[-4, 4]^2$ into 16 squares of size $2 \times 2$. This distribution is defined as a mixture of uniform distributions on those squares in an interleaving fashion. We assign a weight of $18.75\%$ to each 4 squares on the left and $6.25\%$ to each 4 squares on the right. The initial points of the MCMC samplers are taken as the middle of each square.

Table 3: **Covariance settings for the bi-modal Gaussian mixture**. We denote by $\mathrm{logdiag}(a, b)$ with $0 < a < b$ the diagonal matrix in $\mathbb{R}^{d \times d}$ whose diagonal is a log-linear interpolation between $a$ and $b$. Moreover, $Q_d \in \mathbb{R}^{d \times d}$ is a random orthogonal matrix built by doing the QR decomposition of a random matrix distributed according to $\mathrm{U}([0, 5]^{d \times d})$. The seed used to generate this last matrix is fixed at $42$.

| Type | Difficulty | Cond. number | Covariance |
|------|-----------|--------------|-----------|
| Diagonal | Isotropic | $1$ | $(5 \times 10^{-2})^2 \times \mathrm{I}_d$ |
| Diagonal | Medium | $10^2$ | $(5 \times 10^{-2})^2 \times \mathrm{logdiag}(10^{-2}, 1)$ |
| Diagonal | Hard | $10^4$ | $(5 \times 10^{-2})^2 \times \mathrm{logdiag}(10^{-4}, 1)$ |
| Full | Medium | $10^2$ | $(5 \times 10^{-2})^2 \times Q_d \times \mathrm{logdiag}(10^{-2}, 1) \times Q_d^T$ |
| Full | Hard | $10^4$ | $(5 \times 10^{-2})^2 \times Q_d \times \mathrm{logdiag}(10^{-4}, 1) \times Q_d^T$ |

**The $\phi^4$ field system.** The $\phi^4$ model is a simplified framework often used as a continuous version of the Ising model, aiding in the exploration of phase transitions within statistical mechanics. As per Gabrié et al. (2022), we focus on a discretized version of this model, set on a one-dimensional grid with size $d = 32$. Each configuration is represented as a $d$-dimensional vector $(\phi_i)_{i=1}^d$. To further constrain the system, we fix the field to zero at both ends by setting $\phi_0 = \phi_{d+1} = 0$.

The negative log-density of the distribution is then expressed as follows

$$\ln \pi_h(\phi) = -\beta \left( \frac{ad}{2} \sum_{i=1}^{d+1} (\phi_i - \phi_{i-1})^2 + \frac{1}{4ad} \sum_{i=1}^{d} (1 - \phi_i^2)^2 + h\phi_i \right) . \tag{51}$$

We selected parameter values that make the system bimodal, setting $a = 0.1$ and the inverse temperature $\beta = 20$, while adjusting the value of $h$. We define $w_+$ as the statistical frequency of configurations where $\phi_{d/2} > 0$, and $w_-$ as the frequency where $\phi_{d/2} < 0$. When $h = 0$, the system is symmetric under the transformation $\phi \to -\phi$, so we anticipate $w_+ = w_-$. However, for $h > 0$, the negative mode becomes more prevalent.

For large values of $d$, the relative likelihood of the modes can be approximated using Laplace expansions at the 0th and 2nd orders. Letting $\phi_+^h$ and $\phi_-^h$ represent the local maxima of (51), these approximations provide the following results respectively

$$\frac{w_-}{w_+} \approx \frac{\pi_h(\phi_-^h)}{\pi_h(\phi_+^h)} , \quad \frac{w_-}{w_+} \approx \frac{\pi_h(\phi_-^h) \times |\det H_h(\phi_-^h)|^{-1/2}}{\pi_h(\phi_+^h) \times |\det H_h(\phi_+^h)|^{-1/2}} , \tag{52}$$

where $H_h$ is the Hessian of the function $\phi \to \ln \pi_h(\phi)$. We compute the modes $\phi_+^h$ and $\phi_-^h$ using a gradient descent on the distribution's potential. Since we do not have access to ground truth samples in practice, we compare the Laplace approximations defined in (52), considered as our ground truth, to a Monte Carlo estimation of $w_-/w_+$.

**Bayesian logistic regression models.** Finally, we evaluate the performance of a Bayesian logistic model, defined for any pair $(x, y) \in \mathbb{R}^{\dim} \times \{0, 1\}$ by the likelihood $p(y|x; w, b) = \mathrm{Bernoulli}(y; \sigma(w^T x + b))$, where $w \in \mathbb{R}^{\dim}$ is a weight vector, $b \in \mathbb{R}$ is an intercept and $\sigma$ is the sigmoid function. Given a dataset $\mathcal{D} = \{(x_j, y_j)\}_{j=1}^M \subset \mathbb{R}^{\dim} \times \{0, 1\}$ of size $M$, we aim to sample from the posterior distribution $p(w, b|\mathcal{D}) \propto p(w, b) \prod_{j=1}^M p(y_j|x_j; w, b)$ where $p(w, b) = p(w)p(b)$ is a fixed prior distribution. Following Blessing et al. (2024), we consider four real-world settings of binary classification problem: **Ionosphere** ($\dim = 35$, $M = 351$), **Sonar** ($\dim = 61$, $M = 208$), **German Credit** ($\dim = 25$, $M = 1000$), **Breast Cancer** ($\dim = 31$, $M = 569$). Each of these datasets is randomly split into a training subset $\mathcal{D}_{\mathrm{train}}$ of size $0.8M$ and a test subset $\mathcal{D}_{\mathrm{test}}$ of size $0.2M$. In this setting, we define the target distribution $\pi = p(w, b|\mathcal{D}_{\mathrm{train}})$ and evaluate the sampling quality by computing the (unnormalized) average predictive log-likelihood $\log p(w, b|\mathcal{D}_{\mathrm{test}})$, which is expected to be maximized.

Table 4: **Complexity of each algorithm during the overall sampling procedure**. We track the number of target evaluations and neural network evaluations. We consider that computing $\pi$ is as expensive as computing $\nabla \log \pi$. We assume an infinite parallel computational capabilities. Note that the cost of the training procedure can be amortized to sample multiple times.

| Method | Number of target evaluations | Number of neural network evaluations |
|---|---|---|
| SMC | $\mathcal{O}(KM)$ | 0 |
| RE | $\mathcal{O}(M)$ | 0 |
| PIS/DDS/DIS/CMCD | $\mathcal{O}(NK)$ | $\mathcal{O}(NK)$ |
| PDDS | $\mathcal{O}(A(N + KM))$ | $\mathcal{O}(A(N + KM))$ |
| iDEM | $\mathcal{O}(AN)$ | $\mathcal{O}(ANK)$ |
| LRDS | $\mathcal{O}(N)$ | $\mathcal{O}(NK)$ |

In each case, the prior is carefully chosen as a Gaussian distribution with the following parameters:

- **Ionosphere**: $p(w) = \mathrm{N}(0, 5.25\,\mathrm{I}_{34})$ and $p(b) = \mathrm{N}(4.25, 0.25^2)$,
- **Sonar**: $p(w) = \mathrm{N}(0, 4.5\,\mathrm{I}_{60})$ and $p(b) = \mathrm{N}(-2.5, 0.5^2)$,
- **German Credit**: $p(w) = \mathrm{N}(0, 1.25\,\mathrm{I}_{24})$ and $p(b) = \mathrm{N}(3.25, 0.5^5)$,
- **Breast Cancer**: $p(w) = \mathrm{N}(0, 3.75\,\mathrm{I}_{30})$ and $p(b) = \mathrm{N}(31, 2^2)$.

We draw the reader's attention to the fact that these posterior distributions don't exhibit explicit multi-modality features. Therefore, for EBM-LRDS, the tilting EBM parameterization is rather based on a Gaussian approximation of the target distribution than a GMM approximation. The initial point of the MCMC samplers are sampled from the prior distribution.

## H.2 COMPUTATIONAL COMPARISON OF THE PRESENTED METHODS

This section aims at comparing the computational budget of each algorithm. In particular, we track the number of neural network evaluations and the number of target evaluations. For each algorithm, we take the following notations.

- For SMC : $K$ (number of annealing steps), $M$ (number of MCMC steps per level)
- For RE : $M$ (total number of MCMC steps)
- For RDS (PIS/DDS/LRDS), DIS and CMCD : $N$ (number of training steps), $K$ (number of discretization steps)
- For PDDS : $N$ (number of training steps), $M$ (number of MCMC steps per level), $K$ (number of discretization steps), $A$ (number of adaptation steps)
- For iDEM : $N$ (number of training steps), $K$ (number of discretization steps), $A$ (number of adaptation steps)

We detail the complexities of each algorithm in Table 4. Note that, in LRDS, the complexity of MCMC sampling (to compute $\hat{\pi}^{\mathrm{ref}}$) or the complexity of learning the reference process were ignored as they are negligeable compared to the training and sampling budgets; see Appendix H.3 for details.

## H.3 HYPER-PARAMETERS OF EACH SAMPLING ALGORITHM

In this section, we detail every hyper-parameter for each algorithm involved in the computations of the results in Section 5. The computationally-related parameters were chosen to ensure a comparable execution clock-time between the different algorithms on the same hardware.

**Construction of the MCMC dataset for $\hat{\pi}^{\mathrm{ref}}$.** We build the dataset of reference samples by running MCMC samplers initialized in the modes of the target distribution. We run 4 chains per mode. We use the Metropolis-Adjusted Langevin algorithm (MALA), see Algorithm 5, as default sampler, except for the Checkerboard distribution where we use the Random Walk Metropolis Hastings (RWMH) algorithm as the score is constant. In both cases, we adapt the step size automatically for 8192 warmup steps to aim at a $70\%$ acceptance rate. The datasets are 60000 samples long.

**Annealed MCMC methods (SMC, RE).** For both algorithms, we incorporate prior knowledge into the base distribution by taking $\pi^{\text{base}} = \text{N}(\mathbf{m}, \Sigma)$ where $\mathbf{m}$ and $\Sigma$ are computed using maximum likelihood on samples from $\hat{\pi}^{\text{ref}}$. In each case, the annealed density path is a linearly-spaced geometric interpolation, where $\pi_k \propto (\pi^{\text{base}})^{(K-k)/K}\pi^{k/K}$. For SMC (see Algorithm 7), we use $K = 128$ annealing levels with $N = 1024$ particles and $L = 64$ MCMC steps per level (with a 4096 steps warmup) and set the ESS resampling threshold as $\alpha = 1$. For RE (see Algorithm 9), we use $K = 128$ annealing levels with 256 independent chains per level and a swapping frequency every $S = 8$ steps for a total of $M = 32$ steps (with a 8192 steps warmup). Both algorithms use MALA as MCMC backbone with a step-size automatically tuned to achieve 70% acceptance rate.

**Variational diffusion-based methods (PIS, DDS, DIS, CMCD, LRDS).** For PIS and DDS we use the implementation provided by Richter et al. (2023), while we use the implementation provided by Blessing et al. (2024) for DIS and CMCD. For PIS, DDS and DIS, we set the hyper-parameter $\sigma$ as advised by Appendix I.2 by computing the mean and the variance of the samples from $\hat{\pi}^{\text{ref}}$ to add prior knowledge. For CMCD, we sample a version of $\pi$ which is shifted by $\mathbf{m}$, defined as the mean computed on samples from $\hat{\pi}^{\text{ref}}$, and we set $\sigma$ as the square root of the average variance estimated on those samples. For PIS, DIS and DDS, we use $K = 100$ linearly spaced time discretization steps while CMCD learns its own time discretization grid of size $K = 128$ with a final time set at $T = 10$.

For all of them, we perform 4096 optimization steps with a batch of size 2048. The neural network at stake is a Fourier MLP, as in Zhang & Chen (2022), with 4 layers of width 64. In the case of PIS, DDS and DIS, we use a target-informed parameterization by adding $\text{NN}(t) \times \nabla \log \pi(x)$ (where NN is a time-dependent scalar neural network) to the Fourier MLP, as suggested by the respective authors. As recommended by Vargas et al. (2024), we do not consider this extra-parameterization in CMCD, since the drift of the generative process is already informed by $\pi$. We highlight that, by default, we do not use this target-informed parameterization in LRDS, since the reference process is specifically designed to be it-self target-informed, hence avoiding useless evaluations of the target score. Additionally, as recommended by Zhang & Chen (2022), we design the LRDS guidance network such that $g^{\theta_0} = 0$. This ensures that the very first sampling phase is driven solely using the reference process.

For all diffusion-based algorithms and target distributions, we ensured that the noising schedules were chosen to ensure that the target distribution gets successfully noised at time $T$. For LRDS, we use an exponential integration of the respective time-reversed SDE as default transition kernel; see Appendix D.3.2 and Appendix D.3.3.

**Adaptive diffusion-based methods (iDEM, PDDS).** For iDEM, we use the implementation provided by the original paper (Akhound-Sadegh et al., 2024). In order to provide prior information leveraging $\hat{\pi}^{\text{ref}}$, (i) we standardize the target distribution by using the empirical mean and variance of $\hat{\pi}^{\text{ref}}$ (ii) we preload the buffer with samples from $\hat{\pi}^{\text{ref}}$. Following the design choices of the original paper, we use a Variance Exploding noising scheme with a geometric variance schedule $\sigma_t$. Because we standardized the target distribution, we decided to take $\sigma_T = 5$ for each target distribution to ensure that the distribution is properly noised. We used the same hyper-parameters as in their Gaussian mixture experiment with 400 epochs instead of 1000 to ensure computational fairness. For PDDS, we also use the implementation provided by the original paper (Phillips et al., 2024). In order to provide prior information leveraging $\hat{\pi}^{\text{ref}}$, we standardized the target distribution by leveraging the empirical mean and variance of $\hat{\pi}^{\text{ref}}$ thus bypassing the original VI step of their algorithm. We also significantly increased the default computational budget by using 128 discretization steps, 2048 particles and training for 100000 steps with a batch size of 512. The rest of the hyper-parameters are the default ones.

**LRDS reference fitting details.** For GMM-LRDS, the EM algorithm is taken from Pedregosa et al. (2011). We use a diagonal covariance for Rings distribution, Checkerboard distribution, diagonal Gaussian mixtures and Bayesian posterior distributions, and use a full covariance otherwise. In the full covariance case, we regularize the covariance to ensure their positivity. For EBM-RDS, we fit multi-level EBMs with Algorithm 11 using the Replica Exchange annealed sampler (see Algorithm 9) as backbone. This sampler has the advantage of being massively parallel and thus suited for GPU computations. We perform 200 epochs with a batch size of 32 for each noise level. To increase gradient accuracy and enhance efficiency, we accumulate the gradients over 10 steps where

we keep the same negative samples but update the positive ones. We perform 32 MCMC steps to sample a batch of negative samples with a swap happening every 8 steps. We keep 16 MCMC steps out of 32 to compute the expectations. To compensate the short length of our MCMC chains, we leverage a persistent buffer to kickstart the chains at each noise level. Lastly, we utilise the GMM tilting EBM parameterization detailed at the end of Appendix F where the neural network parameterized as $x \mapsto \text{NN}(t, x)^T x$ as suggested by Salimans & Ho (2021). We leverage the GMM tilting initialization of the EBM to perform exact MCMC sampling at the very first gradient step. Moreover, we preconditioned the network as advised by Karras et al. (2022) by leveraging the Gaussian approximation of $\hat{\pi}^{\text{ref}}$.

# I ADDITIONAL RESULTS

## I.1 LINK BETWEEN THE TARGET AND THE REFERENCE PROCESSES

Let $\pi^{\text{ref}} \in \mathscr{P}(\mathbb{R}^d)$ be an arbitrary distribution. We recall that under mild assumptions on $\pi^{\text{ref}}$, see e.g., Cattiaux et al. (2023), the time-reversal of $\mathbb{P}^{\text{ref}}$ is associated to the SDE

$$\mathrm{d}Y_t^{\text{ref}} = -f(T-t)Y_t^{\text{ref}}\mathrm{d}t + \beta(T-t)s_{T-t}^{\text{ref}}(Y_t^{\text{ref}})\mathrm{d}t + \sqrt{\beta(T-t)}\mathrm{d}B_t , \ Y_0^{\text{ref}} \sim \mathbb{P}_T^{\text{ref}} . \quad (53)$$

**Link with Schrödinger Bridge.** Since $\mathbb{P}^\star$ and $\mathbb{P}^{\text{ref}}$ are based on the same noising diffusion process, these path measures are linked by the relation $\mathbb{P}^\star = \pi \otimes \mathbb{P}_{|0}^{\text{ref}}$. Hence, using the KL chain rule (Léonard, 2014), it is clear that $\mathbb{P}^\star$ is solution to the following minimization problem over path measure space

$$\arg\min\{\text{KL}(\mathbb{P} \mid \mathbb{P}^{\text{ref}}) \ : \ \mathbb{P} \in \mathscr{P}(\mathbf{C}_T), \mathbb{P}_0 = \pi\} ,$$

which is often referred to as a half-Schrödinger bridge problem in stochastic control literature.

**Link with Doob's h-transform.** This relation on path measure space can also be written as $\mathbb{P}^\star = \frac{\mathrm{d}\pi}{\mathrm{d}\pi^{\text{ref}}} \cdot \mathbb{P}^{\text{ref}}$, see Appendix D.1. Therefore, solely based on the SDE (53) describing $(\mathbb{P}^{\text{ref}})^R$, $(\mathbb{P}^\star)^R$ may be expressed as a Doob's h-transform of $(\mathbb{P}^{\text{ref}})^R$ via the SDE

$$\mathrm{d}Y_t = -f(T-t)Y_t\mathrm{d}t + \beta(T-t)\{s_{T-t}^{\text{ref}}(Y_t) + h_{T-t}(Y_t)\}\mathrm{d}t + \sqrt{\beta(T-t)}\mathrm{d}B_t , Y_0 \sim \mathbb{P}_T^\star ,$$

where the Doob's control function $h : [0,T] \times \mathbb{R}^d \to \mathbb{R}^d$ is defined, for any $y_t \in \mathbb{R}^d$ and any $t \in [0,T]$, by

$$h_{T-t}(y_t) = \nabla \log \mathbb{E}_{(\mathbb{P}^{\text{ref}})_{T|t}^R}\left[\frac{\mathrm{d}\pi}{\mathrm{d}\pi^{\text{ref}}}(Y_T) \mid Y_t = y_t\right] = \nabla \log \mathbb{E}_{\mathbb{P}_{0|T-t}^{\text{ref}}}\left[\frac{\mathrm{d}\pi}{\mathrm{d}\pi^{\text{ref}}}(X_0) \mid X_t = y_t\right] .$$

Additionally, we have for any $x_t \in \mathbb{R}^d$,

$$\frac{p_t^\star}{p_t^{\text{ref}}}(x_t) = \frac{1}{p_t^{\text{ref}}(x_t)}\int_{\mathbb{R}^d} p_{t|0}^\star(x_t|x_0)\mathrm{d}\pi(x_0)$$

$$= \int_{\mathbb{R}^d} \frac{\mathrm{d}\pi}{\mathrm{d}\pi^{\text{ref}}}(x_0)\frac{p_{t|0}^{\text{ref}}(x_t|x_0)\mathrm{d}\pi^{\text{ref}}(x_0)}{p_t^{\text{ref}}(x_t)}$$

$$= \mathbb{E}_{\mathbb{P}_{0|t}^{\text{ref}}}\left[\frac{\mathrm{d}\pi}{\mathrm{d}\pi^{\text{ref}}}(X_0) \mid X_t = x_t\right] .$$

By combining previous computations, we obtain $h_t = \nabla \log p_t^\star/p_t^{\text{ref}} = g_t$ for any $t \in [0,T]$.

## I.2 OPTIMAL SETTING OF ISOTROPIC GAUSSIAN REFERENCE DISTRIBUTION

The goal of this section is to explain how to set the hyperparameter $\sigma \in (0,\infty)$ in PIS and DDS settings so as to 'optimize' the reference distribution $\pi^{\text{ref}} = \text{N}(0, \sigma^2 \text{I}_d)$, when targeting a multi-modal distribution. The same reasoning can also be applied in the DIS setting to set the base distribution $\pi^{\text{base}}$, chosen as an isotropic Gaussian distribution.

Let $\pi \in \mathscr{P}(\mathbb{R}^d)$ be our target distribution. Assume that we are given a *diagonal Gaussian approximation* $\tilde{\pi} = \text{N}(\mathbf{m}, \Sigma)$ of $\pi$, with mean $\mathbf{m} \in \mathbb{R}^d$ and covariance $\Sigma = \text{diag}(\Gamma^2) \in \mathbb{R}^{d \times d}$, where

$\Gamma \in (0, \infty)^d$. We propose to set $\pi^{\mathrm{ref}}$ as 'close' as possible to $\tilde{\pi}$ in PIS and DDS by solving the following variational problem

$$\arg\min\{\mathrm{KL}(\tilde{\pi} \mid \mathrm{N}(0, \sigma^2\,\mathrm{I}_d)) \,:\, \sigma \in (0, \infty)\}\,.$$

Let $\sigma > 0$. We have

$$\mathrm{KL}(\tilde{\pi} \mid \mathrm{N}(0, \sigma^2\mathrm{I}_d)) = \frac{1}{2}\left[\log\frac{|\sigma^2\mathrm{I}_d|}{|\Sigma|} - d + \sigma^{-2}\mathbf{m}^T\mathbf{m} + \mathrm{Tr}(\sigma^{-2}\Sigma)\right]$$

$$= \frac{1}{2}\left[d\log(\sigma^2) - \sum_{j=1}^{d}\log\Gamma_j^2 - d + \sigma^{-2}\mathbf{m}^T\mathbf{m} + \sigma^{-2}\sum_{j=1}^{d}\Gamma_j^2\right]\,.$$

Moreover, the derivative with respect to $\sigma^2$ is given by

$$\frac{\partial}{\partial\sigma^2}\mathrm{KL}(\tilde{\pi} \mid \mathrm{N}(0, \sigma^2\mathrm{I}_d)) = \frac{1}{2}\left[d\sigma^{-2} - \sigma^{-4}\left(\mathbf{m}^T\mathbf{m} + \sum_{j=1}^{d}\Gamma_j^2\right)\right]\,.$$

In particular, the optimal solution of this variational problem is

$$\sigma_\pi^2 = d^{-1}\left(\mathbf{m}^T\mathbf{m} + \sum_{j=1}^{d}\Gamma_j^2\right)\,,$$

which motivates us to set $\pi^{\mathrm{ref}} = \mathrm{N}(0, \sigma_\pi^2\,\mathrm{I}_d)$.

In practice, when we do not have samples from the target distribution, we rather rely on a diagonal Gaussian approximation of the empirical distribution $\hat{\pi}^{\mathrm{ref}}$ (obtained with local MCMC samplers) and set $\pi^{\mathrm{ref}} = \mathrm{N}(0, \sigma_{\hat{\pi}^{\mathrm{ref}}}^2\,\mathrm{I}_d)$.

**Application to Gaussian mixtures.** In the particular case where $\pi$ is a Gaussian mixture, we have an analytical formula for $\tilde{\pi}$.

Let $\pi = \sum_{j=1}^{J}w_j\mathcal{N}(\mathbf{m}_j, \Sigma_j)$ be our target distribution with $w_{1:J} \in \Delta_J$, where the $j$-th component has mean $\mathbf{m}_j \in \mathbb{R}^d$ and covariance $\Sigma_j = \mathrm{diag}(\lambda_j) \in \mathbb{R}^{d\times d}$ with $\lambda_j \in (0, \infty)^d$. Consider $X \sim \pi$ and $Z \sim \mathcal{M}(w_1, \ldots, w_J)$. We first have

$$\mathbb{E}[X] = \mathbb{E}\left[\mathbb{E}\left[X \mid Z\right]\right] = \sum_{j=1}^{J}w_j\mathbf{m}_j\,.$$

We define the *diagonal variance* of $\pi$ by $\mathrm{diagVar}(\pi) = (\mathrm{Var}[X]_1, \ldots, \mathrm{Var}[X]_d) \in (0, \infty)^d$ where $\mathrm{Var}[X]_i = \mathbb{E}[(X_i - \mathbb{E}[X]_i)^2]$ for any $i \in \{1, \ldots, d\}$. By the law of total variance, we have

$$\mathrm{Var}[X]_i = \mathbb{E}\left[\mathrm{Var}\left[X \mid Z\right]_i\right] + \mathrm{Var}\left[\mathbb{E}\left[X \mid Z\right]\right]_i$$

$$= \sum_{j=1}^{J}w_j(\lambda_j^2)_i + \sum_{j=1}^{J}w_j\left((\mathbf{m}_j)_i - \mathbb{E}\left[X\right]_i\right)^2\,.$$

By characterisation of Gaussian distributions, $\mathrm{N}(\mathbb{E}[X], \mathrm{diagVar}(\pi))$ is the closest diagonal Gaussian distribution to $\pi$ in the Kullback-Leibler sense. Therefore, in this particular setting, $\tilde{\pi}$ may be defined by $\mathbf{m} = \mathbb{E}[X]$ and $\Gamma^2 = \mathrm{diagVar}(\pi)$. We use this approach when computing $\sigma_\pi^2$ in the numerical experiment commented in Figure 1.

## I.3 FAILURE OF LOCAL MCMC SAMPLERS ON MULTIMODAL DISTRIBUTIONS

As we show in Figure 7, *local* MCMC samplers such as the *Random Walk Metropolis Hastings* (RWMH) algorithm (Metropolis et al., 1953), the *Metropolis-adjusted Langevin Algorithm* (MALA) (Roberts & Tweedie, 1996), the *Hamiltonian Monte Carlo* (HMC) algorithm (Duane et al., 1987; Brooks et al., 2011) or the *No-U Turn Sampler* (NUTS) (Hoffman & Gelman, 2014) tend to produce Markov chains that get trapped in modes. The resulting samples are thus not representative of the global landscape.

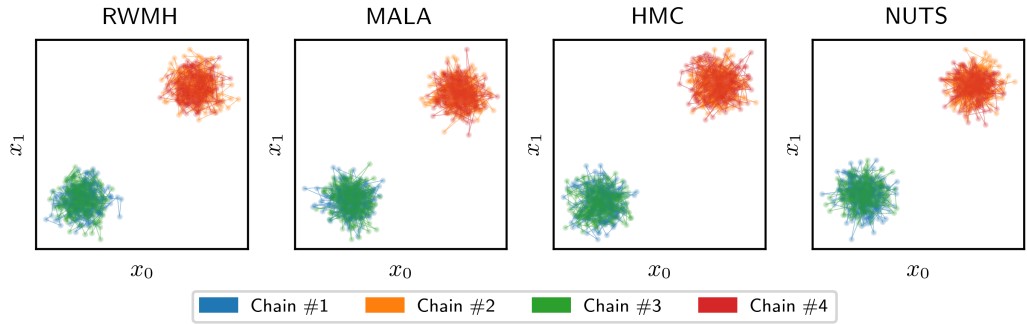

Figure 7: Samples from different MCMC samplers when sampling from a bi-modal Gaussian mixture in 2 dimensions - There are 4 different chains (in different colors). The MCMC samplers ran for 4096 warmup steps before producing those 1024 samples. We only display 1 sample every 4 steps.

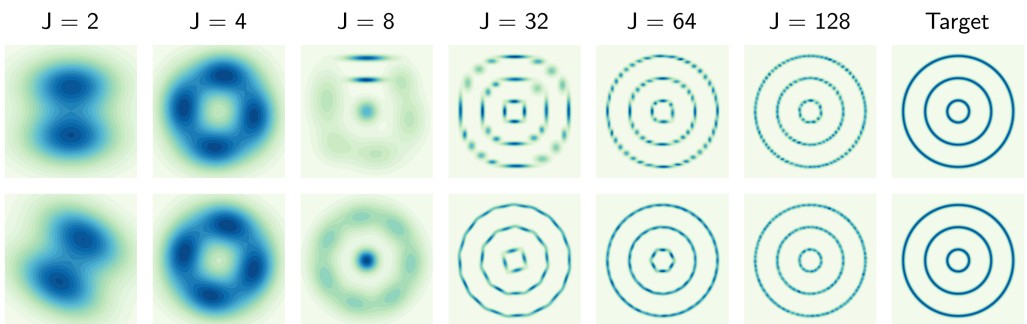

Figure 8: Increasingly expressive GMM approximations of the rings distribution - **(Top)** Approximation using diagonal covariances **(Bottom)** Approximation using full covariances

## I.4 LIMITATIONS OF GAUSSIAN MIXTURE MODELS

Figure 8 shows the progressive GMM approximation of the Rings distribution by leveraging an increasing number of components. This figure shows that, in this case, getting a good approximation of the distribution requires significantly more components that the number of modes (here, 3). This highlights the limitation of the approximation power of Gaussian Mixture models.

## I.5 FURTHER EXPERIMENTAL RESULTS

In this section, we display detailed results for the experimental settings presented in Section 5, as well as additional results that assess the robustness and the superiority of LRDS compared to its competitors for a large diversity of settings. When ground truth samples are available (i.e., in all settings except $\phi^4$ and Bayesian regression), we display statistical estimations of the integral probability metrics defined in Appendix B, namely, the regularized Wasserstein distance with regularization $\varepsilon = 10^{-3}$ ($W_{2,\varepsilon}$), the Maximum Mean Discrepancy (MMD) and the sliced Kolmogorov-Smirnov distance (KS).

### I.5.1 HIGH-DIMENSIONAL BI-MODAL GAUSSIAN MIXTURES

Below, we display the results of all considered samplers, including LRDS, when targeting the bi-modal Gaussian mixtures defined in Appendix H.1 with increasing dimension $d \in \{8, 16, 32, 64\}$. To assess the sampling quality at a global level, we compute for each sampler the absolute error 'Mode Err.' when estimating the strongest mode weight (in %, comprised between $0\%$, the best, and $66.7\%$, the worst) via Monte Carlo approximation. All metrics are computed based on $8192$ samples. In particular,

Table 5: Results for bi-modal Gaussian mixtures with **diagonal** covariance and **medium** conditioning, averaged over 16 sampling runs (same setting as Table 2). Bold font indicates best result, orange cells refer to settings with uninformative mode weight estimation (i.e., uniform mixture), red cells denote mode collapse. N/A denotes settings where results could not be obtained due to numerical issues. The MMD and KS results are displayed with 100-factor rescaling.

| Algorithm | $d=16$ | | | | $d=32$ | | | | $d=64$ | | | |
|---|---|---|---|---|---|---|---|---|---|---|---|---|
| | Mode Err. ↓ | $W_{2,\varepsilon}$ ↓ | MMD ↓ | KS ↓ | Mode Err. ↓ | $W_{2,\varepsilon}$ ↓ | MMD ↓ | KS ↓ | Mode Err. ↓ | $W_{2,\varepsilon}$ ↓ | MMD ↓ | KS ↓ |
| SMC | 11.4%±9.1% | 0.38±0.00 | 13.91±10.68 | 10.80±7.84 | 15.8%±8.5% | 0.70±0.01 | 18.68±9.67 | 14.63±7.59 | 15.2%±7.5% | 1.16±0.01 | 17.87±8.72 | 14.14±6.71 |
| RE | 16.5%±1.3% | 0.38±0.00 | 20.57±1.78 | 15.20±1.31 | 15.9%±1.4% | 0.70±0.00 | 19.23±2.03 | 14.54±1.55 | 17.0%±1.4% | 1.17±0.00 | 20.36±1.83 | 15.68±1.39 |
| LV-PIS | 6.0%±3.4% | 0.43±0.00 | 7.63±3.07 | 6.86±2.49 | 33.2%±0.1% | 1.01±0.03 | 34.90±0.58 | 28.43±1.52 | 33.0%±0.1% | 1.62±0.02 | 34.03±0.71 | 29.33±0.75 |
| LV-DDS | 11.8%±9.3% | 0.40±0.01 | 13.83±10.23 | 11.29±8.06 | 31.5%±2.9% | 0.86±0.04 | 33.96±3.29 | 28.00±2.65 | 33.1%±0.1% | 1.49±0.02 | 34.45±0.61 | 28.95±0.89 |
| LV-DIS | 16.4%±0.50% | 0.42±0.00 | 21.36±0.75 | 16.16±0.54 | 16.5%±0.40% | 0.77±0.00 | 21.13±0.96 | 16.18±0.76 | 16.8%±0.50% | 1.24±0.01 | 20.71±0.81 | 15.93±0.56 |
| LV-CMCD | 32.2%±15.4% | 0.60±0.06 | 46.89±17.35 | 33.46±13.03 | 50.1%±8.8% | 1.96±0.52 | 76.17±13.71 | 48.79±8.03 | 16.3%±10.6% | 3.47±0.31 | 53.15±1.85 | 41.79±2.64 |
| iDEM | 33.3%±0.0% | 1.75±0.17 | 53.82±0.36 | 36.86±0.92 | 66.7%±0.0% | 4.16±0.22 | 85.52±0.53 | 61.10±1.26 | 11.7%±0.4% | 117.82±0.14 | 90.13±0.12 | N/A |
| PDDS | **0.8%±0.6%** | 0.40±0.00 | **1.66±0.68** | **2.59±0.29** | 66.7%±0.0% | 11.22±0.08 | 105.11±0.24 | N/A | N/A | N/A | N/A | N/A |
| GMM-LRDS | 1.7%±0.6% | 0.38±0.00 | 2.58±0.96 | 2.69±0.66 | **2.7%±0.8%** | 0.71±0.00 | **3.64±1.07** | **3.38±0.76** | **4.1%±0.6%** | 1.19±0.00 | **5.25±0.98** | **4.50±0.70** |

Table 6: Results for bi-modal Gaussian mixtures with **diagonal** covariance and **isotropic** conditioning, averaged over 16 sampling runs. Bold font indicates best result, orange cells refer to settings with uninformative mode weight estimation (i.e., uniform mixture), red cells denote mode collapse. N/A denotes settings where results could not be obtained due to numerical issues. The MMD and KS results are displayed with 100-factor rescaling.

| Algorithm | $d=16$ | | | | $d=32$ | | | | $d=64$ | | | |
|---|---|---|---|---|---|---|---|---|---|---|---|---|
| | Mode Err. ↓ | $W_{2,\varepsilon}$ ↓ | MMD ↓ | KS ↓ | Mode Err. ↓ | $W_{2,\varepsilon}$ ↓ | MMD ↓ | KS ↓ | Mode Err. ↓ | $W_{2,\varepsilon}$ ↓ | MMD ↓ | KS ↓ |
| SMC | 16.0%±10.1% | 0.65±0.01 | 19.11±12.55 | 14.06±8.74 | 12.3%±9.6% | 1.18±0.01 | 14.57±11.46 | 11.03±8.03 | 11.0%±8.8% | 1.94±0.00 | 12.98±10.51 | 9.80±7.38 |
| RE | 15.2%±1.2% | 0.66±0.00 | 18.72±1.66 | 13.42±1.23 | 16.1%±1.3% | 1.19±0.00 | 19.50±1.70 | 14.14±1.26 | 16.5%±1.3% | 1.94±0.00 | 19.46±1.71 | 14.39±1.24 |
| LV-PIS | 1.9%±1.2% | 0.66±0.00 | 2.66±1.45 | 2.63±0.89 | 1.3%±0.6% | 1.20±0.01 | 2.15±0.68 | 2.23±0.42 | 2.8%±0.6% | 1.98±0.00 | 2.94±0.90 | 2.81±0.61 |
| LV-DDS | **0.7%±0.5%** | 0.65±0.00 | **0.86±0.70** | **1.59±0.30** | **0.8%±0.5%** | 1.19±0.00 | **1.09±0.60** | **1.56±0.24** | 1.6%±0.8% | 1.95±0.00 | **2.13±0.97** | **2.23±0.61** |
| LV-DIS | 16.4%±1.3% | 0.66±0.00 | 19.99±1.73 | 14.16±1.21 | 16.7%±0.7% | 1.20±0.00 | 20.12±1.00 | 14.54±0.70 | 16.6%±0.6% | 1.96±0.00 | 19.79±0.90 | 14.59±0.65 |
| LV-CMCD | 0.90%±0.60% | 0.68±0.00 | 2.84±0.55 | 2.92±0.45 | 1.7%±0.7% | 1.24±0.00 | 2.77±0.36 | 2.76±0.36 | 1.1%±0.8% | 2.05±0.00 | 3.43±0.75 | 3.15±0.65 |
| iDEM | 66.7%±0.0% | 2.23±0.17 | 83.92±0.50 | 59.68±2.16 | 33.3%±0.0% | 2.48±0.15 | 49.83±0.35 | 35.02±0.62 | **0.3%±0.2%** | 179.86±0.12 | 97.17±1.05 | N/A |
| PDDS | 0.9%±0.6% | 0.70±0.00 | 1.27±1.25 | 2.50±0.49 | 0.9%±0.6% | 1.22±0.00 | 1.53±0.87 | 2.41±0.31 | 0.9%±0.7% | 1.96±0.00 | 0.85±1.25 | 2.05±0.56 |
| GMM-LRDS | 1.6%±0.6% | 0.66±0.00 | 2.07±0.99 | 2.23±0.54 | 2.8%±0.5% | 1.19±0.00 | 3.72±0.89 | 3.23±0.58 | 4.6%±0.6% | 1.96±0.00 | 5.65±0.88 | 4.56±0.61 |

Table 7: Results for bi-modal Gaussian mixtures with **diagonal** covariance and **hard** conditioning, averaged over 16 sampling runs. Bold font indicates best result, orange cells refer to settings with uninformative mode weight estimation (i.e., uniform mixture), red cells denote mode collapse. N/A denotes settings where results could not be obtained due to numerical issues. The MMD and KS results are displayed with 100-factor rescaling.

| Algorithm | $d=16$ | | | | $d=32$ | | | | $d=64$ | | | |
|---|---|---|---|---|---|---|---|---|---|---|---|---|
| | Mode Err. ↓ | $W_{2,\varepsilon}$ ↓ | MMD ↓ | KS ↓ | Mode Err. ↓ | $W_{2,\varepsilon}$ ↓ | MMD ↓ | KS ↓ | Mode Err. ↓ | $W_{2,\varepsilon}$ ↓ | MMD ↓ | KS ↓ |
| SMC | 12.4%±8.3% | 0.24±0.00 | 15.40±10.16 | 12.00±7.57 | 16.5%±9.5% | 0.47±0.00 | 20.22±11.66 | 15.71±8.58 | 11.4%±9.8% | 0.80±0.00 | 13.88±11.92 | 11.16±8.79 |
| RE | 16.3%±1.4% | 0.25±0.00 | 20.76±1.95 | 15.40±1.38 | 17.0%±1.7% | 0.47±0.00 | 21.31±2.19 | 16.11±1.58 | 16.4%±1.4% | 0.81±0.00 | 19.99±1.75 | 15.54±1.38 |
| LV-PIS | 8.4%±3.4% | 0.45±0.00 | 13.52±2.50 | 12.48±2.01 | 32.2%±0.3% | 1.29±0.02 | 33.89±0.60 | 29.92±0.57 | 32.5%±0.2% | 2.24±0.04 | 33.91±0.76 | 30.49±0.61 |
| LV-DDS | 24.7%±8.8% | 0.32±0.03 | 26.89±9.03 | 22.92±7.59 | 40.5%±13.9% | 0.73±0.02 | 44.97±19.24 | 37.20±12.88 | 38.1%±15.4% | 1.59±0.04 | 42.64±20.91 | 35.46±13.67 |
| LV-DIS | 16.6%±0.6% | 0.38±0.02 | 23.00±0.87 | 19.69±0.69 | 16.7%±0.7% | 0.65±0.03 | 23.01±1.12 | 19.33±0.99 | 16.8%±0.6% | 1.06±0.02 | 22.94±0.84 | 18.97±0.65 |
| LV-CMCD | 21.5%±17.8% | 4.37±1.19 | 62.31±7.46 | N/A | 31.5%±2.2% | 1.54±0.35 | 46.02±13.29 | 37.50±1.30 | 44.1%±15.3% | 0.87±0.03 | 51.17±19.36 | 41.30±14.11 |
| iDEM | 33.3%±0.0% | 2.12±0.11 | 51.43±0.33 | 38.07±1.02 | 66.7%±0.0% | 7.53±0.52 | 85.77±0.63 | 55.56±1.03 | 33.3%±0.0% | 175.16±0.83 | 102.13±0.20 | N/A |
| PDDS | N/A | N/A | N/A | N/A | N/A | N/A | N/A | N/A | N/A | N/A | N/A | N/A |
| GMM-LRDS | **2.1%±1.0%** | 0.25±0.00 | **3.41±1.26** | **3.46±0.90** | **1.7%±0.9%** | 0.47±0.00 | **3.00±1.12** | **2.99±0.80** | **4.5%±1.8%** | 0.82±0.00 | **6.12±2.21** | **5.25±1.62** |

- Table 5 corresponds to Gaussian mixtures with diagonal covariance and medium conditioning, completing the results of Table 2 presented in the main paper,

- Table 6 corresponds to Gaussian mixtures with diagonal covariance and isotropic conditioning,

- Table 7 corresponds to Gaussian mixtures with diagonal covariance and hard conditioning,

- Table 8 corresponds to Gaussian mixtures with full covariance and medium conditioning,

- Table 9 corresponds to Gaussian mixtures with full covariance and hard conditioning.

For all of these bi-modal settings, we observe that the mode weight estimation error is consistent with the values of probability metrics (except for the regularized Wasserstein distance, which is not discriminative between the methods). In particular, GMM-LRDS is on par or superior to all competitors in each setting, except the least challenging setting ('Isotropic' conditioning, see Table 6), where LV-DDS is more performant.

Table 8: Results for bi-modal Gaussian mixtures with **full** covariance and **medium** conditioning, averaged over 16 sampling runs. Bold font indicates best result, orange cells refer to settings with uninformative mode weight estimation (i.e., uniform mixture), red cells denote mode collapse. N/A denotes settings where results could not be obtained due to numerical issues. The MMD and KS results are displayed with 100-factor rescaling.

| Algorithm | $d = 8$ | | | | $d = 16$ | | | | $d = 32$ | | | |
|---|---|---|---|---|---|---|---|---|---|---|---|---|
| | Mode Err. ↓ | $W_{2,\varepsilon}$ ↓ | MMD ↓ | KS ↓ | Mode Err. ↓ | $W_{2,\varepsilon}$ ↓ | MMD ↓ | KS ↓ | Mode Err. ↓ | $W_{2,\varepsilon}$ ↓ | MMD ↓ | KS ↓ |
| SMC | 17.3%±7.1% | 0.17±0.00 | 21.55±8.84 | 16.07±6.33 | 13.5%±4.5% | 0.38±0.00 | 16.35±5.47 | 12.52±4.05 | 10.3%±5.8% | 0.70±0.00 | 12.18±6.67 | 9.72±5.15 |
| RE | 16.6%±1.0% | 0.17±0.00 | 21.34±1.54 | 15.30±1.11 | 16.4%±1.4% | 0.38±0.00 | 20.65±1.90 | 15.23±1.34 | 16.3%±1.2% | 0.70±0.00 | 19.90±1.74 | 15.04±1.24 |
| LV-PIS | 6.8%±2.4% | 0.20±0.00 | 9.89±2.78 | 8.67±1.98 | 25.4%±12.6% | 0.52±0.06 | 28.21±13.13 | 23.37±10.61 | 33.1%±0.2% | 1.07±0.39 | 35.02±0.70 | 27.85±3.09 |
| LV-DDS | 1.6%±0.8% | 0.17±0.00 | 2.64±0.87 | 2.63±0.53 | 3.3%±0.9% | 0.39±0.00 | 4.83±1.08 | 4.27±0.75 | 4.0%±1.1% | 0.75±0.00 | 6.85±1.23 | 5.95±1.01 |
| LV-DIS | 16.5%±0.5% | 0.22±0.00 | 22.50±0.88 | 17.33±0.66 | 16.3%±1.1% | 0.46±0.02 | 21.65±1.55 | 16.93±0.13 | 16.7%±0.6% | 0.82±0.01 | 21.95±0.94 | 17.16±0.72 |
| LV-CMCD | 16.6%±0.5% | 0.18±0.00 | 21.76±1.04 | 16.06±0.81 | 20.8%±12.7% | 1.23±0.96 | 36.92±5.06 | 26.70±4.59 | 16.6%±0.50% | 0.75±0.00 | 20.89±0.95 | 15.90±0.71 |
| iDEM | 66.7%±0.0% | 2.06±0.19 | 85.27±0.40 | 61.42±1.09 | 66.7%±0.0% | 3.95±0.20 | 84.48±0.53 | 62.98±1.10 | 66.7%±0.0% | 6.89±0.24 | 85.60±0.41 | 62.87±1.31 |
| PDDS | 1.3%±1.1% | 0.19±0.00 | **1.52**±1.58 | 2.65±0.77 | 1.9%±0.9% | 0.40±0.00 | **2.63**±1.53 | 3.09±0.77 | 0.7%±0.6% | 0.73±0.00 | **1.56**±0.89 | 2.61±0.37 |
| GMM-LRDS | **1.3%**±0.6% | 0.17±0.00 | 1.97±0.85 | **2.36**±0.52 | **1.8%**±0.5% | 0.38±0.00 | 2.65±0.87 | **2.72**±0.58 | 2.4%±0.5% | 0.71±0.00 | 3.48±0.89 | 3.24±0.61 |

Table 9: Results for bi-modal Gaussian mixtures with **full** covariance and **hard** conditioning, averaged over 16 sampling runs. Bold font indicates best result, orange cells refer to settings with uninformative mode weight estimation (i.e., uniform mixture), red cells denote mode collapse. N/A denotes settings where results could not be obtained due to numerical issues. The MMD and KS results are displayed with 100-factor rescaling.

| Algorithm | $d = 8$ | | | | $d = 16$ | | | | $d = 32$ | | | |
|---|---|---|---|---|---|---|---|---|---|---|---|---|
| | Mode Err. ↓ | $W_{2,\varepsilon}$ ↓ | MMD ↓ | KS ↓ | Mode Err. ↓ | $W_{2,\varepsilon}$ ↓ | MMD ↓ | KS ↓ | Mode Err. ↓ | $W_{2,\varepsilon}$ ↓ | MMD ↓ | KS ↓ |
| SMC | 14.2%±15.0% | 0.10±0.00 | 17.83±19.13 | 13.83±13.58 | 10.1%±6.9% | 0.24±0.00 | 12.91±8.77 | 10.03±6.21 | 17.9%±10.2% | 0.47±0.00 | 21.61±12.58 | 16.96±9.50 |
| RE | 16.1%±1.2% | 0.11±0.00 | 20.99±1.86 | 15.12±1.33 | 16.8%±1.2% | 0.25±0.00 | 21.54±1.76 | 15.89±1.22 | 16.3%±1.3% | 0.47±0.00 | 20.28±1.79 | 15.39±1.32 |
| LV-PIS | 10.9%±1.1% | 0.20±0.01 | 15.75±1.22 | 14.13±0.83 | 2.2%±1.5% | 0.43±0.00 | 10.36±0.80 | 10.50±0.70 | 32.7%±0.3% | 1.34±0.03 | 34.25±0.75 | 30.15±0.78 |
| LV-DDS | 3.2%±1.8% | 0.12±0.01 | 5.53±1.63 | 5.36±1.05 | 7.5%±4.1% | 0.29±0.00 | 11.39±4.88 | 9.91±3.53 | 10.5%±7.8% | 0.62±0.00 | 16.62±8.01 | 14.10±6.05 |
| LV-DIS | 16.9%±0.5% | 0.28±0.03 | 24.00±0.90 | 21.85±0.90 | 16.6%±0.6% | 0.51±0.03 | 23.85±0.96 | 22.1±0.77 | 16.8%±0.6% | 0.91±0.07 | 24.34±0.91 | 22.76±0.77 |
| LV-CMCD | 16.6%±0.5% | 1.98±0.02 | 18.76±0.56 | 34.42±1.16 | 16.7%±0.6% | 3.67±0.02 | 19.86±0.52 | 36.03±0.10 | 16.7%±0.5% | 6.05±0.02 | 22.65±0.42 | 35.88±0.99 |
| iDEM | 66.6%±0.0% | 0.23±0.00 | 84.95±0.61 | 60.96±1.39 | 66.7%±0.0% | 5.49±0.12 | 86.81±0.47 | 59.71±1.59 | 66.7%±0.0% | 10.33±0.09 | 81.57±0.65 | 53.73±1.36 |
| PDDS | N/A | N/A | N/A | N/A | N/A | N/A | N/A | N/A | N/A | N/A | N/A | N/A |
| GMM-LRDS | **1.8%**±0.7% | 0.11±0.00 | **2.85**±1.00 | **3.22**±0.67 | **1.0%**±0.7% | 0.25±0.00 | **2.22**±0.91 | **2.60**±0.62 | **2.2%**±1.3% | 0.47±0.00 | **3.19**±1.63 | **3.13**±1.17 |

**Execution time.** To demonstrate the applicability of LRDS in practice, we provide in Figure 9 the execution clock-time of each sampling method for several Gaussian mixture sampling settings. The computations were all ran on the same V100 GPU. We notice that (i) the initial MCMC sampling step to build the reference dataset, then used to initialize each sampler, and the reference fitting time from GMM-LRDS are completely negligeable compared to the training time, and that (ii) the training time of GMM-LRDS is on par with previous variational diffusion-based approaches.

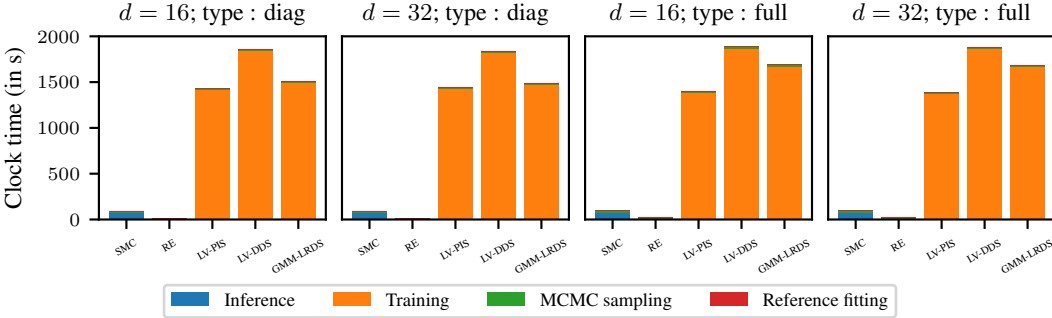

Figure 9: Execution clock-time for bi-modal Gaussian mixtures with **diagonal** covariance (left) and **full** covariance (right), averaged over the conditioning settings.

**Ablation study on GMM-LRDS: effect of the reference distribution.** To asses the robustness of GMM-LRDS with respect to the setting of the reference distribution, we conduct an ablation study that reveals the impact of modifying the location and the entropy of the reference modes. To do so, we set $d = 8$ and define the target distribution $\pi : x \in \mathbb{R}^d \mapsto (2/3)\mathrm{N}(x; -\mathbf{1}_d, 0.05) + (1/3)\mathrm{N}(x; \mathbf{1}_d, 0.05)$, as detailed in Appendix H.1. Instead of defining the reference distribution in GMM-LRDS as a Gaussian mixture fitted on MCMC samples, we propose to consider a flexible reference distribution given by the Gaussian mixture $x \in \mathbb{R}^d \mapsto (2/3)\mathrm{N}(x; \mathbf{m}_1, 0.05\alpha^2\mathrm{I}_d) + (1/3)\mathrm{N}(x; \mathbf{m}_2, 0.05\alpha^2\mathrm{I}_d)$ where $(\mathbf{m}_1, \mathbf{m}_2) \in \mathbb{R}^d \times \mathbb{R}^d$ and $\alpha > 0$.

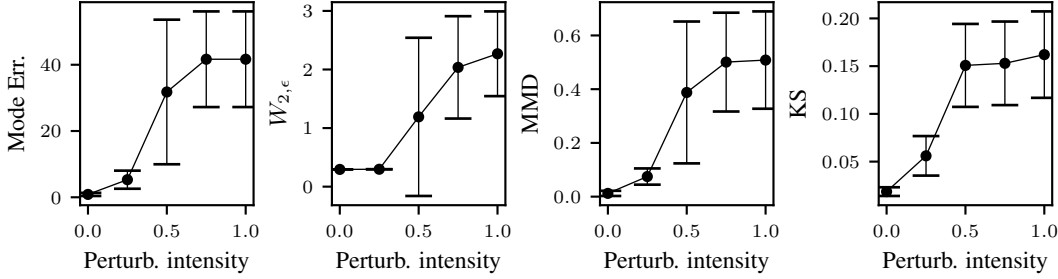

Figure 10: Results of GMM-LRDS for an 8-dimensional bi-modal Gaussian mixture when varying the location of the reference modes: the performance degrades as soon as the reference modes and the target modes are further from each other.

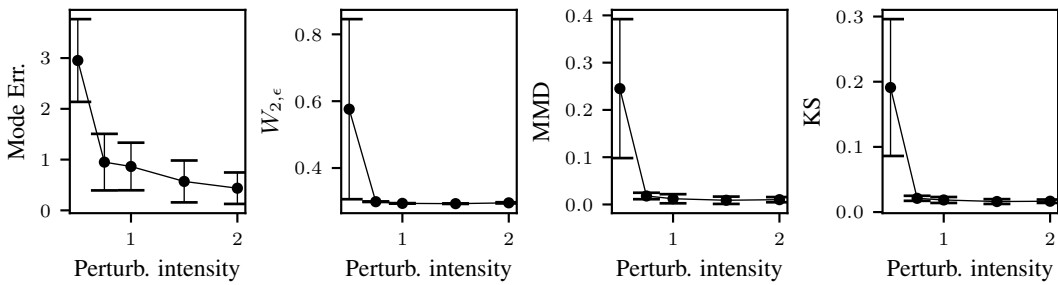

Figure 11: Results of GMM-LRDS for an 8-dimensional bi-modal Gaussian mixture when varying the variance of the reference modes: except for small reference mode variance, the performance is unchanged.

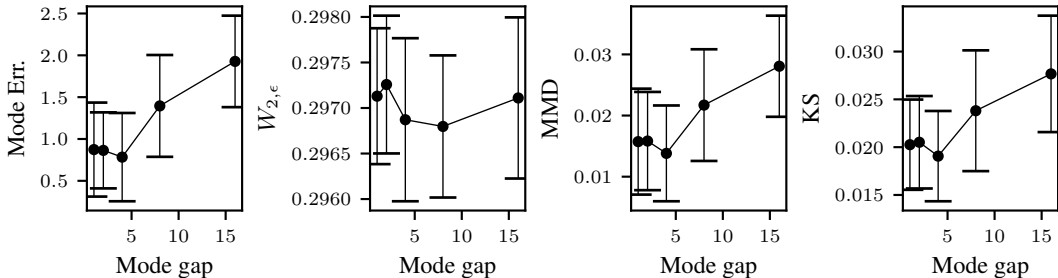

Figure 12: Results of GMM-LRDS for an 8-dimensional bi-modal Gaussian mixture when varying the distance between the modes of the target distribution: the performance does not significantly degrades, despite the increasing sampling complexity.

To observe the impact of the location of the reference modes on GMM-LRDS, we take $\alpha = 1$ (same variance setting as the target) and $(\mathbf{m}_1, \mathbf{m}_2) \sim \mathrm{N}(-\mathbf{1}_d, \sigma_{\mathbf{m}}^2 \mathrm{I}_d) \otimes \mathrm{N}(\mathbf{1}_d, \sigma_{\mathbf{m}}^2 \mathrm{I}_d)$ where $\sigma_{\mathbf{m}} \in \{0, 0.25, 0.5, 0.75, 1.0\}$ is a mean perturbation parameter. Hence, the higher $\sigma_{\mathbf{m}}$, the further the reference modes might be with respect to the target modes. We display in Figure 10 the results of GMM-LRDS for this setting, averaged over 16 sampling runs. This ablation study notably reveals the major need of precision on the location of the target modes (in practice, brought by MCMC sampling) to build the reference distribution.

On the other hand, to understand the effect of the entropy of the reference modes, we take $(\mathbf{m}_1, \mathbf{m}_2) = (-\mathbf{1}_d, \mathbf{1}_d)$ (same mean setting as the target), and $\alpha \in \{0.5, 0.75, 1, 1.5, 2\}$, which can be seen as a variance perturbation parameter. We display in Figure 11 the results of GMM-LRDS for this setting, averaged over 16 sampling runs. Interestingly, our ablation study demonstrates that GMM-LRDS works well as soon as the support of the reference distribution includes the support of the target distribution. In practice, this is verified by MCMC sampling.

Table 10: Results of GMM-LRDS for bi-modal Gaussian mixtures with **diagonal** covariance and **various conditioning** settings ($d = 128$), averaged over 16 sampling runs. The MMD and KS results are displayed with 100-factor rescaling.

| Isotropic conditioning | | | | Medium conditioning | | | | Hard conditioning | | | |
|---|---|---|---|---|---|---|---|---|---|---|---|
| Mode Err. ↓ | $W_{2,\varepsilon}$ ↓ | MMD ↓ | KS ↓ | Mode Err. ↓ | $W_{2,\varepsilon}$ ↓ | MMD ↓ | KS ↓ | Mode Err. ↓ | $W_{2,\varepsilon}$ ↓ | MMD ↓ | KS ↓ |
| 6.6%±0.6% | 3.04±0.00 | 7.84±0.92 | 6.16±0.67 | 6.8%±0.9% | 1.86±0.00 | 8.33±1.29 | 6.81±0.98 | 6.2%±3.6% | 1.32±0.00 | 8.09±3.83 | 6.73±2.92 |

**Ablation study on GMM-LRDS: effect of the distance between the target modes.** To asses the robustness of GMM-LRDS with respect to the complexity of the target distribution, we conduct a second ablation study that illustrates the behaviour of GMM-LRDS for target with higher energy barrier. To do so, we set $d = 8$ and define the target distribution $\pi : x \in \mathbb{R}^d \mapsto (2/3)\mathrm{N}(x; -a\,\mathbf{1}_d, 0.05) + (1/3)\mathrm{N}(x; a\,\mathbf{1}_d, 0.05)$, where $a \in \{1, 2, 4, 8, 16\}$ indicates the complexity level of the target. For each target, we conduct 16 sampling runs of GMM-LRDS, and display the results in Figure 12. Our numerics show that the performance GMM-LRDS remains significantly consistent with increasing $a$, proving the interest of our method for complex multi-modal targets.

**Limitation of LRDS: high dimension.** Although LRDS outperforms its competitors when sampling from challenging bi-modal Gaussian mixtures, its performance tends to decrease when the dimension is large, like most of samplers. To highlight this limitation, we provide in Table 10 the results of GMM-LRDS for various mixture settings with $d = 128$, where we observe that it fails to recover a good estimation of the strongest mode weight (roughly, $10\%$ of relative error).

### I.5.2  MULTI-MODAL GAUSSIAN MIXTURES

We display in Figure 13 the results of all considered samplers, including LRDS, when targeting the multi-modal Gaussian mixtures defined in Appendix H.1 with fixed dimension $d = 8$ and increasing number of modes $L \in \{4, 8, 16, 32, 64\}$. To assess the sampling quality at a global level, we dispense for each sampler the total variation distance between the true weight histogram and the weight histogram obtained via Monte Carlo approximation. All metrics are computed based on 8192 samples. We notably observe that GMM-LRDS has the best performance compared to all other sampling methods, independently of the number of modes in the target distribution. In particular, Figure 13 demonstrates that GMM-LRDS is able to recover both global features and local features for a complex multi-modal distribution.

**Execution time.** To demonstrate the applicability of LRDS in practice, we provide in Figure 14 the execution clock-time of each sampling method for all multi-modal Gaussian mixture sampling settings. The conclusions are the same as in the bi-modal setting: GMM-LRDS has equivalent training time to previous variational diffusion-based approaches, with negligeable extra cost due to reference fitting.

**Limitation of LRDS: high number of modes.** As depicted in Figure 13 (top left), the recovery of the mode weights for a multi-modal target distribution gets worse as soon as the number of modes is large. Although GMM-LRDS performs better than its competitors, it suffers from this limitation.

### I.5.3  RINGS AND CHECKERBOARD DISTRIBUTIONS

In Figure 15 and Figure 16, we illustrate the impact of the number $J$ of Gaussian mixture components when running GMM-LRDS to sample from Rings and Checkerboard distributions, detailed in Appendix H.1,by taking $J \in \{1, 8, 16, 32, 64\}$. In particular, we consider mixture models with diagonal covariance (first two rows) and full covariance (last two rows), fitted on MCMC samples from the target via EM algorithm. For both settings, we observe that setting $J$ large enables to recover with more precision the support of the target distribution, leading consequently to better performance of GMM-LRDS. In the special Rings setting, where the target energy landscape is narrow and complex, we also notice that mixtures with full covariance provide better estimation of the support of the distribution. On the other hand, we remark that GMM-LRDS struggles to recover the geometry of the Checkerboard distribution, while EBM-LRDS is more performant, see Figure 6.

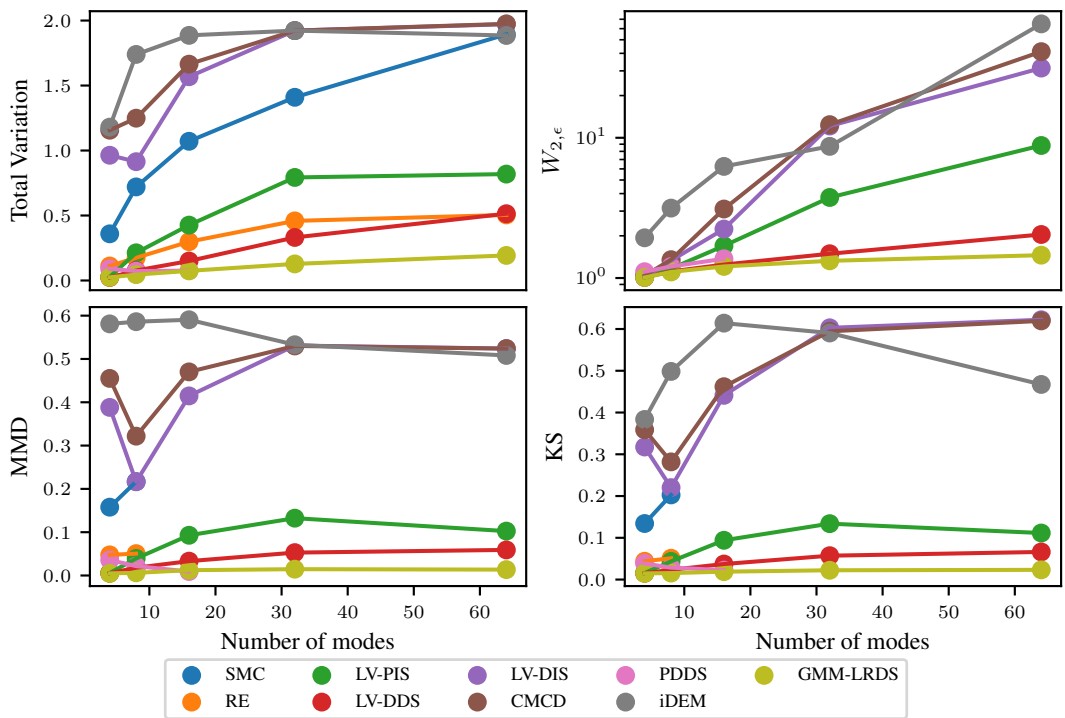

Figure 13: Results for multi-modal Gaussian mixtures, averaged over 16 sampling runs. (**Top Left**): Total variation distance between weight histograms. (**Top Right**): Results with regularized Wasserstein distance. (**Bottom Left**): Results with MMD distance. (**Bottom Right**): Results with KS distance. Incomplete curves refer to settings with missing results due to numerical issues.

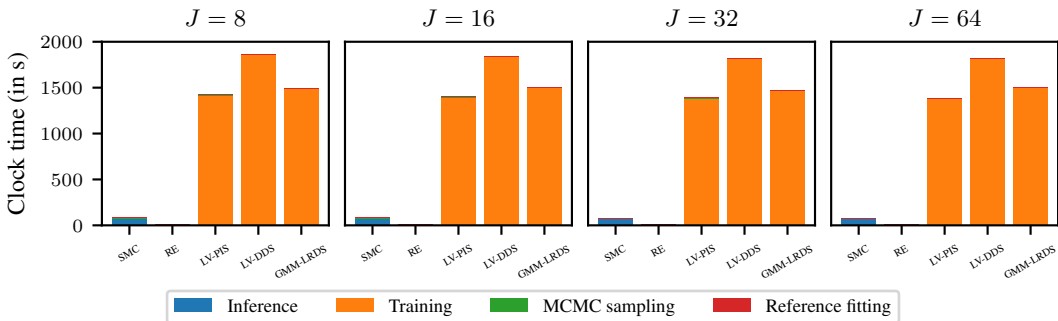

Figure 14: Execution clock-time for multi-modal Gaussian mixtures with increasing number of mixture components (from left to right).

### I.5.4 $\phi^4$ FIELD SYSTEM

Table 11 shows that all competing algorithms fail at recovering the weight ratio of the $\phi^4$ system as they are highly prone to mode collapse either on the 'negative' mode or on the 'positive' mode.

### I.5.5 BAYESIAN LOGISTIC REGRESSION MODELS

Finally, we display in Table 12 the results of all considered samplers, including LRDS, when targeting the Bayesian posterior distributions obtained from real-world data and defined in Appendix H.1. To assess the sampling quality, we compute the average predictive log-likelihood (i.e. the expected posterior distribution on the test dataset) via Monte Carlo approximation with 8192 samples. Since these distributions do not exhibit explicit multi-modal characteristics, we observe that standard MCMC-based techniques such as SMC and RE perform better than deep learning-based approaches. Interestingly, LRDS is on par or superior to previous diffusion-based approaches. For completeness, we also provide in Figure 17 the execution clock-time of all samplers in this setting.

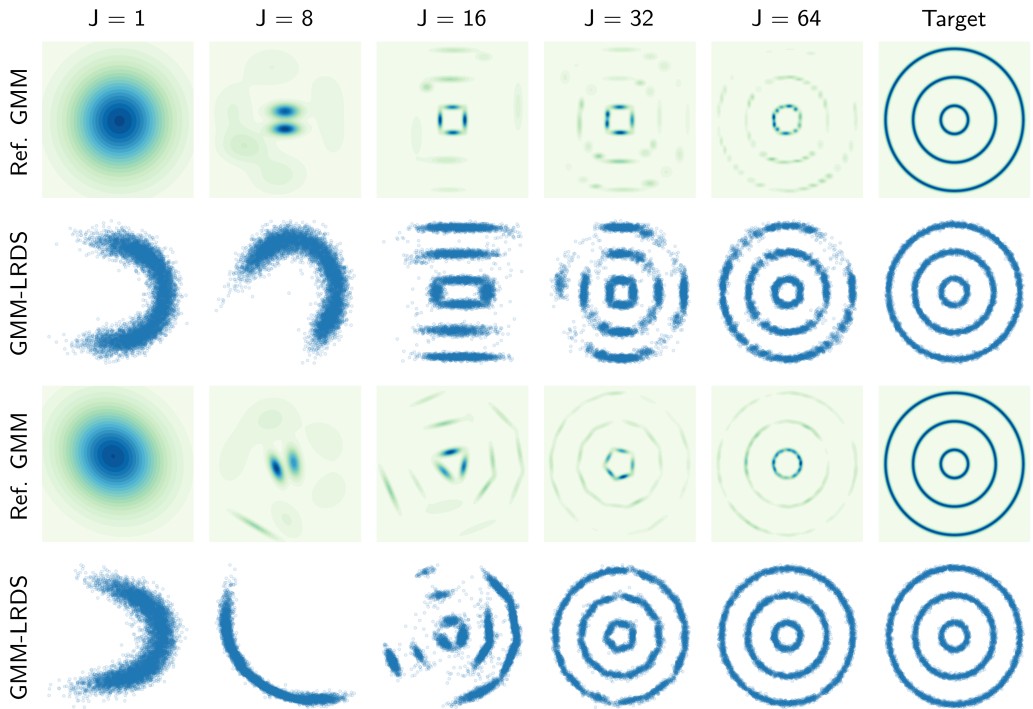

Figure 15: Results of GMM-LRDS for Rings distribution with increasingly expressive GMM for reference distribution - (**Top two rows**) GMM density and GMM-LRDS samples for **diagonal** covariance parameterization (**Bottom two rows**) GMM density and GMM-LRDS samples for **full** covariance parameterization.

Table 11: **Estimation of the weight ratio for the $\phi^4$ system** with different values of $h$, averaged over 16 runs. The Laplace approximations, that stand for ground truth, are displayed at the bottom of the table. Orange cells refer to settings with uninformative mode weight estimation (i.e., uniform mixture), red cells denote mode collapse. N/A denotes settings where results could not be obtained due to numerical issues.

| Algorithm | $h = 0$ | $h = 9 \times 10^{-4}$ | $h = 2 \times 10^{-3}$ | $h = 2.5 \times 10^{-3}$ | $h = 3.5 \times 10^{-3}$ |
|---|---|---|---|---|---|
| **SMC** | $1.28_{\pm 1.19}$ | $1.71_{\pm 1.47}$ | $1.51_{\pm 1.38}$ | $1.60_{\pm 1.52}$ | $1.57_{\pm 1.43}$ |
| **RE** | $1.00_{\pm 0.12}$ | $1.01_{\pm 0.12}$ | $1.04_{\pm 0.13}$ | $1.11_{\pm 0.13}$ | $1.07_{\pm 0.14}$ |
| **LV-PIS** | $\infty$ | $\infty$ | $\infty$ | $\infty$ | $\infty$ |
| **LV-DDS** | $0.00_{\pm 0.00}$ | $\infty$ | $0.00_{\pm 0.00}$ | $\infty$ | $\infty$ |
| **LV-DIS** | $1.07_{\pm 0.03}$ | $1.05_{\pm 0.03}$ | $1.05_{\pm 0.06}$ | $1.16_{\pm 0.12}$ | $1.17_{\pm 0.07}$ |
| **CMCD** | $1.01_{\pm 0.05}$ | $0.98_{\pm 0.07}$ | $1.01_{\pm 0.07}$ | $0.96_{\pm 0.07}$ | $0.96_{\pm 0.02}$ |
| **iDEM** | $0.59_{\pm 0.01}$ | $35.9_{\pm 1.43}$ | $98.6_{\pm 10.6}$ | $1.81_{\pm 0.03}$ | $170_{\pm 17.8}$ |
| **PDDS** | N/A | N/A | N/A | N/A | N/A |
| **Laplace 0th order** | 1.00 | 1.35 | 1.95 | 2.30 | 3.22 |
| **Laplace 2nd order** | 1.00 | 1.34 | 1.90 | 2.23 | 3.08 |

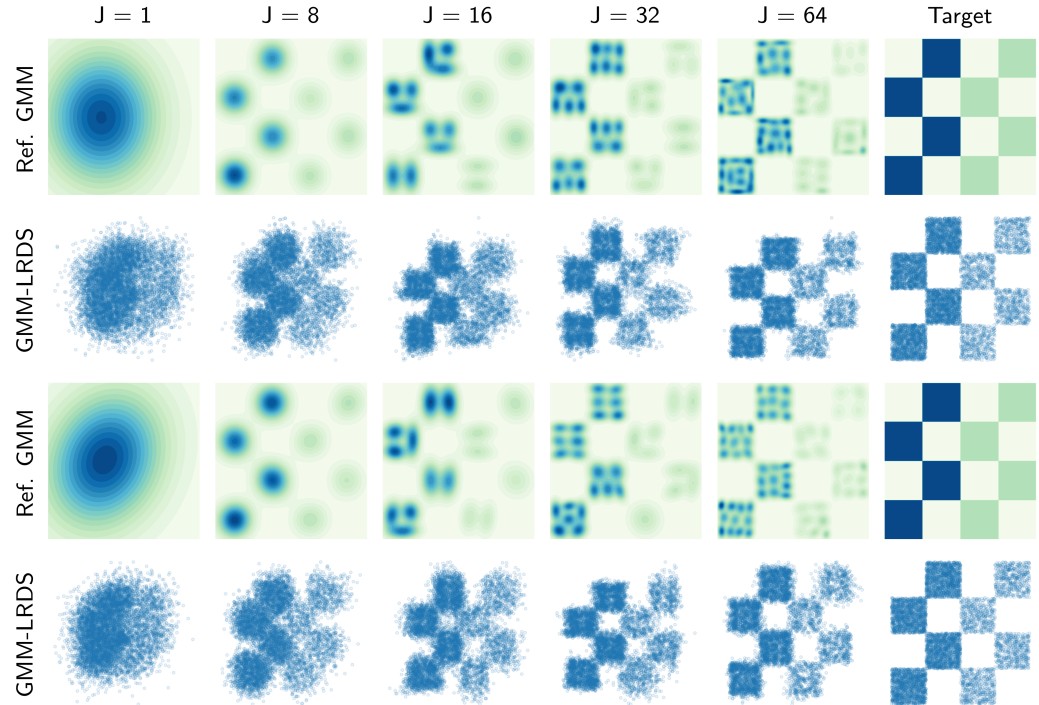

Figure 16: Results of GMM-LRDS for Checkerboard distribution with increasingly expressive GMM for reference distribution - **(Top two rows)** GMM density and GMM-LRDS samples for **diagonal** covariance parameterization **(Bottom two rows)** GMM density and GMM-LRDS samples for **full** covariance parameterization.

Table 12: **Average predictive log-likelihood** for Bayesian posterior distributions obtained from logistic regression tasks, averaged over 16 runs. Bold font indicates best performance.

| Algorithm | Ionosphere ($d = 36$) ↑ | Sonar ($d = 62$) ↑ | German Credit ($d = 26$) ↑ | Breast Cancer ($d = 32$) ↑ |
|---|---|---|---|---|
| SMC | $-139.4_{\pm 0.3}$ | $-191.4_{\pm 0.2}$ | $\mathbf{-122.2}_{\pm 0.1}$ | $\mathbf{-89.2}_{\pm 0.5}$ |
| RE | $-139.5_{\pm 0.2}$ | $-191.3_{\pm 0.1}$ | $\mathbf{-122.2}_{\pm 0.1}$ | $-90.1_{\pm 0.3}$ |
| LV-PIS | $-156.0_{\pm 1.0}$ | $-202.2_{\pm 0.2}$ | $-153.7_{\pm 2.2}$ | $-203.9_{\pm 3.3}$ |
| LV-DDS | $-147.7_{\pm 0.5}$ | $-195.2_{\pm 0.7}$ | $-151.2_{\pm 1.3}$ | $-128.4_{\pm 0.9}$ |
| LV-DIS | $-224.3_{\pm 22.6}$ | $-318.7_{\pm 47.0}$ | $-206.5_{\pm 1.0}$ | $-152.1_{\pm 1.9}$ |
| CMCD | $-970.6_{\pm 2.7}$ | $-1645.4_{\pm 6.2}$ | $-171.9_{\pm 0.0}$ | $-175.3_{\pm 5.7}$ |
| iDEM | $-153.7_{\pm 0.2}$ | $-4634.1_{\pm 12.9}$ | $-132.4_{\pm 0.1}$ | $-320.8_{\pm 0.9}$ |
| PDDS | $-139.6_{\pm 0.3}$ | $-191.3_{\pm 0.1}$ | $\mathbf{-122.2}_{\pm 0.1}$ | $-102.2_{\pm 0.1}$ |
| GMM-LRDS | $\mathbf{-138.0}_{\pm 0.5}$ | $\mathbf{-190.5}_{\pm 1.0}$ | $-129.0_{\pm 2.7}$ | $-100.1_{\pm 2.1}$ |
| EBM-LRDS | $-148.0_{\pm 15.4}$ | $-191.8_{\pm 0.9}$ | $-131.2_{\pm 5.2}$ | $-92.7_{\pm 4.0}$ |

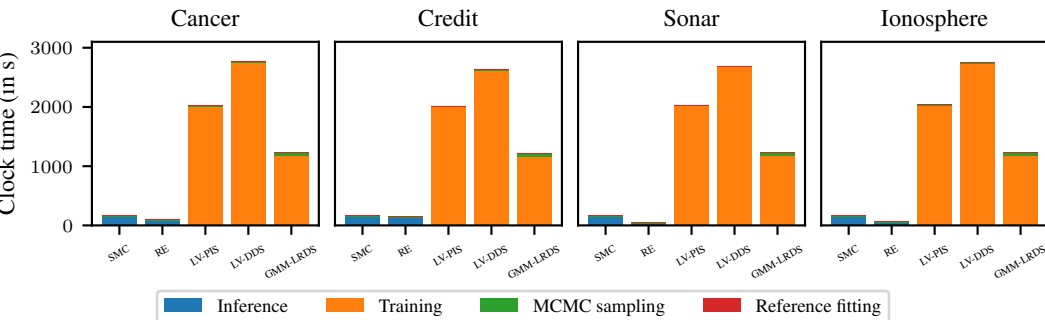

Figure 17: Execution clock-time for Bayesian posterior distributions obtained from logistic regression tasks 'Ionosphere', 'Sonar', 'German Credit' and 'Breast Cancer' (from left to right).

