# OpenReview forum: "Learned Reference-based Diffusion Sampler for multi-modal distributions"
_ICLR.cc/2025/Conference — ICLR 2025 Poster_

### Official Review · Reviewer_yQsx · 2024-10-18

**Soundness:** 4
**Presentation:** 3
**Contribution:** 3
**Rating:** 8
**Confidence:** 3

**Summary:**

Unlike previous score-based diffusion sampling methods, this paper takes a step further by learning a reference process based on GMMs or energy-based models (EBMs) to facilitate sampling from multi-modal distributions. The energy-based models (EMMs) presented are novel compared to recent literature, enabling the learning of references for various challenging distributions that GMM-based references cannot handle. Additionally, the authors summarize the key differences and innovations of their method compared to other diffusion sampling techniques in Table 1.

The authors also compare their approach with other state-of-the-art algorithms, including LV-PIS, LV-DDS, and LV-DIS, on various challenging distributions. And the results are satisfactory.

**Strengths:**

1. The paper proposes using GMMs or EBMs, based on prior knowledge of the modes, as a reference for sampling multi-modal densities. The EBM framework is particularly novel, as it helps learn initial references through a neural network for sampling a range of distributions that standard GMMs reference cannot handle.

2. The paper is technically sound, presenting solid theories, lemmas, and propositions in the Appendix, which provides a good supplement and well support their algorithms.

3. The paper compares their algorithm with various other state-of-the-art algorithms, and the results are satisfactory.

**Weaknesses:**

1. While the paper highlights leveraging prior knowledge of mode locations to avoid hyper-parameter tuning challenges, I believe the sensitivity still depends on the choice of reference distributions. The authors may illustrate how different choices of references (e.g., varying the number, means, variances of GMM components, or EBM architectures) impact the sampling results.

2. In the experiments, the modes of the target distributions (such as the high-dimensional Gaussian, rings, and checkerboard) are relatively close together. Have the authors tried target distributions with varying number of modes (e.g., 10, 100, 500) with increasing separation, and how does the algorithm perform in these cases?

3. The authors primarily evaluate mode weight recovery error, but there are other important evaluation metrics, such as Maximum Mean Discrepancy (MMD) and Wasserstein distance. Since it seems possible to obtain true samples for each of the experiments, I believe calculating MMD and Wasserstein distance would be feasible.

**Questions:**

Regarding the weaknesses, my questions are as follows:

1. How sensitive is your algorithm to the choice of different reference distributions (e.g., in GMMs, the variances/means of the modes; and in EBMs, the choice of hyper-parameters)?

2. How does the model perform when the modes of the target distributions are widely separated or when the number of modes is higher? What about the training and sampling complexities?

3. Can the authors present additional evaluation metrics to provide more straightforward assessments of sample quality?

---

> ### Author Response · Authors · 2024-11-21
> **Answer to Reviewer yQsx [1/2]**
>
> We thank Reviewer **yQsx** for their positive comments and their acknowledgement of the paper’s merits. We now tackle the main concerns raised by the reviewer.
>
> > While the paper highlights leveraging prior knowledge of mode locations to avoid hyper-parameter tuning challenges, I believe the sensitivity still depends on the choice of reference distributions. The authors may illustrate how different choices of references (e.g., varying the number, means, variances of GMM components, or EBM architectures) impact the sampling results.
>
> > How sensitive is your algorithm to the choice of different reference distributions (e.g., in GMMs, the variances/means of the modes; and in EBMs, the choice of hyper-parameters)?
>
> We thank the reviewer for their valuable feedback and detail below the response to their question.
> - In the GMM-LRDS setting, we recall that all of the parameters of the reference distribution - except the number of mixture components - are *learned* based on MCMC samples (namely, the means\variances of the components and the mixture weights) and therefore are completely data-driven, see Section 3.1 for more details. The sensitivity of LRDS to the value of the mixture weights is already presented in Figure 1 (right), which demonstrates that it has very few impact on the LRDS outcome. To address the sensitivity of LRDS to the other parameters, we provide in the **Appendix I.5.1** a small additional ablation study showing how the sampler reacts to intentionally tempered variances and means of the modes. We notably observe that the impact is completely limited. Regarding the number of mixture components, we acknowledge that it is fixed a priori, upon knowledge of the target distribution. Intuitively, the more components there are, the better the reference distribution will cover the target distribution. In the **Appendix I.5.3**, we provide a complementary ablation study on the impact of the number of components for the Rings setting. Following the reviewer’s suggestion, we will include these results in the revised version of the paper.
> - In the EBM-LRDS setting, the question of finding the optimal EBM architecture is beyond the scope of this paper, since the result would be completely task-dependent. Analogously to previous diffusion-based sampling methods that included deep learning tools, we keep the same neural network architecture fixed for all sampling tasks.

---

> > ### Author Response · Authors · 2024-11-21
> > **Answer to Reviewer yQsx [2/2]**
> >
> > > In the experiments, the modes of the target distributions (such as the high-dimensional Gaussian, rings, and checkerboard) are relatively close together. Have the authors tried target distributions with varying number of modes (e.g., 10, 100, 500) with increasing separation, and how does the algorithm perform in these cases?
> >
> > > How does the model perform when the modes of the target distributions are widely separated or when the number of modes is higher? What about the training and sampling complexities?
> >
> > We thank the reviewer for their interesting question. Following their suggestion, we provide in the **Appendix I.5.1** and **Appendix I.5.2** additional experimental results that tackle the mentioned challenges:
> > - We first conduct an additional experiment where the target is a Gaussian mixture with two modes, located in $-a. 1_d$ and $a. 1_d$ for increasing values of $a$, with respective weights $2/3$ and $1/3$ and local variance $0.05$, where $1_d$ denotes the vectors with all ones in dimension $d$. We consider $d=8$ and $a=1$ (original setting), $2, 4,8,16$. We observe that LRDS still outperforms their competitors in this setting.
> > - Then, we design a novel experiment where the target is a 8-dimensional Gaussian mixture with an increasing number of modes $L$ (namely, $4, 8,16, 32, 64$), uniformly picked in the hypercube $[-L,L]^d$, with local variance $0.05$. We set the mixture weights with strictly increasing values between the modes, such that the highest weight is 3 times larger than the lowest weight. To assess the recovery of the mode weight, we report the value of the Total Variation (TV) distance between the estimated weight histograms and the true weight histogram. Among all competing methods, LRDS achieves the best results. For $L=64$, all methods have poor performance: indeed, dealing with a high number of modes is a recurrent limitation of sampling methods.
> >
> > > The authors primarily evaluate mode weight recovery error, but there are other important evaluation metrics, such as Maximum Mean Discrepancy (MMD) and Wasserstein distance. Since it seems possible to obtain true samples for each of the experiments, I believe calculating MMD and Wasserstein distance would be feasible.
> >
> > > Can the authors present additional evaluation metrics to provide more straightforward assessments of sample quality?
> >
> > We acknowledge that the target-specific metrics used in Section 5 (that aim to estimate the recovery of the mode weights) do not entirely reflect the performance of the sampling procedures. We refer to **Paragraph 1 of the general response** for an explanation of our motivation to use those metrics. For completeness, we provide in the **Appendix I** the results with probability distance metrics suggested by the reviewer. In particular, we observe that LRDS still outperforms its competitors.

---

> > > ### Comment · Reviewer_yQsx · 2024-11-22
> > > **Reply to the Authors**
> > >
> > > I thank the authors for addressing my concerns and for their efforts in including additional ablation studies and evaluation metrics. I now have no further concerns about the paper.
> > >
> > > Overall, I believe the paper is of high quality and will make a valuable contribution to the field of diffusion sampling. As a result, I have increased my rating.

---

### Official Review · Reviewer_FUSS · 2024-11-03

**Soundness:** 3
**Presentation:** 3
**Contribution:** 3
**Rating:** 8
**Confidence:** 2

**Summary:**

This paper introduces the Learned Reference-based Diffusion Sampler (LRDS), a novel approach for sampling from multi-modal distributions. The authors identify a fundamental issue with existing variational diffusion-based methods (their high sensitivity to hyperparameters) and address this by proposing a methodology that leverages prior knowledge of target mode locations. LRDS is implemented in two variants: (1) GMM-LRDS, which is well-suited for a variety of target distributions while being relatively lightweight, and (2) EBM-LRDS, a more computationally intensive approach for harder sampling problems. Through extensive experiments on challenging multi-modal settings, the authors demonstrate that LRDS outperforms previous diffusion-based samplers.

**Strengths:**

1. The paper demonstrates a critical limitation in existing diffusion-based sampling methods. To solve the hyperparameter sensitivity issues in previous approaches, the authors provide a well-grounded theoretical method for their solution. The mathematical framework is developed and presented clearly, making their contributions theoretically sound and academically valuable.

2. The experimental evaluation is encompassing a diverse range of challenging distributions. The authors' comparative analysis against multiple baseline methods across different settings provides evidence for their method's superiority.

3. The paper offers two complementary approaches (GMM-LRDS and EBM-LRDS) that provide flexibility in handling different types of multi-modal distributions. These two versions of the method demonstrate the validity of the reference-based sampling approach proposed in this paper. Figures 5 and 6 show that the proposed method generally works well for various distributions.

**Weaknesses:**

1. A limitation of the proposed method lies in its computational overhead, particularly in the pre-training phase of the reference process models. The authors could have provided a more detailed analysis of the computational costs and optimization strategies as the complexity and scale of the target distribution increases.

2. The method's reliance on prior knowledge of mode locations (target distribution) represents a substantial practical limitation. While the authors acknowledge this requirement, the authors could address how the method might be adapted for scenarios where such information is incomplete.

3. There are concerns about the method's applicability to real-world data distributions, despite its impressive performance on synthetic examples. While the paper demonstrates effectiveness in controlled settings, the approach of learning mode information from reference samples may not translate well to real-world scenarios where modes are less well-defined and data structures are more complex and noisy. The lack of validation on real-world datasets leaves uncertainty about the method's practical utility and generalizability beyond carefully constructed example distributions.

**Questions:**

1. what the impact would be if random sampling was used to obtain reference samples rather than sampling that reflects a target distribution such as mcmc?

2. I understand that challenge mode collapse settings are rare in general real-world datasets. Although these experiments and analyzes may not be appropriate for the contribution of this paper, I wonder whether this sampling strategy can be applied to general real-world applications.

---

> ### Author Response · Authors · 2024-11-21
> **Answer to reviewer FUSS [1/2]**
>
> We thank Reviewer **FUSS** for their positive feedback on the originality and the quality of our contribution. We now would like to bring answers to their concerns.
>
> > There are concerns about the method's applicability to real-world data distributions, despite its impressive performance on synthetic examples. While the paper demonstrates effectiveness in controlled settings, the approach of learning mode information from reference samples may not translate well to real-world scenarios where modes are less well-defined and data structures are more complex and noisy. The lack of validation on real-world datasets leaves uncertainty about the method's practical utility and generalizability beyond carefully constructed example distributions. [...] I understand that challenge mode collapse settings are rare in general real-world datasets. Although these experiments and analyzes may not be appropriate for the contribution of this paper, I wonder whether this sampling strategy can be applied to general real-world applications.
>
> We thank the reviewer for their feedback and refer to **Paragraph 2** of the general response for a description of the additional experiments based on a real-world setting.
>
> > A limitation of the proposed method lies in its computational overhead, particularly in the pre-training phase of the reference process models. The authors could have provided a more detailed analysis of the computational costs and optimization strategies as the complexity and scale of the target distribution increases.
>
> We refer to **Table 4** for a theoretical analysis of the complexity of the various sampling algorithms used in our benchmark. Following the reviewer’s request, we provide in the **Appendix I** various tables which summarize the wall clock execution time for all sampling methods considered in the paper, including LRDS, when sampling from Gaussian mixtures and Bayesian posterior distributions. As asked, we detail the training time, the inference time as well as the reference training time (i.e., MCMC stage + GMM/EBM training time for LRDS). In particular, we observe that the cost of the last stage is negligible with respect to the cost of the training stage. Finally, we would like to emphasize that the training hyperparameters of LRDS are the same for all target distributions considered in the paper, independently of their complexity or scale, see **Appendix H.3**.
>
> > The method's reliance on prior knowledge of mode locations (target distribution) represents a substantial practical limitation. While the authors acknowledge this requirement, the authors could address how the method might be adapted for scenarios where such information is incomplete.
>
> We first would like to emphasize that a wide variety of multi-modal sampling applications in computational chemistry, see [4,5,6], are aware about the location of the target modes (given by an expert) and aim at recovering the relative weights between them, which is non trivial at all.
>
> In the case where this knowledge is only partial, i.e. if some modes of the target distribution are not covered well by the support of the reference distribution, those modes will be unlikely explored by the LRDS sampler, leading to mode collapse. This limitation is shared by all other competing methods that require substantial knowledge of the target distribution to set their hyperparameters, as we proceed in our experimental settings. To circumvent this issue, one could intentionally ‘expand’ the reference distribution support in LRDS (for instance, increasing the variances of the modes learned by GMM-LRDS) to expect covering unknown modes. However, this approach has no guarantees of success.
>
> > what the impact would be if random sampling was used to obtain reference samples rather than sampling that reflects a target distribution such as mcmc?
>
> As far as we understand, the reviewer suggests to employ in LRDS a reference distribution that is agnostic to the target distribution. In fact, this is exactly what previous reference-based methods [1,2,3] proposed, which leads to bad performance as we show in **Section 5**. To provide more intuition on this result, we recall that the LRDS variational loss is based on a density ratio between the reference distribution and the target distribution; hence, the larger the ratio mismatch (which can happen with random reference samples), the more challenging the learning problem becomes.

---

> > ### Author Response · Authors · 2024-11-21
> > **Answer to reviewer FUSS [2/2]**
> >
> > [1] Qinsheng Zhang, & Yongxin Chen (2022). Path Integral Sampler: A Stochastic Control Approach For Sampling. In International Conference on Learning Representations.
> >
> > [2] Francisco Vargas, Will Sussman Grathwohl, & Arnaud Doucet (2023). Denoising Diffusion Samplers. In The Eleventh International Conference on Learning Representations .
> >
> > [3] Lorenz Richter, & Julius Berner (2024). Improved sampling via learned diffusions. In The Twelfth International Conference on Learning Representations.
> >
> > [4] Sucerquia et al, Ab initio metadynamics determination of temperature-dependent free-energy landscape in ultrasmall silver clusters, J. Chem. Phys. 156, 154301 (2022) https://doi.org/10.1063/5.0082332
> >
> > [5] Noé, Frank, Simon Olsson, Jonas Köhler, and Hao Wu. “Boltzmann Generators: Sampling Equilibrium States of Many-Body Systems with Deep Learning.” Science 365, no. 6457 (September 6, 2019): eaaw1147. https://doi.org/10.1126/science.aaw1147.
> >
> > [6] Ding, Xinqiang, and Bin Zhang. “DeepBAR: A Fast and Exact Method for Binding Free Energy Computation.” The Journal of Physical Chemistry Letters 12, no. 10 (March 18, 2021): 2509–15. https://doi.org/10.1021/acs.jpclett.1c00189.

---

> > > ### Author Response · Authors · 2024-11-30
> > >
> > > Dear Reviewer FUSS,
> > >
> > > As the discussion period nears its conclusion, we would like to kindly inquire whether we have fully addressed your concerns. In our rebuttal, we expanded our evaluation by incorporating additional real-world distributions (beyond the $\phi^4$ field-system setting), with a particular emphasis on Bayesian logistic regressions. We also provided references and detailed explanations to support the realism of our setting and assumptions. Furthermore, we conducted an in-depth ablation study on the computational aspects of our algorithm, carefully examining the impact of each parameter.
> > >
> > > If there are any remaining issues, we would be more than happy to provide further clarifications. Otherwise, we would greatly appreciate it if you could consider updating your score.

---

### Official Review · Reviewer_GBCd · 2024-11-03

**Soundness:** 2
**Presentation:** 1
**Contribution:** 2
**Rating:** 3
**Confidence:** 3

**Summary:**

The authors propose a sampling method (named as LRDS) which extends existing works to sample from a probability density under the assumptions that they know the location of modes of the distribution while do not have access to the ground truth samples. Inspired by the observations that the existing works are sensitive to hyperparameters, the authors propose two methods (GMM-LRDS and EBM-LRDS) that surpasses the performance of the baselines on sampling from multimodal distributions.

**Strengths:**

- Trying to solve interesting and fundamental problems.
- Performance improvement over the baselines for estimating mode weight is shown.

**Weaknesses:**

- Please revise the introduction entirely from the high-level perspective. Currently, Introduction is written as if sort of a problem definition and preliminary knowledge (which are usually put in related works or method section).
    1. What is the task to solve?
    2. What are the real-world applications?
    3. What other baselines have been proposed? And what are the limitations of them?
    4. What is the proposed method concisely and what is the improvement that is theoretically and empirically shown in the paper?
    - It should be understandable to some extent for non-expert readers in this domain.
    - I believe this is a basic rule of thumb in paper writing.
- Weak motivation
    - Why do we need to care about this paper’s assumptions (L40-L41)? Is it a realistic scenario? What are the examples in practice?
    - Why do we need to sample from multi modal distributions? (L35-L36) Why not simply sampling from a single-mode distribution?
- Weak experiments
    - Higher dimensional dataset experiments are needed (e.g., MNIST or Cifar10) to show that the proposed methods can be applicable beyond the toy-level experiments.
    - Quantitative experiments
        - The measure for computing the distance/divergence with the target distribution is required.
        - How is Table 2 (mode weight estimation error) connected to the actual task which is approximating the target distribution?
        - It seems like each column of Table 2 is almost identical. Is there something meaningful to report?

- Overall, the proposed method itself is interesting. However, its presentation (paper writing) needs to be significantly revised and the experiments are not good enough to show the performance improvements in sampling and to verify that the proposed method can be applicable in the real world applications.

**Questions:**

1. Fig. 4. Is GMM-LRDS as fast as the Laplace approximations?
2. It is very difficult to find strengths of the proposed methods in Fig. 5. LV-DDS looks better, actually.

---

> ### Author Response · Authors · 2024-11-21
> **Answer to reviewer GBCd [1/3]**
>
> We thank Reviewer **GBCd** for acknowledging the interest of our submission. Below, we respond to their main concerns.
>
> > The measure for computing the distance/divergence with the target distribution is required. How is Table 2 (mode weight estimation error) connected to the actual task which is approximating the target distribution?
>
> We acknowledge that the target-specific metrics used in **Section 5** (that aim to estimate the recovery of the mode weights) do not entirely reflect the performance of the sampling procedures. We refer to **Paragraph 1** of the general response for an explanation of our motivation to use those metrics. For completeness, we provide in the Appendix I the results with probability distance metrics suggested by the reviewer. In particular, we observe that LRDS still outperforms its competitors.
>
> > the experiments are not good enough to show the performance improvements in sampling
>
> We respectfully disagree with the reviewer on this precise statement as most of the results in **Section 5** demonstrate that our method outperforms competitors on target-specific weight metrics by a large margin. This shows that LRDS outperforms its competitor in capturing global features of multi-modal distribution. These results are confirmed by the results with additional metrics given in the **Appendix I**.
>
> > Higher dimensional dataset experiments are needed (e.g., MNIST or Cifar10) to show that the proposed methods can be applicable beyond the toy-level experiments. [...] the experiments are not good enough [...] to verify that the proposed method can be applicable in the real world applications.
>
> As explained in **Paragraph 2** of the general response, most of related works considered synthetic target densities to validate their sampling approach, which can already be considered challenging. To address the specific remark of the reviewer on the lack of ‘image’ experiments, we would like to highlight that image sampling is commonly related to generative modeling, where ground truth samples are already available (which is not the case in our setting). In particular, we are unaware of a standard image sampling task solely based on probability density evaluations and are open to suggestions to conduct new experiments for this setting.
>
> > its presentation (paper writing) needs to be significantly revised. Please revise the introduction entirely from the high-level perspective. Currently, Introduction is written as if sort of a problem definition and preliminary knowledge (which are usually put in related works or method section).
>
> We thank the reviewer for their valuable feedback. Nonetheless, we would like to highlight that all of the other reviewers (**NXKL**, **FUSS** and **yQsx**) praised the precise writing of our paper, as mentioned in the beginning of the general response.
>
> We now would like to address the specific concerns made on the introduction of the submission. With all respect, we humbly believe that the criticism made by the reviewer is not well founded since our introduction already answers the questions 1-2-3-4 mentioned by the reviewer in the same order. To be more specific:
> - The problem definition (question 1) is introduced in **L31-41**,
> - The sampling applications (question 2) are mentioned in **L31-41**,
> - Regarding the different baselines on multi-modal sampling with their main limitations (question 3), the reviewer may find a brief overview of Annealed MCMC samplers (**L43-L57**), Annealed VI samplers (**L66-L73**) and recent diffusion-based samplers (**L74-L85**).
> - Finally, regarding the summary of our method and its improvements (question 4), the ‘Contributions’ paragraph (**L86-L96**) explains that LRDS can be seen as an instance of an existing diffusion-based framework where auxiliary models are trained to boost the existing ones. This paragraph also states that our method brings two main improvements compared to related methods when sampling from multi-modal distributions: (i) the lack of crucial hyper-parameters that would need ground truth samples to be tuned and (ii) the recovery of accurate relative mode weights for challenging distributions. Nonetheless, we agree with the reviewer on the fact that this paragraph does not ‘concisely’ describe our method from a high-level perspective. This choice was made on purpose as it would require prior technical knowledge on variational diffusion-based samplers to understand the subtle algorithmic differences (which is not expected from non-expert readers)

---

> ### Author Response · Authors · 2024-11-21
> **Answer to reviewer GBCd [2/3]**
>
> > Why do we need to care about this paper’s assumptions (L40-L41)? Is it a realistic scenario? What are the examples in practice?
>
> We first would like to insist that no existing sampling algorithm is able today to obtain the accurate mode weights of challenging distributions, without a priori information on the target distributions. Most current methods depend on hyperparameter tuning that requires non-trivial information on the target such as the location of the modes, or even ground truth samples, see **Table 8** in [1] for example. In the context where light prior information is given (here, only the location of the modes of the target distribution), we propose a methodology that outperforms most competitors.
>
> Regarding the question addressing how realistic this assumption is, we would like to emphasize that a wide variety of multi-modal sampling applications are aware about the location of the target modes (given by an expert) and aim at recovering the relative weights between them. This is for example the case when looking for free energy differences between known isomers of a molecule [2,3] or the fraction of ligand-protein binding events [4].
>
> > Why do we need to sample from multi modal distributions? (L35-L36) Why not simply sampling from a single-mode distribution?
>
> The problem of sampling from distributions with single-mode is a very well known problem, for which various algorithms (mainly, MCMC-based) [5,6] have been proposed in the past years, with robust performance (even in high dimensional settings) and strong theoretical guarantees.
>
> However, multi-modal distributions are ubiquitous in scientific applications (e.g., [2,3,4]), and in this case, samplers that perform well on single-mode targets are not adapted at all (see **Appendix I.3** which illustrates this limitation). This is explained by the fact that sampling independently from each uni-modal component of the target distribution leads to strongly biased estimations of the full energy landscape. Inspired by the impactful success of diffusion-based generative models, recent sampling literature aims at using the diffusion framework to specifically focus the challenging multi-modal sampling task (see [1] for a review).
>
> > It seems like each column of Table 2 is almost identical. Is there something meaningful to report?
>
> The reviewer is right in noticing that the columns of **Table 2** are very close. This highlights that (i) LRDS can achieve good metrics even in increasing dimensions and (ii) that the curse of dimensionality is not the fundamental limitation of the competitors, i.e., the problem lies in how multi-modality itself is handled.
>
> > Fig. 4. Is GMM-LRDS as fast as the Laplace approximations?
>
> In **Figure 4**, the Laplace approximation is not a sampling algorithm. It refers to an estimation of the relative weight $w_-/w_+$ of the $\phi^4$ distribution, for varying parameter h (which reflects the level of imbalance between the two modes), based on the analytical expression of the target density. We use the 0th order and the 2nd order of this approximation to obtain a truthful interval around the exact relative weight of the PhiFour distribution (recalling that ground truth samples are not available). We will clarify this point more clearly in the next revision to avoid any confusion.
>
> > It is very difficult to find strengths of the proposed methods in Fig. 5. LV-DDS looks better, actually.
>
> The reviewer is right. This very toy experiment shows that some samplers among competitors can deal with “simple” multimodality (namely RE, DDS, DIS), as well as EBM-LRDS. However, in the case where the target is a Gaussian mixture with high energy barriers (see **Table 2** and Appendix) or the $\phi^4$ distribution (see **Figure 4**), most of them fail in mode weight recovery, which demonstrates that they cannot handle multi-modality from a general perspective.

---

> > ### Author Response · Authors · 2024-11-21
> > **Answer to reviewer GBCd [3/3]**
> >
> > [1] Blessing, D., Jia, X., Esslinger, J., Vargas, F., & Neumann, G. (2024). Beyond ELBOs: A Large-Scale Evaluation of Variational Methods for Sampling. In Proceedings of the 41st International Conference on Machine Learning (pp. 4205–4229). PMLR.
> >
> > [2] Sucerquia et al, Ab initio metadynamics determination of temperature-dependent free-energy landscape in ultrasmall silver clusters, J. Chem. Phys. 156, 154301 (2022) https://doi.org/10.1063/5.0082332
> >
> > [3] Noé, Frank, Simon Olsson, Jonas Köhler, and Hao Wu. “Boltzmann Generators: Sampling Equilibrium States of Many-Body Systems with Deep Learning.” Science 365, no. 6457 (September 6, 2019): eaaw1147. https://doi.org/10.1126/science.aaw1147.
> >
> > [4] Ding, Xinqiang, and Bin Zhang. “DeepBAR: A Fast and Exact Method for Binding Free Energy Computation.” The Journal of Physical Chemistry Letters 12, no. 10 (March 18, 2021): 2509–15. https://doi.org/10.1021/acs.jpclett.1c00189.
> >
> > [5] Roberts, G. O., & Tweedie, R. L. (1996). Exponential Convergence of Langevin Distributions and Their Discrete Approximations. Bernoulli, 2(4), 341–363. https://doi.org/10.2307/3318418
> >
> > [6] Matthew D. Hoffman, & Andrew Gelman (2014). The No-U-Turn Sampler: Adaptively Setting Path Lengths in Hamiltonian Monte Carlo. Journal of Machine Learning Research, 15(47), 1593–1623.

---

> > > ### Author Response · Authors · 2024-11-30
> > >
> > > Dear Reviewer GBCd,
> > >
> > > As the discussion period comes to a close, we would like to kindly check in to ensure we have fully addressed your concerns. In our rebuttal, we have incorporated additional real-world distributions (beyond the $\phi^4$ field-system setting), focusing on Bayesian logistic regressions, and clarified the case of image distributions in our general response. We have also provided references and detailed explanations to support the realism of our setting and assumptions. Furthermore, in the revised manuscript, we have included several new metrics, as you suggested, which demonstrate that our algorithm is a superior sampling method for multimodal distributions compared to its competitors.
> > >
> > > If there are any remaining questions or concerns, we would be happy to provide further clarification. Otherwise, we would greatly appreciate it if you could consider updating your score.

---

> ### Comment · Reviewer_GBCd · 2024-11-30
>
> I appreciate the authors for their responses to my questions.
> I had two primary concerns. First, weak motivations and second, weak experiments.
>
> 1.1 ) Weak motivation —— Introduction
>   - I still do think Introduction has too much information about technical details while has too little information on 1) why the problem that this method tries to solve is important in practice, and 2) what would be the practical benefits of the proposed method that the previous methods were short of. I believe though I somehow understood the reviewer’s points from the response. I think our definition of “practical (real-world) applications” is different. To me, Bayesian statistics, statistical mechanics, and molecular dynamics are too general to frame as specific applications. What are the specific applications in practice that the proposed methods can benefit? For example, which application of Bayesian statistics? The author mentioned that “We show that GMM-LRDS and EBM-LRDS outperform previous diffusion-based samplers on challenging multi-modal settings.” Without understanding 1) what the applications of sampling from challenging multi-modal setting are, and 2) why it is important, I cannot say that I understand the value of the proposed method.
>
> 1.2 ) Weak motivation —— Assumptions
>   - I do not understand the answer of the reviewers on how realistic the assumption is. The assumption is:  “we have access to the location of the modes as prior information on π. However, we do not assume to have access a priori to ground truth samples from π.” Is it practical that we don’t have access to samples but know the modes of the distribution? What could be the example cases? I would appreciate it if this point can be elaborated.
>
> 2.1) Weak experiments
>   - most of my concerns are resolved here, but I still do not understand why this method cannot be extended to higher-dimensional experiments. It does not have to be image experiments. (As I mentioned, MNIST or Cifar10 experiments are just examples.)
>
>  Overall, some of my concerns are resolved, but I still do not understand 1) the practical significance of the proposed methods and 2) if the assumption is realistic in practice  and 3) extensibility towards higher dimensional distributions. Thus, I would keep my initial rating. I would appreciate it if the authors could provide further clarifications.

---

> > ### Author Response · Authors · 2024-12-01
> > **Answer to reviewer GBCd [1/2]**
> >
> > **Answer to 1.1 and 1.2 about the motivations of our setting.**
> >
> > We hear your concerns about the fact you find that our introduction and our setting both lack motivation. We first would like to recall that the problem of sampling from an unnormalized distribution is an ubiquitous and important task in many scientific domains. In our previous answer, we mentioned meaningful use-cases [1,2,3] of multi-modal distributions. To specifically address your concern, we detail here two of the previously given references and highlight how they fit our framework.
> >
> > 1. In [2], the authors study configurations of metal nanoclusters (also called isomers) under external conditions such as temperature. In the case of silver or gold materials, experimental and simulated absorption spectrums at low temperature indicate the coexistence of **several known isomers**. If the distribution of those isomers is known to be uniform for 12 atoms of gold at room temperature, experiments also highlighted that **proportions** between those configurations could vary a lot with temperature change. According to [2], identifying those **proportions** (often expressed via free-energy differences) is key to address isomerization questions or to understand the influence of stabilizers (like solvents or organic matter). In our generic framework, those configurations can be assimilated to the modes of the target distribution. While we know the location of these modes, we are unable to have a priori samples since we don’t have access to the accurate **proportions**.
> > 2. In [3], the authors study the *Bovine Pancreatic Trypsin Inhibitor* protein (BPTI). By running Molecular Dynamics (MD) on a super-computer (~ 3 months of compute), scientists discovered two distincts structures : the near crystallographic “X” structure and the open “O” structure (see their Fig. 5). In our setting, those **two known structures** would correspond to the modes of the target distribution. Being able to compute the **free energy difference** between these two structures (which is related to their respective probability proportions) would improve the understanding of this protein. According to the authors, estimating this quantity using classic MD and a GTX1080 Ti GPU would **take 30 days**, which proves the interest in accelerating MD with more elaborated techniques.
> >
> > In these two examples, although the location of the modes is known, sampling from the target distribution is still a very challenging problem, and is motivated by significant applications. While reference to [2,3] was given in our previous response, we acknowledge that more details could have been provided. We will include the two previous paragraphs in the revised version of our paper to put our work into perspective as the reviewer adviced.
> >
> > **Answer to 1.1 and 1.2 about the benefits of our setting.**
> >
> > Note that directly applying our algorithm on problems considered in [2,3] is beyond the scope of our work as our primary goal is to showcase the ability of LRDS to accurately sample from simpler systems, which share the same multimodality. Those distributions can still be considered challenging as previous methods are unable to sample them properly. Although our numerical examples use less complex distributions compared to [2,3], they illustrate well LRDS's practical benefits, especially in estimating mode weights. Extending our method to more complex distributions, with necessary adaptations, is a promising direction for future work.
> >
> > We then would like to underline that our work is above all methodological, with a special focus on multi-modal distributions, in line with previous diffusion-based sampling methods based on machine learning tools ([a] (ICLR 2022), [b] (ICLR 2023), [c] (ICLR 2024), [d] (ICLR 2024), …). In particular, we adopt the exact same approach as those papers by (1) considering a generic mathematical setting where we have access to a target density without ground truth samples, (2) proposing a general sampling algorithm and (3) verifying its applicability on challenging distributions with numerical experiments. Regarding (3), we follow a very similar experimental setup compared to the works previously cited. On the other hand, we include our $\phi^4$ experiment that can be considered as a ‘real-world’ example drawn from statistical mechanics. On this matter, we bring to the reviewer’s attention the fact that all our competitors fail in contrast to LRDS.

---

> ### Author Response · Authors · 2024-12-01
> **Answer to reviewer GBCd [2/2]**
>
> **Answer to 2.1 on high dimensional experiments**
>
> * Up to our knowledge, existing sampling methods based on diffusion schemes currently fail at sampling in very large dimensions [a,b,c,d]. We show in our work that they even fail in moderate dimensions for multi-modal targets and propose a solution to their default. While we do not provide a complete solution to the problem of sampling from high dimensional multi-modal distributions, we bring some stones to address this task. Solving this problem in full generality is a long standing problem that we do not pretend to solve.
> * Given the clear superiority of LRDS in moderate dimensions (where previous methods fail), we leave its application to the very high-dimensional setting as future work.
>
> We hope that our responses have addressed the reviewer’s concerns regarding the practical significance, realism of the assumptions, and extensibility to higher-dimensional distributions. If there are any remaining uncertainties, we would be happy to clarify further. We kindly ask the reviewer if they would consider updating their score based on these clarifications.
>
> [a] Qinsheng Zhang, & Yongxin Chen (2022). Path Integral Sampler: A Stochastic Control Approach For Sampling. In International Conference on Learning Representations.
>
> [b] Francisco Vargas, Will Sussman Grathwohl, & Arnaud Doucet (2023). Denoising Diffusion Samplers. In The Eleventh International Conference on Learning Representations .
>
> [c] Lorenz Richter, & Julius Berner (2024). Improved sampling via learned diffusions. In The Twelfth International Conference on Learning Representations.
>
> [d] Francisco Vargas, Shreyas Padhy, Denis Blessing, & Nikolas Nusken (2024). Transport meets Variational Inference: Controlled Monte Carlo Diffusions. In The Twelfth International Conference on Learning Representations.

---

### Official Review · Reviewer_NXKL · 2024-11-04

**Soundness:** 4
**Presentation:** 3
**Contribution:** 3
**Rating:** 6
**Confidence:** 2

**Summary:**

This paper introduces a new approach to sampling from multi-modal distributions called Learned Reference-based Diffusion Sampler (LRDS). The proposed method is an extension of previous variational diffusion-based samplers and tries to address their limitations related to their sensitivity to small perturbations of hyperparameters. The authors propose two approaches. The first one GMM-LRDS is based on Gaussian Mixture Models, a more lightweight approach targeting simpler target distributions. The second one EBM-LRDS uses a more computationally expensive Energy-Based Model, allowing one to sample from harder target distributions.

**Strengths:**

1. The paper's structure is logical and well-organized, and the writing is clear and concise, making it easy to follow and understand.
2. The paper demonstrates a strong mathematical foundation that supports the authors' claims. The inclusion of detailed mathematical proofs in the appendix adds to the paper's technical depth.
3. In contrast to previous methods, which rely on hyperparameter tuning for reference distributions, the proposed approach learns these hyperparameters.

**Weaknesses:**

1. While the figures are generally well-presented, some may benefit from additional context or explanation in the main text. For example, the importance of parameter $h$ in Figure 4 is not obvious without checking the appendix.
2. The authors present the experiments only on the synthetic datasets. The paper would greatly benefit from the inclusion of experiments on real-world datasets (e.g., images), which would help to demonstrate the practical applicability.
3. The authors provide the computational complexity of the overall sampling procedure for different methods in the appendix; however, the memory complexity and exact timings might help the reader.

**Questions:**

I would like to suggest authors to
1. add the comparison on real-world datasets.
2. add the exact timing comparison of the discussed methods for training and sampling stages.
3. expand the experiments section with an ablation study of the LRDS limitations.

---

> ### Author Response · Authors · 2024-11-21
> **Answer to reviewer NXKL**
>
> We thank Reviewer **NXKL** for their valuable feedback on our work. Below we provide a detailed answer to the reviewer’s concerns.
>
> > The authors present the experiments only on the synthetic datasets. The paper would greatly benefit from the inclusion of experiments on real-world datasets (e.g., images), which would help to demonstrate the practical applicability. [...] I would like to suggest authors to: 1.add the comparison on real-world datasets.
>
> We thank the reviewer for their beneficial remark and refer to **Paragraph 2** in the general response for a description of the additional experiments based on a real-world setting. To address the specific remark of the reviewer on the lack of ‘image’ experiments, we would like to highlight that image sampling is commonly related to generative modeling, where ground truth samples are already available (which is not the case in our setting). In particular, we are unaware of a standard image sampling task solely based on probability density evaluations and are open to suggestions to conduct new experiments for this setting.
>
> > The authors provide the computational complexity of the overall sampling procedure for different methods in the appendix; however, the memory complexity and exact timings might help the reader [...] I would like to suggest authors to: 2.add the exact timing comparison of the discussed methods for training and sampling stages.
>
> Following the reviewer’s question, we provide in **Appendix I** various tables which summarize the wall clock execution time for all sampling methods considered in the paper, including LRDS, when sampling from Gaussian mixtures and Bayesian posterior distributions. As asked, we detail the training time, the inference time as well as the reference training time (i.e., MCMC stage + GMM/EBM training time for LRDS).
>
> > I would like to suggest authors to: 3. expand the experiments section with an ablation study of the LRDS limitations.
>
> We acknowledge that our original submission does not put much emphasis on LRDS limitations. In practice, LRDS shares some of the  limitations of the competing methods: (i) the curse of dimensionality, which is well highlighted in the **Appendix I.5.1**, and (ii) a high number of modes, as depicted in the **Appendix I.5.2**.
>
> > While the figures are generally well-presented, some may benefit from additional context or explanation in the main text. For example, the importance of parameter h in Figure 4 is not obvious without checking the appendix.
>
> We apologize for this imprecision in the main paper, which is mainly due to the lack of space. We will put much effort on detailing the target distributions in the main part upon acceptance of the paper. As detailed in **Appendix H.1**, the non-negative parameter h in the $\phi^4$ experiment controls how much the target is imbalanced between the two modes : while $h=0$ corresponds to a perfectly equilibrated target, taking h large makes the negative mode more predominant . The challenge is to predict quantitatively the weight ratio.
>
> [1] Blessing, D., Jia, X., Esslinger, J., Vargas, F., & Neumann, G. (2024). Beyond ELBOs: A Large-Scale Evaluation of Variational Methods for Sampling. In Proceedings of the 41st International Conference on Machine Learning (pp. 4205–4229). PMLR.

---

> > ### Author Response · Authors · 2024-11-30
> >
> > Dear Reviewer NXKL,
> >
> > As the discussion period draws to a close, we would like to kindly check in once more to ensure we have fully addressed your concerns. In our rebuttal, we have expanded our evaluation to include additional real-world distributions (beyond the $\phi^4$ field-system setting), with a particular focus on Bayesian logistic regressions, and clarified the case of image distributions in our general response. Additionally, we have conducted a detailed ablation study on the computational aspects of our algorithm, thoroughly analyzing the impact of each parameter.
> >
> > If there are any remaining issues, we would be happy to provide further clarification. Otherwise, we would greatly appreciate it if you could consider updating your score.

---

### Author Response · Authors · 2024-11-21
**General response [1/3]**

We thank the reviewers for their insightful comments and are encouraged by their positive feedback. Overall, the reviewers recognized the mathematical value and the novelty of our work. In particular:
- Reviewer **NXKL** explained that *‘the paper's structure is logical and well-organized, and the writing is clear and concise, making it easy to follow and understand’*,
- Reviewer **FUSS** recognized that our paper *‘demonstrates a critical limitation in existing diffusion-based sampling methods’* and acknowledged that *‘the mathematical framework is developed and presented clearly, making [our] contributions theoretically sound and academically valuable’*,
- Reviewer **yQsx** finally highlighted that *‘the paper is technically sound, presenting solid theories, lemmas, and propositions in the Appendix’*.

Regarding their concerns, we provide detailed responses to each of them but summarize here their main feedback. **We revised our manuscript following the reviewers’ suggestions and highlighted the changes in blue.**

---

> ### Author Response · Authors · 2024-11-21
> **General response [2/3]**
>
> ## 1. About the choice of the metric when sampling from multi-modal distributions
>
> First, we would like to recall that the main motivation of LRDS is to recover the good proportions between the modes of a given target distribution, which is one of the main challenges when sampling from complex multi-modal distributions. Indeed, standard sampling techniques (such as MCMC samplers) are already very efficient at capturing the local energy landscape around the modes, but may be unable to correctly estimate global features. This explains why we intentionally measure the quality of sampling through the perspective of relative mode weight recovery.
>
> As pointed out by reviewers **GBCd** and **yQsx**, the results of our numerical experiments omit statistical estimations of standard probability metrics, such as the Wasserstein distance or the Maximum Mean Discrepancy (MMD) with respect to ground truth samples (note that this is not possible in the PhiFour experiment). This choice was made on purpose as these metrics provide a full description of the sampling quality (including global and local aspects) without assessing separately the mode weight recovery error. However, following the suggestions of the reviewers, we display in the **Appendix I** the results of all of the experiments conducted in our paper (excluding PhiFour) with the Wasserstein, MMD and Kolmogprov-Smirnov metrics. One can observe that LRDS still outperforms its competitors for those metrics.

---

> > ### Author Response · Authors · 2024-11-21
> > **General response [3/3]**
> >
> > ## 2. Experiments with real-world data
> >
> > The reviewers highlighted the fact that a number of our experiments corresponded to synthetic settings. We would like to point out that most of the recent diffusion-based sampling methods [0,1,2,3,4] also considered synthetic target densities that exhibit multimodality characteristics; see also [5] for an overview. We also decided to focus on standard synthetic multi-modal distributions (Gaussian mixtures, Rings, Checkerboard), which are challenging as no previous sampling method is able to compute the accurate relative mode weight for all of these settings, as shown in Section 5, while having a tractable ground truth to compare to. On the other hand, we proposed the challenging PhiFour experiment, see Figure 4, which is a realistic sampling setting from statistical mechanics.
> >
> > As most of the reviewers (**NXKL**, **GBCd**, **FUSS**) deemed that further ‘real world’ data experiments would be a valuable addition to the paper, we have conducted additional experiments based on the benchmark of [5] to fill this gap. We consider target densities related to Bayesian Logistic Regression tasks on Credits, Cancer, Ionosphere, Sonar datasets. We follow a similar setting as [5] and display in the **new Appendix I.5.5** the results from LRDS and its competitors. In particular, we observe that LRDS is on par with previous methods, and that EBM-LRDS may be more performant than GMM-LRDS in certain settings.  Note however that those targets don’t exhibit explicit multi-modality features, which makes this setting less appealing for the scope of this paper.
> >
> > [0] Qinsheng Zhang, & Yongxin Chen (2022). Path Integral Sampler: A Stochastic Control Approach For Sampling. In International Conference on Learning Representations.
> >
> > [1] Francisco Vargas, Will Sussman Grathwohl, & Arnaud Doucet (2023). Denoising Diffusion Samplers. In The Eleventh International Conference on Learning Representations .
> >
> > [2] Lorenz Richter, & Julius Berner (2024). Improved sampling via learned diffusions. In The Twelfth International Conference on Learning Representations.
> >
> > [3] Francisco Vargas, Shreyas Padhy, Denis Blessing, & Nikolas Nusken (2024). Transport meets Variational Inference: Controlled Monte Carlo Diffusions. In The Twelfth International Conference on Learning Representations.
> >
> > [4] Phillips, A., Dau, H.D., Hutchinson, M., De Bortoli, V., Deligiannidis, G., & Doucet, A. (2024). Particle Denoising Diffusion Sampler. In Proceedings of the 41st International Conference on Machine Learning (pp. 40688–40724). PMLR.
> >
> > [5] Blessing, D., Jia, X., Esslinger, J., Vargas, F., & Neumann, G. (2024). Beyond ELBOs: A Large-Scale Evaluation of Variational Methods for Sampling. In Proceedings of the 41st International Conference on Machine Learning (pp. 4205–4229). PMLR.

---

### Author Response · Authors · 2024-11-26

Dear reviewers,

As the period allowing for modifications on the manuscript is approaching its end, we would like to know whether our rebuttal adressed your concerns. If there are any remaining issues, we kindly ask you to share them as soon as posssible in order to give us sufficient time to respond thoroughly.

Thank you again for your time and valuable feedback!

---

### Meta-Review · Area_Chair_o3LX · 2024-12-19

**Metareview:**

This paper introduces Learned Reference-based Diffusion Sampler (LRDS), a two-step methodology to address hyperparameter tuning challenges in score-based diffusion sampling, particularly for multi-modal distributions. LRDS leverages prior knowledge by first learning a reference diffusion model tailored for multimodality and then using it to guide the training of a diffusion-based sampler. Experimental results show that LRDS effectively exploits prior knowledge and outperforms competing methods on complex target distributions.

The paper has an original contribution, with detailed theoretical analysis and good experimental performances. Therefore, I am recommending for an acceptance of this paper.

**Additional Comments On Reviewer Discussion:**

One of the recurrent  weakness expressed in the reviews was the lack on experiments on real-world data. Authors conducted additionnal experiments in this direction, that are satisfactory.

---

### Decision · Program_Chairs · 2025-01-22

Accept (Poster)